# TusoAI: Agentic Optimization for Scientific Methods

**Alistair Turcan**[1], **Kexin Huang**[2,3], **Lei Li**[1], **Martin Jinye Zhang**[1]
[1]Carnegie Mellon University    [2]Stanford University    [3]Phylo
aturcan@cs.cmu.edu, martinzh@andrew.cmu.edu

## Abstract

Scientific discovery is often slowed by the manual development of computational tools needed to analyze complex experimental data. Building such tools is costly and time-consuming because scientists must iteratively review literature, test modeling and scientific assumptions against empirical data, and implement these insights into efficient software. Large language models (LLMs) have demonstrated strong capabilities in synthesizing literature, reasoning with empirical data, and generating domain-specific code, offering new opportunities to accelerate computational method development. Existing LLM-based systems either focus on performing scientific analyses using existing computational methods or on developing computational methods or models for general machine learning without effectively integrating the often unstructured knowledge specific to scientific domains. Here, we introduce TusoAI, an agentic AI system that takes a scientific task description with an evaluation function and autonomously develops and optimizes computational methods for the application. TusoAI integrates domain knowledge into a knowledge tree representation and performs iterative, domain-specific optimization and model diagnosis, improving performance over a pool of candidate solutions. We conducted comprehensive benchmark evaluations demonstrating that TusoAI outperforms state-of-the-art expert methods, MLE agents, and scientific AI agents across diverse tasks. Applying TusoAI to two key open problems in genetics improved existing computational methods and uncovered new biology missed by previous methods. Our code is publicly available at https://github.com/Alistair-Turcan/TusoAI.

## 1 Introduction

Scientific discoveries are often bottlenecked by the slow, manual development of computational tools needed to analyze experimental data. For example, genetics studies have uncovered tens of thousands of disease-associated variants, yet robust computational methods are critically needed to harmonize multi-modal, multi-scale data and uncover the underlying mechanisms (Lappalainen & MacArthur, 2021). Developing such tools is slow and costly because scientists must iteratively (i) review extensive literature, (ii) test modeling and scientific assumptions against empirical data, and (iii) implement these insights into efficient, scalable code. For instance, building robust computational methods to link enhancers with target genes from single-cell multiome data has taken multiple expert groups many years (Dorans et al., 2025), hindered by challenges such as *cis*-regulatory modeling, latent confounding, noisy data, and computational scalability. Large language models (LLMs) have demonstrated strong capabilities in performing human-like analysis (Luo et al., 2025), such as synthesizing relevant literature (Asai et al., 2024), reasoning about biological and modeling assumptions using empirical data (Gao et al., 2024), and generating efficient, domain-specific code (Rasheed et al., 2025). Integrating LLMs with scientific domain knowledge and iterative data experimentation holds great promise to accelerate computational method development, thereby advancing discoveries in science and medicine.

Existing work has produced general-purpose AI agents across scientific domains, including biomedicine (Huang et al., 2025; Jin et al., 2025) and chemistry (M. Bran et al., 2024). These systems primarily focus on performing scientific data analyses rather than developing new computational methods; the former involves assembling and executing pipelines of data formatting and

existing tools, whereas the latter requires creating new algorithms or models for specific pipeline steps, involving substantial design, optimization, and validation. In parallel, several studies have developed machine learning engineering (MLE) agents that can design new algorithms for general ML applications (Guo et al., 2024; Trirat et al., 2024; Jiang et al., 2025; Nam et al., 2025), but these approaches do not address domain-specific challenges inherent in scientific research. Developing *AI agents for scientific method development* that integrate structured domain knowledge and systematically explore data-specific assumptions has considerable potential to accelerate the creation of robust computational methods for science and medicine.

Here, we introduce TusoAI, an agentic AI system that takes a scientific task description with an evaluation function, and autonomously develops and optimizes computational methods for the application (Figure 1). TusoAI mimics a scientist's cycle of method development, integrating structured domain knowledge with iterative, domain-specific optimization and model diagnosis, improving performance over a pool of candidate solutions. We demonstrate that TusoAI achieves superior performance across a range of algorithmic, statistical, machine learning, and deep learning applications in science. Our key contributions are:

1. We develop TusoAI, an AI agent specifically tailored for scientific method discovery by integrating structured domain knowledge.

2. We propose a novel framework, featuring (i) knowledge tree for structured representation of domain knowledge, (ii) hierarchical planning with Bayesian updates to balance solution quality and diversity, and (iii) fine-grained generation that integrates model optimization with diagnostic feedback.

3. We benchmark TusoAI on 6 single-cell analysis tasks and 5 scientific deep learning tasks, consistently outperforming baseline methods and frequently surpassing existing expert-designed algorithms.

4. Applying TusoAI to two key open problems in genetics improved existing computational methods and uncovered new biology missed by existing methods.

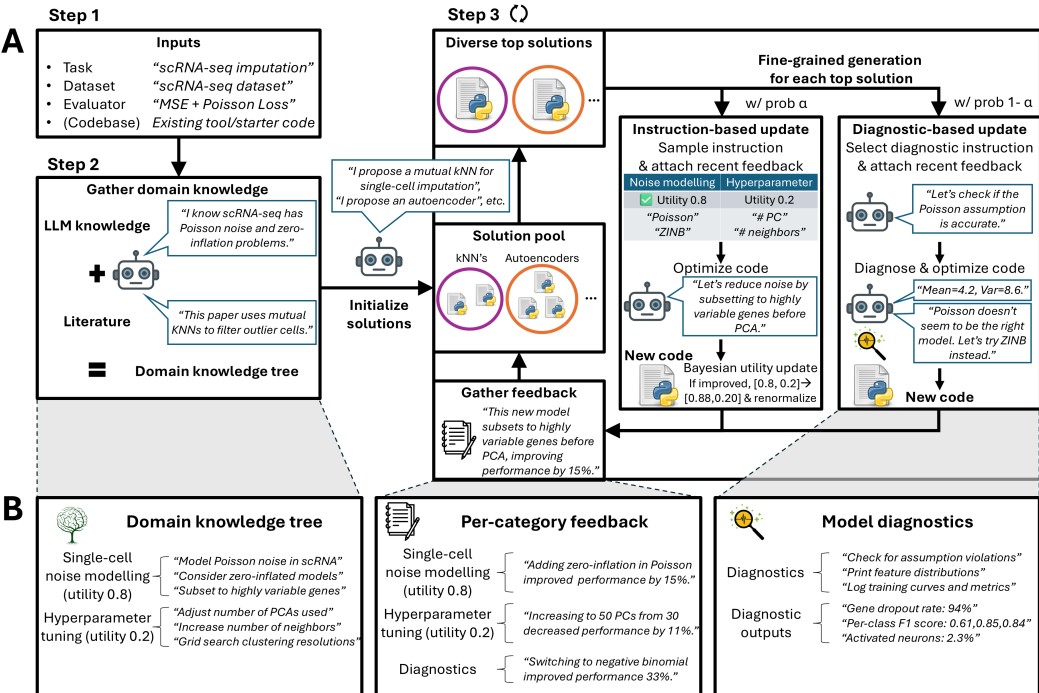

Figure 1: **Scientific method development with TusoAI. (A)** Method overview. **(B)** Example domain knowledge tree (categories and instructions per category), feedback, and diagnostics.

## 1.1 RELATED WORK

**LLM-based scientific AI agents.** Several works have developed general-purpose AI agents capable of autonomously executing various scientific research tasks. Biomni (Huang et al., 2025) provides a unified agentic environment with tools and databases spanning 25 biomedical domains, integrating LLM reasoning with retrieval-augmented planning and code execution to compose complex workflows. Stella (Jin et al., 2025) employs a multi-agent architecture for autonomous biomedical data analysis, achieving self-evolution by dynamically updating its template library and tool collection. ChemCrow (M. Bran et al., 2024) is a chemistry-focused agent that integrates 18 expert-designed tools and follows the "Thought, Action, Action Input, Observation" format to iteratively reason toward answers. These methods emphasize end-to-end data analysis with established tools, whereas our work focuses on developing new computational methods for domain-specific tasks. Other works have leveraged LLMs to develop application-specific methods, such as single-cell perturbation prediction (Tang et al., 2025), diagnosis prediction (Tan et al., 2025), and mathematical discovery (Romera-Paredes et al., 2024). In contrast, TusoAI targets computational method development across scientific tasks. InternAgent (Team et al., 2025) and its precursor Dolphin (Yuan et al., 2025) iteratively evolve and implement research ideas through an optimization process augmented with literature review. As a concurrent effort, Aygün et al. (2025) combine LLMs with tree search and existing model ensembles to improve scientific algorithms, addressing a similar problem but with a different approach from ours, which integrates a domain knowledge tree with fine-grained iterative optimization and Bayesian updates. As their code is not publicly available, direct comparison is not possible, but a key distinction of our work is to perform fine-grained optimizations with domain knowledge that do not require existing models and can operate on a small portion of a much larger method.

**LLM-based general machine learning agents.** Several recent works have developed AI agents for general machine learning engineering. AIDE (Jiang et al., 2025) frames ML engineering as a code optimization problem, combining an LLM with tree search to iteratively improve solutions. R&D Agent (Yang et al., 2025) similarly explores ML architectures in a dynamic feedback loop. DS-Agent (Guo et al., 2024) combines an LLM with case-based reasoning (CBR), retrieving potentially successful solutions from top-ranked Kaggle solutions, and refining them through iterative optimization. MLE-STAR (Nam et al., 2025) retrieves candidate models from the web to form an initial solution, then improve it by targeting specific ML components and ensembling. AutoML-Agent (Trirat et al., 2024) employs retrieval-augmented planning and multi-agent coordination to generate an optimal plan, but executes the plan once without iterative refinement. These methods are less suited to scientific method development, where domain knowledge is unstructured, existing ML models may be unavailable, and search spaces are continually evolving. We address these challenges through structured domain knowledge representation and a novel hierarchical planning procedure with Bayesian updates during iterative optimization.

**Classical automatic machine learning (AutoML) frameworks.** Classical (non-LLM) AutoML frameworks aim to construct high-performing ML models from scratch by searching over key components such as feature preprocessing, model architectures, hyperparameters, and pipeline composition. Notable examples include auto-sklearn (Feurer et al., 2015), H2O (LeDell et al., 2020), AutoGluon (Erickson et al., 2020), and TPOT (Olson & Moore, 2016). Within deep learning, neural architecture search (NAS) methods specialize in optimizing neural architectures, with examples such as DARTS (Liu et al., 2018) and AMBER (Zhang et al., 2021). While effective for standard ML tasks, these approaches are constrained by predefined search spaces and are less suited to scientific domains, where domain knowledge and optimization objectives are unstructured and continually evolving, making LLM-based agents a more natural fit as they can pair a principled optimization objective with heuristic search procedures.

## 2 PROBLEM FORMULATION

We consider the problem of automatic scientific algorithm optimization with LLMs. Given a general solution space $\mathcal{S}^{\text{full}}$ (e.g., all Python scripts) and an evaluator $h(\cdot) : \mathcal{S}^{\text{full}} \mapsto \mathbb{R}$, the objective is to find the optimal solution $s^* = \arg\max_{s \in \mathcal{S}^{\text{full}}} h(s)$. $h(\cdot)$ can be any evaluation metric, such as AUC, average of several metrics, or domain-specific measures (e.g., enrichment of inferred disease genes against an expert-curated set). We assume access to a task description $\mathcal{T}$ (e.g., "single-cell RNA-seq

imputation"), a domain-specific knowledge base (e.g., scientific papers), and a general LLM that can be instantiated as agents. The agent can, for example, summarize domain priors from $\mathcal{T}$, retrieve information from the knowledge base, and refine a candidate solution $s$ based on instructions. The goal is to iteratively implement and improve solutions to maximize $h(\cdot)$ within a time budget. We consider two settings: a *cold start*, where optimization begins from scratch, and a *warm start*, where an initial solution $s_{\text{init}}$ (e.g., a state-of-the-art method) is given for further improvement.

## 3 METHODS

TusoAI takes as input a task description $\mathcal{T}$, a dataset $\mathcal{D}$, an evaluator $h(\cdot)$, and optionally an initial solution $s_{\text{init}}$. It outputs an optimized solution $s^*$ (Algorithm 1, variables described in Appendix Table A). TusoAI operates on only a single function of an arbitrarily large codebase, allowing it to flexibly build upon scientific methods with extensive scaffolding. Developing computational methods for scientific domains poses several challenges. First, domain-specific knowledge is often unstructured, which we address using a knowledge tree that organizes information into categories and within-category instructions. Second, approaches and optimization strategies can vary widely, which we manage through hierarchical planning with Bayesian updates to promote diversity while ensuring solution quality. Third, understanding complex data patterns is challenging, which we mitigate with fine-grained generation that integrates model optimization with diagnostic feedback.

TusoAI consists of 3 steps. First, it gathers domain knowledge by summarizing key scientific papers, ensuring that optimization instructions reflect established best practices and recent advances rather than relying solely on LLM priors. Second, it builds a two-level knowledge tree of structured instructions: (1) categories of optimization strategies and (2) specific instructions within each category, promoting both diversity and relevance. Categories and instructions are first drafted by the LLM and then refined through additional LLM queries in conjunction with paper summaries to ensure diversity and scientific rigor; we also predefine a diagnostic category $\mathcal{I}_{\text{diag}}$ to guide data logging and model diagnosis. Third, after initializing candidate solutions, it iteratively selects diverse top performers and improves them through either instruction-based or diagnostic-based optimization. Due to the knowledge tree structure, instruction categories can be sampled adaptively via a Bayesian strategy informed by past performance, while feedback comparing new and prior solutions helps discourage repetition. Examples of instructions generated are provided at Appendix C.

---

**Algorithm 1** TusoAI

**Input:** Task $\mathcal{T}$; dataset $\mathcal{D}$; evaluator $h(\cdot)$; optional initial solution $s_{\text{init}}$.
**Hyperparameters:** Time budget $T_{\text{budget}}$ (default 8 hrs).
1: **Gather domain knowledge:** $\mathcal{P} \leftarrow A_{\text{paper}}(\mathcal{T})$       ▷ Paper summaries
2: **Build structured instructions:**
  $\mathcal{C}, \{\pi_c\}_{c \in \mathcal{C}} \leftarrow \text{DraftThenRefine}(A_{\text{cate}}, \mathcal{T}, \mathcal{P})$   ▷ Instruction categories with probabilities
  **For each** $c \in \mathcal{C}$:       ▷ Per-category instructions and feedback
   $\mathcal{I}_c \leftarrow \text{DraftThenRefine}(A_{\text{instr}}, \mathcal{T}, \mathcal{P}, c), \mathcal{F}_c \leftarrow \emptyset$
  $\mathcal{I}_{\text{diag}} \leftarrow \mathcal{I}_{\text{diag}}^{\text{predefined}}, \mathcal{F}_{\text{diag}} \leftarrow \emptyset$    ▷ Diagnostic instructions (predefined) and feedback
3: **Initialize solutions:** $\mathcal{S} \leftarrow A_{\text{init}}(\mathcal{T}, \mathcal{P}, s_{\text{init}}); N_{\text{top}} \leftarrow |\mathcal{S}|$
4: **While** wall-clock time $< T_{\text{budget}}$ **do**
5:   Select $N_{\text{top}}$ diverse top solutions from $\mathcal{S}$
6:   **for each** top $s$ **do**
7:    **if** Bernoulli($\alpha$) **do**     ▷ Instruction-based optimization, defualt $\alpha = 0.8$
8:     Sample $c \sim \text{Cat}(\{\pi_c\}_{c \in \mathcal{C}})$; optimize $s' \leftarrow A_{\text{optim}}(s, \mathcal{I}_c, \mathcal{F}_c)$
9:     **if** $h(s') > h(s)$ **do** $\pi_c \leftarrow 1.1\pi_c$; renormalize $\{\pi_c\}_{c \in \mathcal{C}}$   ▷ Bayesian update utility
10:     $\mathcal{F}_c \leftarrow \mathcal{F}_c \cup \{A_{\text{feedback}}(s, s')\}$     ▷ Gather category-specific feedback
11:    **else**         ▷ Diagnostic-based optimization
12:     $s' \leftarrow A_{\text{diag}}(s, \mathcal{D}, \mathcal{I}_{\text{diag}}, \mathcal{F}_{\text{diag}})$    ▷ Get model&data log info then optimize
13:     $\mathcal{F}_{\text{diag}} \leftarrow \mathcal{F}_{\text{diag}} \cup \{A_{\text{feedback}}(s, s')\}$     ▷ Gather diagnostic feedback
14:    $\mathcal{S} \leftarrow \mathcal{S} \cup \{s'\}$
15:   $N_{\text{sol}} \leftarrow \max(1, N_{\text{top}} - 1)$ every 2 rounds
16: **return** $s^* \in \arg\max_{s \in \mathcal{S}} h(s)$

---

**Step 1: Gather domain knowledge.** TusoAI first retrieves up to 10 key papers from Semantic Scholar (Allen Institute for AI, 2025) relevant to $\mathcal{T}$, ranked by their citation count. For each paper, an agent $A_{\text{paper}}$ creates a 15-point technical summary from the abstract and iteratively refines it using each paragraph of the paper's Methods section (up to 1,200 words to focus solely on technical content without relying on costly deep research agents parsing the entire document). This produces $\mathcal{P} = \{\mathcal{P}_i\}$, where each $\mathcal{P}_i$ is a refined 15-point summary of paper $i$'s method.

**Step 2: Build structured instructions.** TusoAI uses a draft-then-refine strategy to construct optimization categories, where an agent $A_{\text{cate}}$ first drafts candidate categories from the task description $\mathcal{T}$, then refines them by iterating through each paper summary $\mathcal{P}_i \in \mathcal{P}$, adjusting existing categories or adding new ones as needed. Categories are task-specific and can be general (e.g., "regularization", "model architectures") or domain-specific (e.g., "single-cell noise modeling", "genetic feature interactions"). Each category is assigned a probability $\pi_c$ representing its utility in the optimization process; $\pi_c$ is initialized by $A_{\text{cate}}$ so that tasks earlier in the pipeline (e.g., "feature preprocessing") receive higher weight than later ones (e.g., "hyperparameter tuning"). Similarly, TusoAI uses a draft-then-refine strategy to initialize instructions for each category, where an agent $A_{\text{instr}}$ first drafts 10 candidate instructions $\mathcal{I}_c$ from the task description $\mathcal{T}$. These instruction lists are then refined by incorporating 10 additional instructions for each paper summary $\mathcal{P}_i \in \mathcal{P}$. For feedback, TusoAI initializes an empty list $\mathcal{F}_c \leftarrow \emptyset$ for each category, which is updated with category-specific feedback during optimization. A special predefined diagnostic category $\mathcal{I}_{\text{diag}}$ provides instructions for logging diagnostic information useful for model updates, with its own feedback list $\mathcal{F}_{\text{diag}}$. See Appendix C, G for an example of constructed task information and Appendix U for prompt templates used to generate this information.

**Step 3.1: Initialize solutions.** The initialization agent $A_{\text{init}}$ drafts 5 candidate solution descriptions from $\mathcal{T}$ and iteratively refines them using each paper summary in $\mathcal{P}$, adding new descriptions or improving existing ones (e.g., "zero-inflated Poisson with kNN smoothing"). These are basic descriptions designed to start from scratch and explore to avoid simply re-implementing existing solutions. It then attempts to implement and debug each solution; those that successfully compile form the initial solution pool $\mathcal{S}$. Each implementation attempt is limited to 10 minutes with up to 4 bug-fix attempts.

**Step 3.2: Iterative optimization.** Given the current solution set $\mathcal{S}$, TusoAI selects diverse top solutions by clustering them based on code-text similarity and, within each cluster, choosing the shortest solution whose performance is within 0.1% of the cluster's best model; this helps discourage overfitting and randomness while maintaining diversity and concise code. For each cluster's top solution $s$, TusoAI performs either instruction-based optimization (80% probability) or diagnostic-based optimization (20% probability). The resulting solution $s'$ is added to the pool $\mathcal{S} \leftarrow \mathcal{S} \cup s'$. Each implementation attempt is limited to 10 minutes with up to 2 bug-fix attempts. This time regularization ensures the optimization period is not wasted on a few inefficient implementations, and encourages the final method to be scalable.

- **Instruction-based optimization.** The optimization agent $A_{\text{optim}}$ selects an instruction by first sampling an instruction category $c \sim \text{Cat}(\{\pi_c\}_{c \in \mathcal{C}})$, then uniformly draw 3 candidate instructions from $\mathcal{I}_c$, and finally choosing the most promising among them. It then optimizes $s$ to produce $s'$ using the selected instruction in conjunction with 5 most recent feedback entries from $\mathcal{F}_c$. If $h(s') > h(s)$, TusoAI performs a Bayesian-style update to the category utility by setting $\pi_c \leftarrow 1.1\pi_c$ and renormalizing $\{\pi_c\}_{c \in \mathcal{C}}$, representing the prior belief that this category currently contains useful instructions. Finally, the feedback agent $A_{\text{feedback}}$ summarizes the change from $s$ to $s'$ and appends it to $\mathcal{F}_c$ (e.g., "this optimization constructed a kNN on the top 50 PC's rather than on all genes, improving performance by 15%").

- **Diagnostic-based optimization.** The optimization agent $A_{\text{optim}}$ selects an instruction by first uniformly draw 3 candidate instructions from $\mathcal{I}_{\text{diag}}$ and then choosing the most promising among them (e.g., "training curves", "distribution checks", "validation of assumptions"). It then diagnoses and improves $s$ to produce $s'$ using the selected instruction in conjunction with 5 most recent feedback entries from $\mathcal{F}_c$: it runs $s$ to collect diagnostic logs and then uses this information to produce an improved model $s'$. This represents a scientist diagnosing their method's intermediate outputs to further improve upon it. Finally, the feedback agent $A_{\text{feedback}}$ summarizes the change from $s$ to $s'$ and appends it to $\mathcal{F}_{\text{diag}}$.

## 4 EXPERIMENTS

We evaluate TusoAI on 11 scientific applications spanning diverse domains with both ML and non-ML components, including 6 single-cell analysis tasks (Luecken et al., 2025) and 5 scientific deep learning tasks (Tu et al., 2022). The single-cell tasks include denoising (Denoise), cell-type label projection (Label), batch integration (Batch), identification of spatially variable genes (SVG), decomposition of spot-level spatial data into specific cell types (Decomp), and dimensionality reduction for visualization (Visual). The scientific deep learning tasks include omnidirectional vision (Spherical), prosthetics control (NinaPro), medical diagnostics (ECG), earth monitoring (Satellite), and genetic prediction (DeepSea). In each task, we run TusoAI for 8 hours (as per related work (Miller et al., 2025; Aygün et al., 2025)), optimizing performance on a validation dataset, and evaluating final performance on separate testing datasets. Task descriptions used are concise (e.g., "single-cell batch integration") and extracted from the original benchmarks. We define "Avg." and "Avg. Rank" as the average performance across tasks for a method, and the average rank in each task, respectively. We additionally assess code diversity, defined as the text similarity between generated code (Appendix M) and mean time to optimize, defined as the average position of each optimization over the 8 hours, representing how quickly optimizations are achieved.

We conduct comprehensive ablation studies to assess the contribution of different components of TusoAI (Subsection 4.2), and two case studies demonstrating how TusoAI can reveal new biological insights in genetics (Section 5). Full details on experimental setup and evaluation metrics used are in Appendix D, E for single-cell and deep learning tasks, respectively.

**Baseline methods.** We compare TusoAI against the state-of-the-art MLE agent AIDE (Jiang et al., 2025), scientific agents Biomni (Huang et al., 2025) and ChatGPT-Agent (OpenAI, 2025), and top-performing, published application-specific methods. Biomni (LLM backbone Claude-4-Sonnet) and ChatGPT-agent (LLM backbone GPT-5) are used to iteratively build models on data for single-cell tasks; for deep learning tasks, where Biomni and ChatGPT-agent were unable to operate, we substitute the best of ten models constructed by Claude-4-Sonnet and GPT-5. GPT-4o-mini and GPT-5 are accessed through the OpenAI API, and all others with OpenRouter. For application-specific baselines, we use the "top-performing expert" method for single-cell tasks (Luecken et al., 2025), and all baseline methods, including expert models and NAS methods, for the scientific deep learning tasks (Tu et al., 2022). This set of baselines is consistent with related work in scientific optimization (Aygün et al., 2025) and a recent benchmark that identified AIDE and Biomni as top-performers (Miller et al., 2025). We note that most MLE agentic methods cannot apply to scientific tasks outside the standard ML setup. For full details on baseline implementation and setup, see Appendix F.

### 4.1 PERFORMANCE ACROSS BENCHMARK EXPERIMENTS

Results for the 6 single-cell tasks and 5 scientific deep learning tasks are reported in Tables 1 and 2. We reached 2 main conclusions. First, TusoAI consistently outperformed baseline methods across benchmarks when generating code from scratch (average rank of 1.2 for single-cell tasks and 2.8 for scientific deep learning tasks, vs. 3.0 and 4.0 for the second best, resp.). Second, the methods constructed by TusoAI are novel rather than simple re-implementations of existing approaches or calls to standard packages. Examples include: (i) in single-cell denoise, TusoAI designed a non-negative matrix factorization (NMF) approach that models dropout rates, Poisson noise, and performs iterative refinement, distinct from the only other NMF-based approach in the OpenProblems benchmark, ALRA (Linderman et al., 2022); (ii) in SVG, TusoAI adapted known techniques such as modeling expression as a function of spatial coordinates and neighborhood summaries to create a custom, high-performing method; (iii) in Satellite, TusoAI combined preprocessing, training procedures, loss functions, and ensembling techniques to build the top-performing model; and (iv) in Spherical, TusoAI fine-tuned layers of ResNet-50 and augmented the data with random flips and rotations. See Appendix H for full justification of why these new methods are novel. Third, all methods constructed by TusoAI are computationally efficient (<3 minutes for single-cell tasks and <8 minutes for deep learning tasks, Appendix I), owing to the runtime constraints imposed during optimization.

We conducted 2 secondary analyses. First, we assessed the diversity of code produced by TusoAI and AIDE over 8 hours of optimization, quantifying code diversity using cosine similarity of text embeddings between each candidate and its 10 previous and 10 subsequent iterations (Figure 2A).

|  | Denoise | Label | Batch | SVG | Decomp | Visual | Avg | Avg rank |
|---|---|---|---|---|---|---|---|---|
| Expert | 0.28 | 0.85 | 0.71 | 0.66 | 0.49 | **0.44** | 0.57 | 3.7 |
| AIDE* | 0.30 | 0.87 | 0.71 | 0.73 | 0.06 | **0.44** | 0.52 | 3.0 |
| Biomni* | 0.16 | **0.89** | 0.82 | 0.16 | 0.53 | 0.35 | 0.49 | 3.7 |
| ChatGPT-Agent* | 0.03 | 0.81 | **0.83** | 0.60 | **0.74** | 0.38 | 0.57 | 3.5 |
| TusoAI* | **0.35** | **0.89** | **0.83** | **0.80** | 0.64 | **0.44** | **0.66** | **1.2** |

Table 1: **Single-cell benchmarks.** We report performance across 6 single-cell tasks. "*" denotes agentic methods. Best in **bold**, second-best underlined. 95% CIs across 3 random seeds all under 0.01 and thus not shown.

|  | Spherical | NinaPro | ECG | Satellite | DeepSEA | Avg | Avg rank |
|---|---|---|---|---|---|---|---|
| WRN default | $0.14 \pm 0.01$ | **0.93** $\pm 0.00$ | $0.57 \pm 0.00$ | $0.85 \pm 0.00$ | $0.60 \pm 0.00$ | 0.62 | 6.9 |
| DenseNAS random | $0.29 \pm 0.02$ | $0.92 \pm 0.01$ | $0.58 \pm 0.00$ | $0.86 \pm 0.00$ | $0.60 \pm 0.00$ | 0.65 | 5.4 |
| DenseNAS original | $0.27 \pm 0.01$ | $0.90 \pm 0.01$ | $0.60 \pm 0.00$ | $0.86 \pm 0.01$ | $0.60 \pm 0.00$ | 0.65 | 5.8 |
| Perceiver IO | $0.17 \pm 0.00$ | $0.78 \pm 0.02$ | $0.34 \pm 0.00$ | $0.84 \pm 0.00$ | $0.62 \pm 0.00$ | 0.55 | 9.6 |
| XGBoost | $0.03 \pm 0.00$ | $0.78 \pm 0.01$ | $0.44 \pm 0.00$ | $0.64 \pm 0.00$ | $0.50 \pm 0.00$ | 0.48 | 11.6 |
| WRN ASHA | $0.25 \pm 0.00$ | **0.93** $\pm 0.01$ | $0.57 \pm 0.00$ | $0.84 \pm 0.01$ | $0.59 \pm 0.00$ | 0.63 | 7.1 |
| DARTS | **0.52** $\pm 0.03$ | $0.82 \pm 0.01$ | $0.66 \pm 0.00$ | $0.87 \pm 0.00$ | $0.68 \pm 0.00$ | **0.71** | 4.0 |
| AMBER | N/A | N/A | 0.67 $\pm 0.00$ | $0.87 \pm 0.00$ | $0.68 \pm 0.00$ | N/A | N/A |
| Expert | $0.33 \pm 0.01$ | $0.91 \pm 0.01$ | **0.72** $\pm 0.00$ | $0.80 \pm 0.00$ | $0.70 \pm 0.00$ | 0.69 | 4.6 |
| AIDE* | $0.16 \pm 0.01$ | $0.86 \pm 0.00$ | $0.52 \pm 0.01$ | $0.83 \pm 0.01$ | $0.57 \pm 0.00$ | 0.59 | 9.8 |
| GPT-5* | $0.36 \pm 0.00$ | $0.89 \pm 0.00$ | $0.58 \pm 0.03$ | $0.86 \pm 0.01$ | $0.66 \pm 0.00$ | 0.67 | 5.8 |
| Claude-4-Sonnet* | $0.40 \pm 0.00$ | $0.90 \pm 0.00$ | $0.50 \pm 0.01$ | 0.88 $\pm 0.00$ | **0.73** $\pm 0.00$ | 0.68 | 4.6 |
| TusoAI* | 0.42 $\pm 0.01$ | $0.90 \pm 0.00$ | $0.61 \pm 0.00$ | **0.89** $\pm 0.01$ | 0.70 $\pm 0.00$ | 0.70 | **2.8** |

Table 2: **Scientific deep learning benchmarks.** We report performance across 5 scientific deep learning tasks. "*" denotes agentic methods. Performance of non-agentic methods extracted from NASBENCH-360 and transformed to be between 0 and 1 and higher is better. Best in **bold**, second-best underlined. 95% CIs provided across 3 random seeds.

We note that constructing diverse code to escape local optima is often an important consideration in agentic code optimization (Romera-Paredes et al., 2024; Nam et al., 2025; Aygün et al., 2025). TusoAI achieved substantially higher diversity than AIDE throughout the optimization process. For example, in the batch integration benchmark, AIDE repeatedly proposed small variations of UMAP-based dimensionality reduction, whereas TusoAI explored a wide variety of dimensionality reduction, transformation, and scaling techniques. This higher diversity is perhaps due to TusoAI's instruction sampling, feedback, and diagnosis procedures, which encourage diverse solutions. We validate the importance of code diversity in generating strong optimizations (Appendix M). In contrast, AIDE promotes incremental changes at each optimization step to facilitate traceability, which may bias the search toward local tuning rather than full exploration. Second, we characterized the optimization trajectory of TusoAI on the single-cell denoising task (Figure 2B). We identified 5 key developments that led to strong performance: (1) introducing NMF, (2) modeling dropout, (3) modeling Poisson noise, (4) adding iterative refinement, and (5) incorporating a sparsity-balancing step. Notably, during optimization, TusoAI generated many methods that reduced performance before converging on high-performing solutions. Together with the feedback mechanism, this broad exploration allowed TusoAI to efficiently search the solution space and identify top-performing methods. See Appendix J, K for the optimization trajectories in other tasks.

## 4.2 ABLATION STUDIES

We conducted extensive ablation studies to evaluate the impact of each novel component of TusoAI by removing one at a time, including: (i) removing the categorical structure and placing all instructions and feedback into a single category (No categories); (ii) disabling the Bayesian sampling strategy across categories (No Bayesian); (iii) disabling the model diagnosis capability (No diagnosis); and (iv) discarding domain knowledge altogether, such that each iteration simply applies a generic instruction (e.g., "Optimize this model"; No knowledge). Removing these components each negatively affected overall performance (Table 3). We attribute this to reduced code diversity

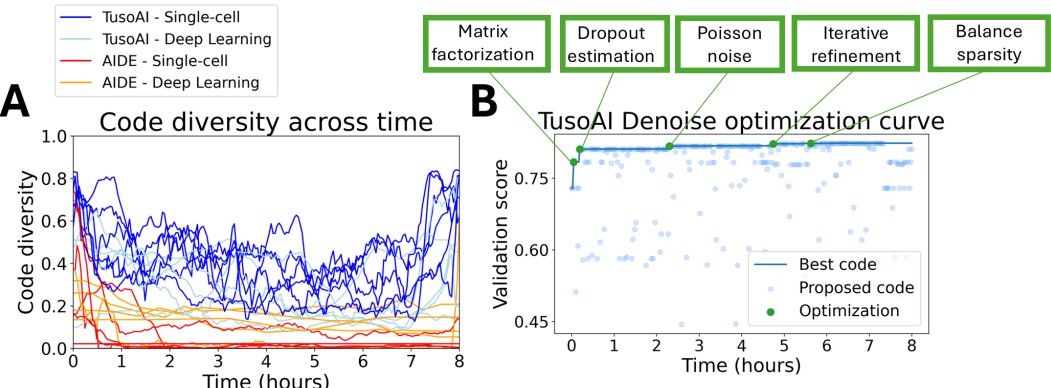

Figure 2: **Behavior of code generated by TusoAI. (A)** Code diversity of TusoAI and AIDE over optimization time, as measured by $1-$ cosine similarity. Each line corresponds to a dataset. **(B)** Performance of the proposed optimization and the best code over optimization time for a representative task "Denoise". Key optimization changes with their occurrence times are annotated.

(mean diversity 0.48 vs. 0.44/0.39/0.38/0.33 for ablated versions, resp.) and computational efficiency (mean time to optimize 2.3 hours vs. 2.4/3.0/2.6/2.4 for ablated versions, resp.). Removing domain knowledge had the strongest impact on performance and diversity, while removing Bayesian updates (thus sampling categories uniformly) most reduced TusoAI's computational efficiency. See Appendix L, N, O for TusoAI's general stability across replicates, further ablation details and ablations varying literature information used.

We next assessed the impact of LLM backbones used by TusoAI, testing across 5 different LLMs: low-latency models GPT-4o-mini (default) and Claude-3.5-Haiku; state-of-the-art reasoning models GPT-5 and Claude-4-Sonnet; and open-source GPT-oss-120b. Results are shown in Table 3. Apart from GPT-oss-120b, TusoAI achieved relatively consistent performance across all LLMs for most tasks, demonstrating robustness. Interestingly, LLMs such as GPT-5 and Claude-4-Sonnet did not consistently outperform their lower-latency counterparts, GPT-4o-mini and Claude-3.5-Haiku. This may be because, while reasoning models can construct highly complex code, their tendency to over-build (e.g., each of GPT-5's methods are 300+ lines of code) makes subsequent iterations difficult to refine; in contrast, low-latency but capable models like GPT-4o-mini and Claude-3.5-Haiku, when paired with an appropriate system design, performed just as well at a fraction of the cost (e.g., optimizing denoising for 8 hours costs 0.24$ with GPT-4o-mini and 22.3$ with GPT-5). See Appendix P, Q for further LLM analysis and cost details, respectively.

| | Denoise | SVG | Decomposition | ECG | Satellite | Avg | Avg rank |
|---|---|---|---|---|---|---|---|
| TusoAI (default) | 0.35 | **0.80** | 0.64 | 0.61 | **0.89** | **0.66** | **2.0** |
| No categories | 0.09 | 0.72 | 0.56 | 0.63 | 0.86 | 0.57 | 3.2 |
| No Bayesian | **0.36** | 0.77 | 0.22 | 0.57 | 0.84 | 0.55 | 3.4 |
| No diagnosis | 0.26 | 0.77 | **0.68** | 0.63 | 0.86 | 0.64 | **2.0** |
| No knowledge | 0.17 | 0.51 | 0.07 | **0.68** | 0.85 | 0.46 | 3.8 |
| GPT-4o-mini (default) | 0.35 | **0.80** | 0.64 | 0.61 | **0.89** | 0.66 | 2.2 |
| GPT-5 | 0.31 | **0.80** | **0.82** | **0.67** | 0.87 | **0.69** | 2.2 |
| Claude 3.5 Haiku | **0.41** | 0.78 | 0.70 | 0.63 | **0.89** | 0.68 | **1.8** |
| Claude 4 Sonnet | 0.32 | 0.78 | 0.53 | 0.59 | 0.84 | 0.61 | 4.2 |
| GPT-oss-120b | 0.39 | 0.74 | 0.13 | 0.61 | 0.85 | 0.54 | 3.8 |

Table 3: **Ablation studies (top) and varying LLM backbone (bottom).** Best in **bold**, second-best underlined.

## 5    CASE STUDIES IN GENETICS

We applied TusoAI to address 2 key challenges in genetics: detecting disease-critical cell populations and linking genetic variants to their target genes; these are central to understanding disease etiology but limited by current computational models. We initialized TusoAI with state-of-the-art methods (scDRS (Zhang et al., 2022b) and pgBoost (Dorans et al., 2025), resp.) and evaluated its ability to improve these approaches and generate new biological insights. We consider the same quantitative evaluation procedure and level of validation for new discoveries as in the original papers. We note these codebases are too large to easily use with existing agentic approaches that require editing the entire Python script. See full details of how we applied TusoAI to each task in Appendix R, S for scDRS and pgBoost, respectively. An additional case study of how TusoAI may optimize an existing deep learning model outside of biology is in Appendix T.

**Detecting disease-critical cell populations.**    scDRS (Zhang et al., 2022b) is a state-of-the-art method that integrates genome-wide association studies (GWAS) with single-cell RNA-seq (scRNA-seq) to identify disease-associated cell populations, but its power is limited by the high noise of scRNA-seq data. Here, we apply TusoAI in conjunction with scDRS and task it with optimizing scDRS's association scoring function. Results are reported in Figure 3. We reached 3 main conclusions. First, the TusoAI-optimized version substantially outperformed the original scDRS in both simulations and real-data benchmarks: it achieved over 40% higher power in causal simulations (Figure 3A) while retaining calibration in null settings (Appendix R), and identified 21% more true cell type–disease associations (17 vs. 14) without false associations in a real-data benchmark (Li et al., 2025). Second, the TusoAI-optimized scoring function is concise and interpretable. It computes association scores in *log–log* rather than *log* space, likely because this transformation better captures polygenic disease signals across many genes, avoiding domination by a few highly expressed genes. This improvement reflects TusoAI's ability to efficiently explore variations built on the original method: it tested 167 different variations in 24 hours and at a cost of $0.37, whereas the original authors evaluated fewer than 10 versions over 3 months. Third, applying the TusoAI-optimized scDRS to a T cell dataset (Cano-Gamez et al., 2020) revealed 26 disease-associated T cell subpopulations (at FDR<0.05, as per original paper) vs. 17 by the original method, including regulatory T cells, central memory T cells, and effector memory T cells associated with primary biliary cirrhosis, consistent with the roles of these T cell populations in autoimmunity (Dominguez-Villar & Hafler, 2018; Seo et al., 2025).

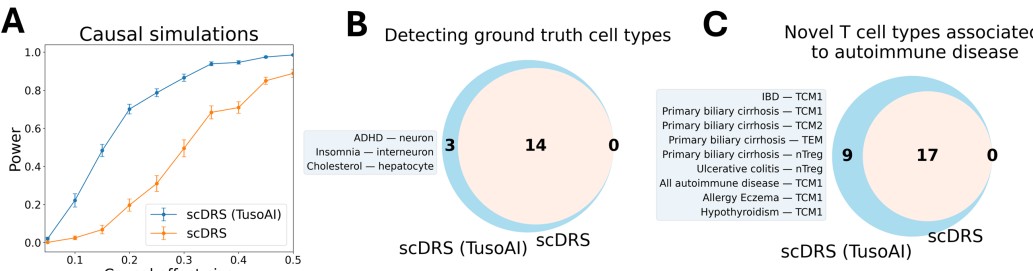

Figure 3: **Optimizing scDRS for detecting cell-disease associations.** **(A)** Assessing power in causal simulations. 95% CI's are calculated across 30 replicates at each perturbation effect size. **(B)** Venn diagram of discovered ground-truth trait-cell type pairs at FDR<0.05 for scDRS and scDRS (TusoAI). New trait-cell type pairs are indicated on the left. **(C)** Venn diagram of discovered trait-T cell subtype pairs at FDR<0.05 for scDRS and scDRS (TusoAI). New trait-cell type pairs are indicated on the left.

**Linking genetic variants to genes using single-cell multiome.**    pgBoost (Dorans et al., 2025) is a state-of-the-art method for linking genetic variants to target genes using single-cell multiome data; it integrates variant–gene distance with multiple linking strategies, but the task remains challenging due to the complexity of genetic regulation (Gazal et al., 2022). Here, we apply TusoAI in conjunction with pgBoost, providing additional positional information for variants and genes, and task it with optimizing distance-based features. Results are reported in Figure 4. We reached 3 main

conclusions. First, the TusoAI-optimized model significantly outperformed the original pgBoost, achieving 13.8% higher enrichment of gold-standard links from fine-mapped eQTLs and 7.2% from activity-by-contact (ABC) links, with particularly large gains across longer variant–gene distances where links are harder to identify (Figure 4A,B). Second, the distance-based features generated by TusoAI are concise and interpretable: 3 are transformed versions of existing features (inverse, squared, and normalized terms), 2 are interactions of gene annotations with distance terms, and the sixth indicates whether the SNP is <50kb from the gene's transcription start site, consistent with literature suggesting the typical enhancer–promoter range of around 70kb (Bower et al., 2025). TusoAI discovered these features by testing 511 combinations of 153 novel distance features within 24 hours at a cost of $0.41, whereas the original authors evaluated 5 features over 1.5 months. Third, applying the TusoAI-optimized pgBoost to fine-mapped SNPs for 94 diseases/traits identified 7 new variant–gene links missed by previous methods (> 95% linking percentile vs. < 95% all others, as per original paper). For example, a fine-mapped variant rs138917529 for glucose and HbA1c was linked to *GCK*, consistent with the roles of Glucokinase in regulating glucose levels related to both glucose metabolism (Froguel et al., 1993) and HbA1c variation (Chakera et al., 2015).

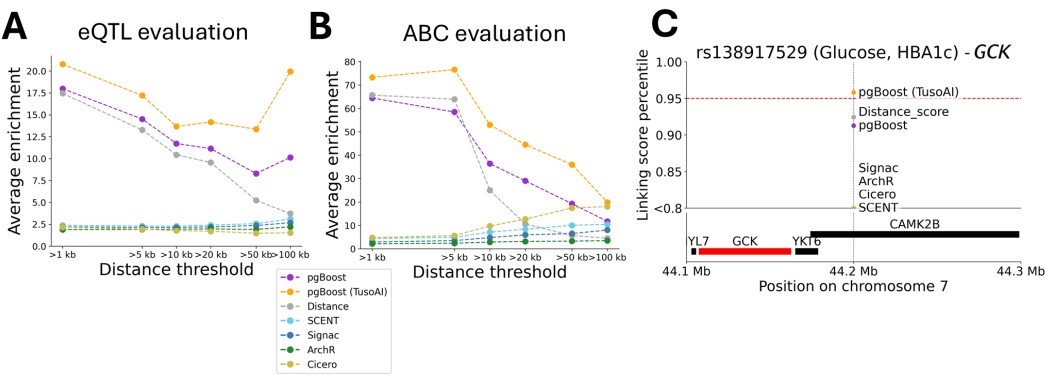

Figure 4: **Optimizing pgBoost for SNP-gene link discovery.** **(A)** Area under the enrichment-recall curve (AUERC, as defined in pgBoost) across distance thresholds for ground truth eQTL variant-gene links. **(B)** AUERC across distance thresholds for ground truth ABC variant-gene links. **(C)** Locus plot of rs138917529 and surrounding genes. Red dashed line indicates linking score percentile cutoff for SNP-gene linking. *GCK* is shaded red, as the gene linked to the focal SNP.

## 6 DISCUSSION

We have presented TusoAI, an agentic system for scientific method optimization. By mimicking a scientist's cycle of method development, TusoAI achieves superior performance on single cell and scientific deep learning benchmarks, and is further able to discover significant optimizations to state of the art methods in genetics which revealed new biology missed by existing methods. We believe TusoAI represents a promising step towards automated scientific method development and optimization, thus accelerating scientific discovery.

We acknowledge several limitations and areas for future work. TusoAI requires a separate validation experiment to base optimization on to prevent overfitting. This experiment should also be quick to run while representing final performance. When optimizing an existing method, TusoAI performs strongest when most of the method is in a single function, and may not perform well if the method is scattered throughout a large codebase. TusoAI does not consider making new evaluation procedures, but rather relies on existing ones and will be vulnerable to the same weaknesses those may have. Several algorithm components are heuristic and may benefit from theoretic justification. We generally focus on biological applications given their complexity and relevance, but achieve reliable performance in diverse scientific domains. Future work may include processing multiple functions in parallel by treating them as separate subtasks, or searching over not just the code-space for methods, but also the data-space for additional useful data to include in a method, as is common in scientific domains.

**Acknowledgment** We thank Richard Border, Ana Prieto, and Boshen Yan for helpful discussions and valuable feedback on the manuscript. This work was supported by a Shurl and Kay Curci Foundation Award and the CMU Center for Machine Learning and Health.

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

# A  ALGORITHM TABLE

| Symbol | Type | Description |
|---|---|---|
| $\mathcal{T}$ | Task description | Short description of task (e.g., single-cell RNA-seq imputation). |
| $\mathcal{D}$ | Dataset | Data used in running the method. |
| $h(\cdot)$ | Evaluator | Scoring function that maps a solution $s$ to a scalar performance score. |
| $s_{\text{init}}$ | Initial solution (optional) | Optional user-provided initial solution to start from. |
| $T_{\text{budget}}$ | Time budget | Maximum wall-clock time allowed for the optimization loop (default: 8 hours). |
| $A_{\text{paper}}$ | Subroutine | "Paper agent": retrieves and summarizes domain-relevant literature given $\mathcal{T}$. |
| $\mathcal{P}$ | Set of documents | Domain knowledge / paper summaries collected by $A_{\text{paper}}(\mathcal{T})$. |
| $A_{\text{cate}}$ | Subroutine | Agent that proposes high-level instruction categories for solving $\mathcal{T}$. |
| $\text{DraftThenRefine}(\cdot)$ | Procedure | Drafting–refinement procedure used to iteratively improve structured text (categories or instructions). |
| $\mathcal{C}$ | Set | Set of instruction categories (e.g., preprocessing, noise modeling, model architecture, etc.). |
| $\{\pi_c\}_{c \in \mathcal{C}}$ | Probabilities | Categorical distribution over categories $\mathcal{C}$, encoding their estimated usefulness. |
| $c$ | Category index | Individual category element from $\mathcal{C}$. |
| $A_{\text{instr}}$ | Subroutine | Agent that drafts and refines concrete instructions for a specific category $c$. |
| $\mathcal{I}_c$ | Set of instructions | Category-specific instructions for optimizing solutions under category $c$. |
| $\mathcal{F}_c$ | Feedback set | Collected feedback specific to category $c$. |
| $\mathcal{I}_{\text{diag}}$ | Instructions | Diagnostic instructions describing how to print informative method information. |
| $\mathcal{F}_{\text{diag}}$ | Feedback set | Feedback collected from diagnostic runs. |
| $A_{\text{init}}$ | Subroutine | Agent that generates initial candidate solutions using $\mathcal{T}$, $\mathcal{P}$, and $s_{\text{init}}$. |
| $\mathcal{S}$ | Solution set | Current pool / archive of candidate solutions explored so far. |
| $N_{\text{top}}$ | Integer | Number of top diverse solutions selected from $\mathcal{S}$ at each iteration. |
| $s$ | Solution | A single candidate solution sampled from the current top set. |
| $\alpha$ | Scalar probability | Probability of using instruction-based optimization instead of diagnostic-based optimization (default: 0.8). |
| $\text{Bernoulli}(\alpha)$ | Distribution | Stochastic decision: with probability $\alpha$ use instruction-based optimization; otherwise use diagnostics. |
| $\text{Cat}(\{\pi_c\}_{c \in \mathcal{C}})$ | Distribution | Categorical distribution over categories $\mathcal{C}$ parameterized by $\{\pi_c\}$. |
| $A_{\text{optim}}$ | Subroutine | Agent that improves a solution $s$ using instructions $\mathcal{I}_c$ and feedback $\mathcal{F}_c$. |
| $s'$ | Solution | New candidate solution obtained by optimizing $s$. |
| $A_{\text{feedback}}$ | Subroutine | Agent that analyzes the change from $s$ to $s'$ and produces textual feedback. |
| $A_{\text{diag}}$ | Subroutine | Diagnostic agent that uses data $\mathcal{D}$ and diagnostic info $(\mathcal{I}_{\text{diag}}, \mathcal{F}_{\text{diag}})$ to improve $s$. |
| $N_{\text{sol}}$ | Integer | Adjusted number of solutions to keep / explore in future rounds (e.g., $N_{\text{sol}} = \max(1, N_{\text{top}} - 1)$). |
| $s^*$ | Solution | Best-found solution at the end of the run, i.e., $s^* \in \arg\max_{s \in \mathcal{S}} h(s)$. |
| wall-clock time | Time | Actual elapsed real-world time since the algorithm started. |

Table 4: Explanation of symbols and subroutines used in Algorithm 1.

# B   EXAMPLE TUSOAI CODE TEMPLATE

**Single-cell denoising template file**

```python
import scanpy as sc
import pandas as pd
import numpy as np
import scipy as sp
import magic
from anndata import read_h5ad
import scprep
from scipy.sparse import csr_matrix
from sklearn.neighbors import NearestNeighbors
from scipy.sparse import issparse
from sklearn.decomposition import PCA
from anndata import AnnData
import random

def mse(adata):
    import anndata
    import scanpy as sc
    import scprep
    import sklearn.metrics

    test_data = anndata.AnnData(X=adata.obsm["test"], obs=adata.obs, var=adata.var)
    denoised_data = anndata.AnnData(
        X=adata.obsm["denoised"], obs=adata.obs, var=adata.var
    )

    # scaling and transformation
    target_sum = 10000

    sc.pp.normalize_total(test_data, target_sum=target_sum)
    sc.pp.log1p(test_data)

    sc.pp.normalize_total(denoised_data, target_sum=target_sum)
    sc.pp.log1p(denoised_data)

    error = sklearn.metrics.mean_squared_error(
        scprep.utils.toarray(test_data.X), denoised_data.X
    )
    return error

def tuso_model(adata):

    adata.obsm["denoised"] = ...
    return adata

def main():
    np.random.seed(42)
    random.seed(42)
    adata = read_h5ad('openproblems_datasets/1k_pbmc_processed.h5ad')
    print("tuso_model_start")
    adata = tuso_model(adata)
    print("tuso_model_end")

    val_metric = 1 - mse(adata)
    print(f"tuso_evaluate: {val_metric}")

main()
```

## C  EXAMPLE INSTRUCTIONS GENERATED BY TUSOAI

**Example categories for single-cell denoising**

- data_preprocessing
- feature_engineering
- model_architecture
- hyperparameter_tuning
- imputation_strategies
- normalization_methods
- evaluation_metrics
- cross_validation
- domain_knowledge_integration
- robustness_techniques
- noise_modeling
- dropout_probability_estimation
- graph_neural_network_optimization
- dropout_pattern_analysis
- pipeline_interaction_analysis
- low_rank_approximation_optimization
- autoencoder_classifier_integration

**Example instructions within a category**

- leveraging graph attention mechanisms to focus on informative cell interactions
- incorporating multi-layer graph convolutions to capture hierarchical gene expression patterns
- implementing edge dropout to enhance model robustness against noise in cell relationships
- utilizing message passing to propagate information across similar cell types effectively
- integrating adaptive learning rates for different graph nodes based on local connectivity
- employing graph pooling techniques to summarize cellular features without losing critical information
- applying graph regularization to maintain structural integrity of the cellular network
- utilizing node embeddings to capture latent features of gene expression profiles
- optimizing neighborhood sizes dynamically based on data density in the graph
- exploring higher-order graph structures to uncover complex relationships in RNA-seq data

## C.1 PREDEFINED DIAGNOSTIC INSTRUCTIONS

> **Example predefined diagnostic instructions**
>
> - altering or adding diagnostic information to be printed
> - altering or adding complex diagnostic information of specific model components
> - printing key statistical assumptions underlying the model (e.g., independence, normality)
> - emitting warnings when model assumptions appear to be violated by the data
> - logging all implicit assumptions made during model selection or preprocessing
> - printing assumptions related to feature distributions or transformations
> - displaying model-specific assumptions such as linearity, homoscedasticity, or no multicollinearity
> - printing assumptions about data completeness, such as missing value tolerance
> - logging expectations about input feature scaling or normalization
> - displaying prior distributions or regularization beliefs embedded in the model
> - printing assumptions about label distribution (e.g., class balance or stratification)
> - emitting diagnostics when data fails to meet i.i.d. assumptions
> - logging assumed causal directions or conditional independencies in the model
> - printing constraints assumed on feature ranges or valid input domains
> - warning if assumptions about sufficient training data volume are not met
> - displaying structural assumptions, such as sparsity or low-rank representations
> - logging assumptions related to stationarity or autocorrelation in time-dependent data

# D  SINGLE-CELL ANALYSIS TASKS SETUP

The OpenProblems benchmark (Luecken et al., 2025) contains 12 single-cell analysis tasks with numerous testing datasets and benchmark metrics for each. We select 6 tasks: single-cell denoising, label projection, batch integration, spatially variable gene identification, spatial decomposition of cell types, and visualization. These were selected with the following criteria. First, we required more than one dataset, such that we can optimize on one dataset, and deploy the learned method on the remaining testing datasets, excluding the 2 cell-cell communication tasks and perturbation prediction. Second, a publicly available Github to ensure we are reproducing the testing procedures correctly, excluding multimodal integration and modality prediction. Third, a method for the task should be able to run in a reasonable amount of time on a CPU, excluding the foundation model benchmark.

In each task, we performed optimization on one dataset which could be run in a reasonable amount of time ($< 2$ minutes for a simple baseline model). The learned methods of each baseline were then applied to the deployment datasets. In selecting benchmark metrics for each task, we had three criteria. First, the metric should not have unavoidable trivial solutions, excluding the Poisson loss metric from denoising, as this can be easily minimized by simply down-weighting lowly expressed genes, including by just scaling genes by their variance or re-normalizing the data. Second, the metrics should be computationally efficient to run, so optimization speed of each method will not be dominated by running metrics. This excluded several metrics from batch integration and visualization. Third, the metric should line up with the task. In the SVG task, it is initially measured in correlation with spatial variability scores, however, the simulation procedure generates binary 0/1 labels of spatial variability, thus we use accuracy of classifying a gene as SVG instead. We also normalize metrics such that each is between 0 and 1 and a higher score is better. The score in denoising is 1-MSE, normalized so that no denoising is 0, and perfect denoising is 1. The score in spatial decomposition is normalized so that a random cell type assignment is 0, and perfect decomposition is 1. See Table 5 for a full breakdown of datasets and metrics used in single-cell tasks.

|  | **Optimization dataset** | **Testing datasets** | **Benchmark metrics** |
|---|---|---|---|
| Denoise | 1K PBMC | 5K PBMC
Pancreatic | MSE |
| Label | 5k cells from Immune Cell Atlas | Diabetic Kidney
GTEX v9
HypoMap
Mouse Pancreatic Islet Atlas
Tabula Sapiens | Accuracy
F1 macro
F1 micro
F1 weighted |
| Batch | 5k cells from Immune Cell Atlas | Diabetic Kidney
GTEX v9
HypoMap
Mouse Pancreatic Islet Atlas
Tabula Sapiens | Graph connectivity
ASW label
ASW batch |
| SVG | Drosophila Stereo-seq E5 | Drosophila Stereo-seq E10
Drosophila Stereo-seq E9
Drosophila Stereo-seq E6 | Accuracy |
| Decomp | TMS Lung (alpha=1.0) | TMS Lung (alpha=0.5)
TMS Lung (alpha=5.0)
Pancreas (alpha=0.5)
Pancreas (alpha=1.0)
Pancreas (alpha=5.0) | $R^2$ |
| Visual | Mouse HSPCT | 5K PBMC
Mouse Myeloid
Zebrafish | Trustworthiness
Distance correlation
Density Preservation |

Table 5: **Single-cell benchmark setup.** Datasets and metrics refer to the setup on the OpenProblems webpage.

# E  DEEP LEARNING TASKS SETUP

The NASBENCH-360 benchmark Tu et al. (2022) contains 10 deep learning tasks across scientific domains with predefined training, validation, and testing splits, as well as evaluation procedures. We select 5 tasks: Spherical, NinaPro, DeepSEA, Satellite, and ECG. These were selected with the following criteria. First, the task should be scientific and somewhat understudied compared to standard ML tasks, excluding the 2 standard image and audio classification tasks. Second, to ensure fair comparison against the precomputed baselines, we removed tasks where we were uncertain about reproducing the evaluation procedure, partly due to recent GitHub or package updates requiring debugging, excluding Cosmic, PSICOV, and DarcyFlow.

In each task, we performed optimization by training a model on the predefined training set and attaining a score on the validation set. The final testing accuracy of optimized models is attained when deploying the model on the predefined test set. We use the same splits and metrics defined in the original paper.

# F    BASELINE IMPLEMENTATIONS

**AIDE.** AIDE takes as input a data folder, task description, and evaluation metric. While originally designed for whole-workflow construction in ML tasks, this can be adapted to general optimization in the following ways. First, AIDE can operate on any data input in the data folder. If specific preprocessing information was needed, we could input this code to the task description. Second, in place of specifying an accuracy metric (e.g., "F1 score"), we instead simply input the entire evaluation function in Python, and found this worked well. As our goal is optimization and not construction, AIDE's initial prompt is tuned until code was consistently generated and optimized upon, typically requiring the same formatting information as other methods. AIDE is run for the same length as TusoAI (8 hours) in the same conda environment on the same CPU (Optimization for AIDE and TusoAI is performed on the same Intel(R) Xeon(R) Gold 5416S.) or GPU (Intel(R) Xeon(R) Silver 4314 CPU @ 2.40GHz), given 4 threads and 50GB of memory. While AIDE does have a default timeout per execution of 1 hour, on attempting to set this to the same time as TusoAI led to consistent crashes on more of half of tasks, thus we left it as is.

**Biomni.** We access Biomni through its web page. Biomni runs on a CPU and can take input files up to a limit, has a runtime execution of 1 hour per iteration. While not specifically designed for optimization, we can upload the same template code and data as TusoAI then ask Biomni to perform an iterative process of updates. In practice, this led to between 2 and 10 iterations per task between 20 minutes and 4 hours. This procedure was signed off by the original authors of Biomni.

**ChatGPT-Agent.** We access ChatGPT-Agent through its web page. ChatGPT-Agent can accept input files up to 25MB and has a runtime execution of 1 hour per iteration. We upload the same template code and data as TusoAI and ask ChatGPT-Agent to perform an iterative process of updates. In practice, this led to between 2 and 7 iterations per task between 10 minutes and 5 hours.

**Expert.** The expert baselines for NASBench-360 are pre-computed from their paper. The original authors found the best performing expert models from the literature for each task. This includes the following methods:

1. DeepSea - The original DeepSea model released alongside the dataset, a 1D convolution model with state of the art performance. (Zhou & Troyanskaya, 2015)

2. NinaPro - Feed-forward neural network with attention modules in place of convolutions. (Josephs et al., 2020)

3. Spherical - a spherical CNN with special operations for spherical signals. This model achieved state of the art performance on spherical MNIST. (Cohen et al., 2018)

4. Satellite - A linear classifier with convolution kernel as feature extractor, achieving state of the art on UCR time series prediction tasks. (Dempster et al., 2020)

5. ECG - ResNet with 1D convolution, achieving state of the art on several time series prediction tasks for medicine. (Hong et al., 2020)

For single-cell tasks, we selected the expert method with the following criteria. First, it should be within the top 3 methods as defined by the existing OpenProblems benchmarking. Second, the OpenProblems Github should have code for reproducing this method. Third, we selected the method that was particularly efficient compared to others, if applicable, defined by a runtime of less than 10 minutes on OpenProblems, with others having greater than 1 hour. This left us with the following expert methods, whose code we extracted from the OpenProblems Github:

1. Denoise – MAGIC, a graph-based diffusion method that imputes missing gene expression values. (Van Dijk et al., 2018)

2. Batch – ComBat, an empirical Bayes approach that removes batch effects across samples. (Zhang et al., 2020)

3. Label – PCA + Logistic Regression, which uses low-dimensional PCs as features for efficient cell-type classification. (Luecken et al., 2025)

4. Decomposition – NNLS, a non-negative least squares model for estimating gene programs or latent factors. (Aliee & Theis, 2021)

5. SVG – SPARK-X, a spatial variance component model that identifies spatially variable genes at scale. (Zhu et al., 2021)

6. Visualize – t-SNE (log10CP10K), a nonlinear embedding of log-transformed counts for 2D visualization. (Luecken et al., 2025)

**Claude-4-Sonnet and GPT-5.** In deep learning tasks where Biomni and ChatGPT-Agent cannot apply due to computational limitations (file size, runtime, GPU access), we substitute the best of 10 models generated by Claude-4-Sonnet and GPT-5. 10 models are generated by prompting these LLMs using the same template that would have been used in Biomni and ChatGPT-Agent. The best is decided by the top performing model on the validation dataset which ran in less than one hour, akin to the runtime limitations of Biomni and ChatGPT-Agent.

# G    EXAMPLE TASK INFORMATION

|  | Denoise |
|---|---|
| **Task Description** | single cell RNA-seq imputation |
| **Drafted Categories** | data_normalization
feature_selection
imputation_modeling
latent_space_representation
noise_handling
batch_effect_correction
hyperparameter_tuning
evaluation_metrics
ensemble_imputation_methods
domain_specific_constraints
count_distribution_modeling |
| **Refined Categories** | data_normalization
feature_selection
imputation_modeling
latent_space_representation
noise_handling
batch_effect_correction
hyperparameter_tuning
evaluation_metrics
ensemble_imputation_methods
domain_specific_constraints
count_distribution_modeling
dropout_probability_estimation
graph_based_representation
dropout_pattern_analysis
pipeline_interaction_analysis
rank_estimation_and_optimization
virtual_class_label_generation |
| **Drafted Solution Descriptions** | k-nearest neighbors imputation
matrix factorization (e.g., PCA, NMF)
autoencoder-based imputation (including DCA)
generative adversarial networks (GANs) for imputation
deep learning models (e.g., U-Net architecture) |
| **Refined Solution Descriptions** | k-nearest neighbors imputation
matrix factorization (e.g., PCA, NMF)
autoencoder-based imputation (including DCA)
deep learning models (e.g., U-Net architecture)
graph neural network (GNN) for imputation
scImpute for dropout imputation
co-occurrence clustering based on dropout patterns
scran normalization with prior clustering
AutoClass model for scRNA-Seq cleaning
low-rank matrix approximation (ALRA) |

# H    NOVELTY OF DISCOVERED METHODS

For the four example methods constructed by TusoAI listed in Section 4.1, we expand on the novelty claim with a thorough literature review of related methods.

In single-cell denoise, TusoAI designed a non-negative matrix factorization (NMF) approach that models dropout rates, Poisson noise, and performs iterative refinement, distinct from the only other NMF-based approach in the OpenProblems benchmark, ALRA (Linderman et al., 2022). Specifically, ALRA applies a low-rank approximation to a globally normalized matrix and performs an adaptive thresholding step to restore zeros, but it does not explicitly model count noise, does not incorporate dropout mechanisms, and does not iteratively refine factors. Another denoising method uses NMF, DenoiseIt (Jeon et al., 2024). This method focuses on identifying noisy features via NMF loadings combined with isolation-forest filtering, and does not perform probabilistic modeling of gene-cell counts or imputation of dropout-affected expression values, again distinct from our approach. Another point of novelty is that TusoAI's learned method is outperforming MAGIC (Van Dijk et al., 2018), which was found to outperform ALRA (Luecken et al., 2025).

In SVG, TusoAI adapted known techniques such as modeling expression as a function of spatial coordinates and neighborhood summaries to create a custom, high-performing method. Many existing SVG detectors are primarily coordinate-based, using Gaussian-process or generalized linear models over spatial locations (SpatialDE, SpatialDE2, SPARK-X, GPcounts, BOOST-GP) (Svensson et al., 2018; Kats et al., 2021; Zhu et al., 2021; BinTayyash et al., 2021; Li et al., 2021), while others rely mainly on neighborhood or graph structure, diffusion, or spatial autocorrelation statistics (Moran's I, SOMDE, scGCO, Sepal, SpaGCN, SpaGFT, nnSVG, Spanve) (Luecken et al., 2025; Hao et al., 2021; Zhang et al., 2022a; Andersson & Lundeberg, 2021; Hu et al., 2021; Chang et al., 2024; Weber et al., 2023; Cai et al., 2023). Graph-based models such as SpaGCN and SpaGFT can incorporate both spatial coordinates and local neighborhoods through graph constructions and convolution or spectral transforms (Hu et al., 2021; Chang et al., 2024), but they do not explicitly combine smooth coordinate regression with fixed neighborhood summary covariates in a single per-gene predictive model as TusoAI does. This joint modeling of coordinate trends and neighborhood summaries enables TusoAI to outperform SPARK-X (Zhu et al., 2021) on our SVG benchmark, despite SPARK-X being among the strongest existing SVG baselines (Luecken et al., 2025).

In Satellite, TusoAI combined preprocessing, training procedures, loss functions, and ensembling techniques to build the top-performing model. First, this is distinct from the expert model, which is a linear classifier (Dempster et al., 2020). Second, these kinds of pipeline-level decisions lie outside the search space of the NAS baselines used in NASBench-360: methods such as DARTS-GAEA, DenseNAS, AMBER, and tuned WRN search only over convolutional architectures under a fixed data preprocessing pipeline, standard loss, and a single-model training recipe, and therefore cannot automatically implement the techniques TusoAI does here.

In Spherical, TusoAI fine-tuned layers of ResNet-50 and augmented the data with random flips and rotations. First, this is distinct from the expert model, which is a spherical CNN (Cohen et al., 2018). Again, these decisions, such as fine-tuning Resnet-50 or augmenting data with rotations and flips lie outside the search space for NAS methods.

# I   RUNTIME OF NEW METHODS

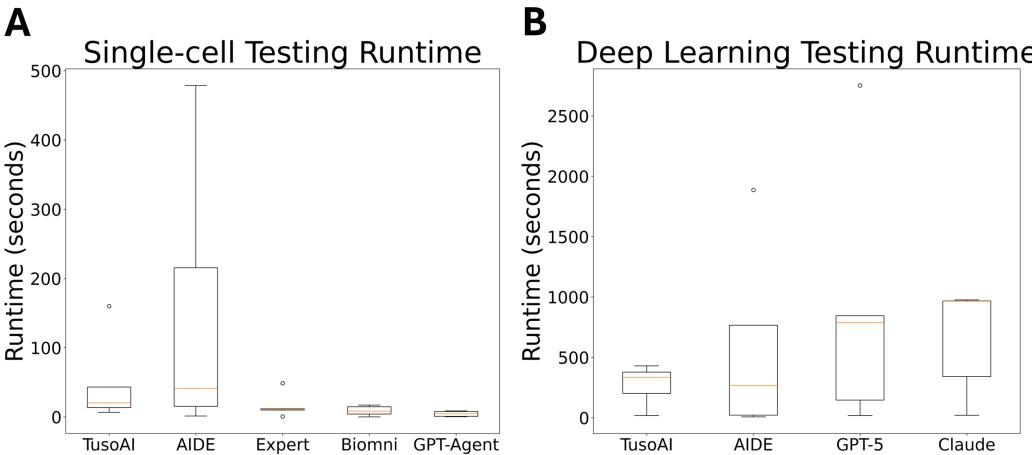

Figure 5: Testing set runtime for **(A)** single-cell tasks (averaged over 3 random data splits) and **(B)** deep learning tasks (averaged over 3 random seeds).

## J  OPTIMIZATION TRAJECTORIES

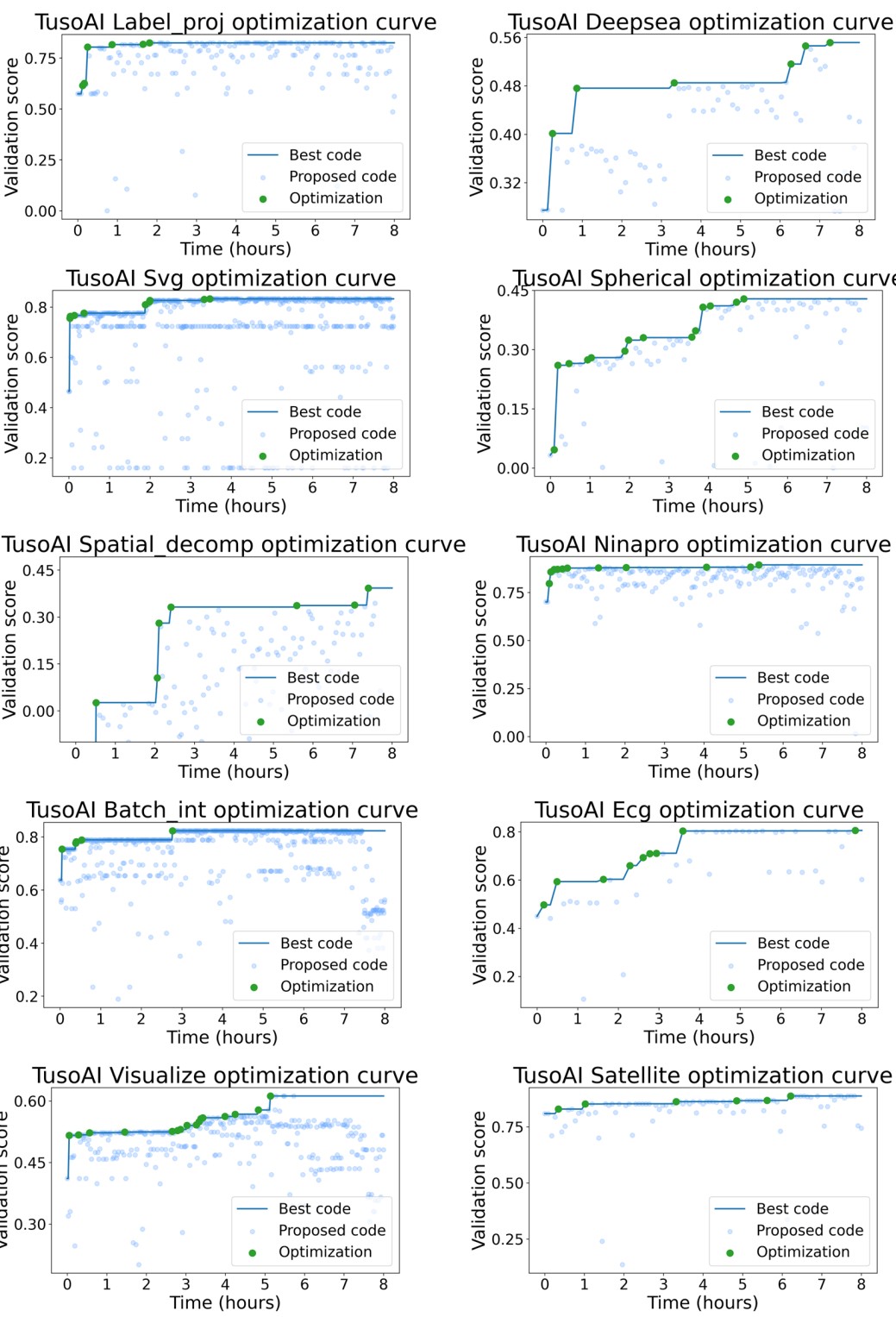

Figure 6: TusoAI's optimization trajectory for all benchmarking tasks. Denoise is in Figure 2.

# K  AIDE OPTIMIZATION TRAJECTORIES

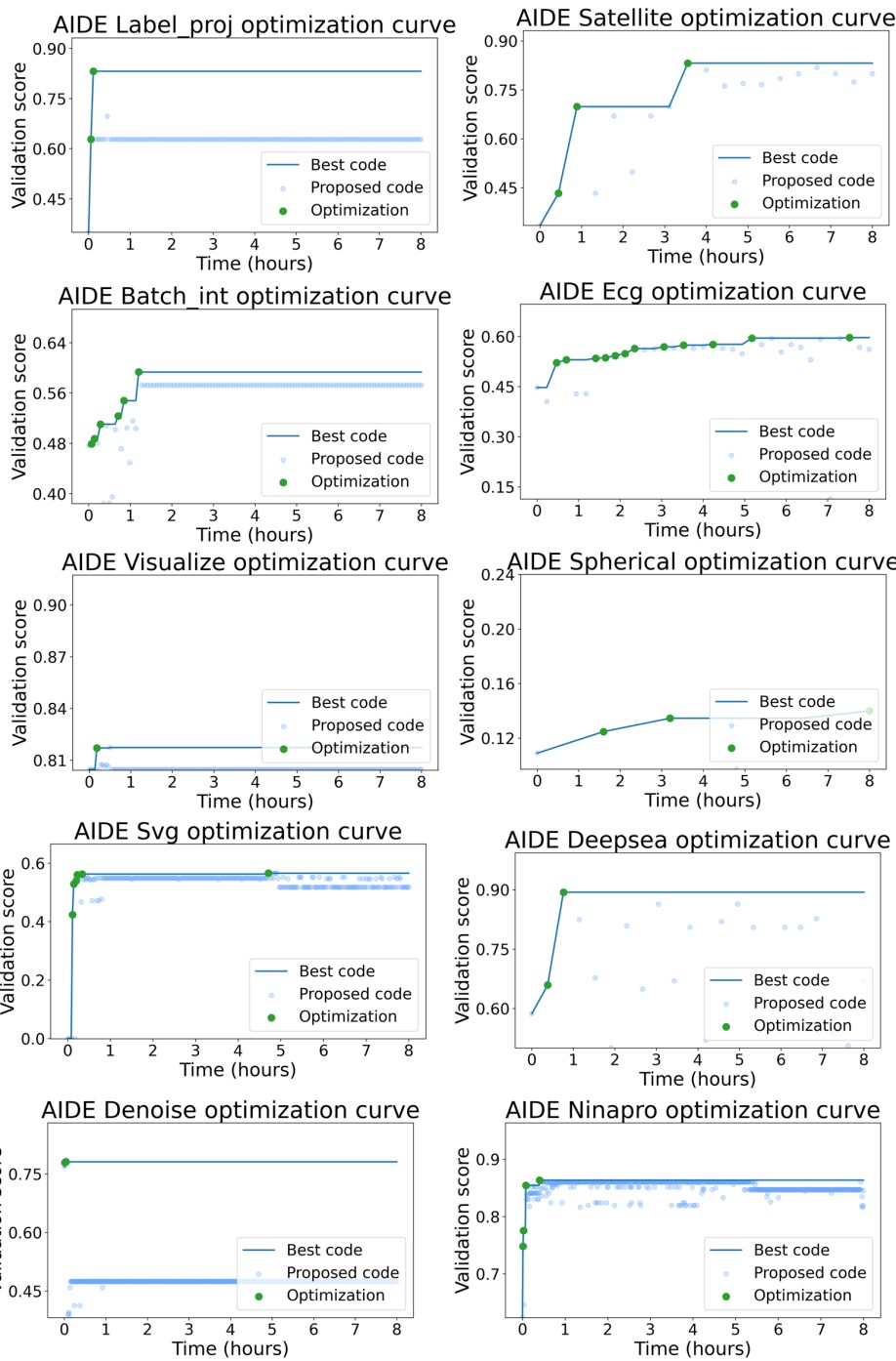

Figure 7: AIDE's optimization trajectory for all benchmarking tasks. AIDE can edit the evaluation function, which occurred in the decomposition task which was thus excluded.

# L    STABILITY ACROSS REPLICATES

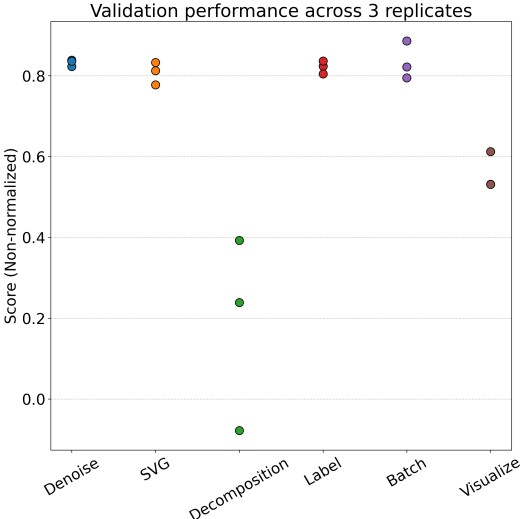

Figure 8: **Validation performance across 3 replicates.** Final validation performance after running TusoAI 3 separate times on each single-cell task.

# M  CODE DIVERSITY

We measure the diversity of generated code, as measured by the cosine similarity of the text embedding of one generated code versus all others, similar to (Aygün et al., 2025). This is performed for TusoAI and AIDE. For each, we first filter out repetitive/uninformative code strings, including comments, imports, evaluation functions and data loading procedures (which will not change over iterations). We then apply sklearn's TfidfVectorizer function to each cleaned code to obtain a text embedding. We can then compute the cosine similarity between pairs of code. Diversity is measured as 1-cosine similarity. We opt for TF-IDF instead of more sophisticated methods like CodeBERT (Feng et al., 2020) which measure semantic similarity, as we observed this overestimated the similarity between code (all cosine similarity $> 0.997$ for all tasks). This is likely due to each iteration always being a slight permutation of the same python method performing the same task. TF-IDF better captures a measure of difference between algorithmic procedures in this case.

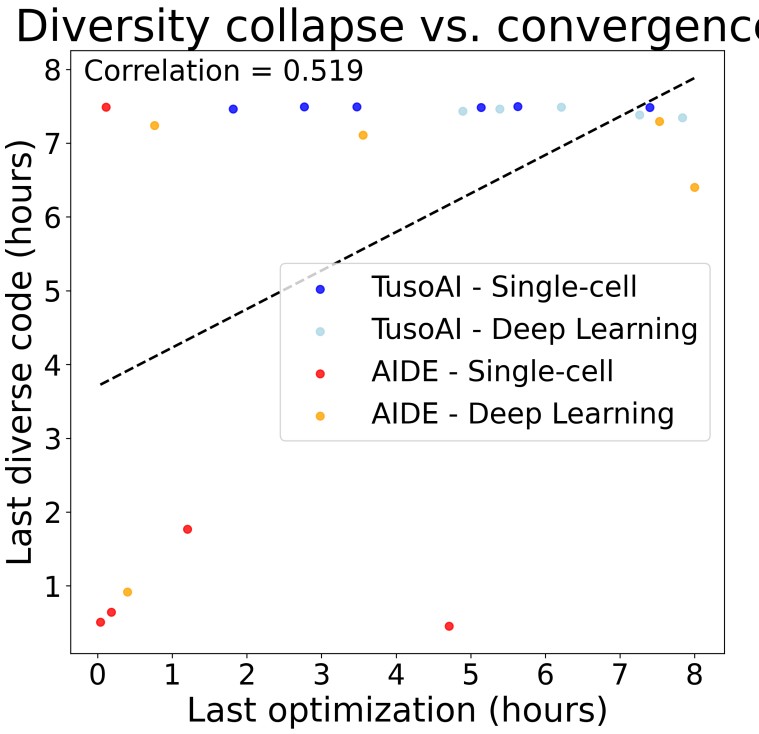

Figure 9: **Code diversity versus optimization ability.** Last diverse code for each trajectory (defined as last position with diversity>0.1) versus last optimization position.

# N ABLATION ANALYSIS

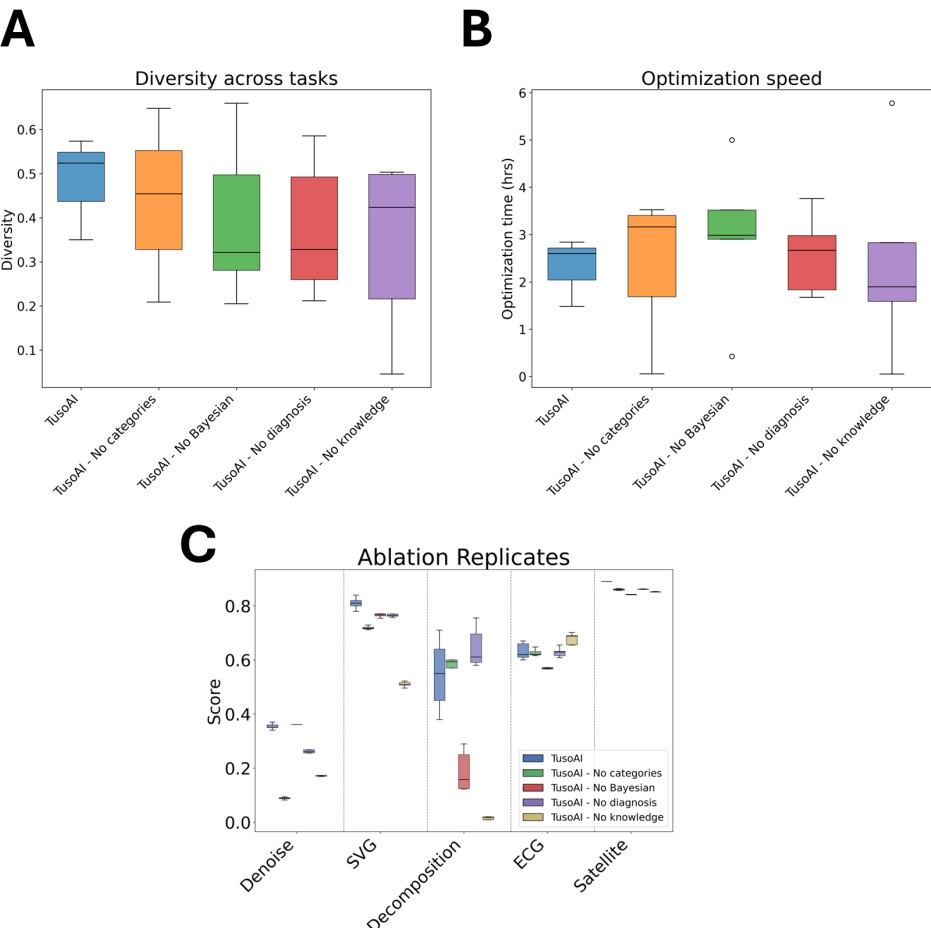

Figure 10: **Additional ablation information.** (A) Box plot across 5 tasks of the mean code diversity. (B) Box plot across 5 tasks of the mean time to optimize. (A) Box plot of the final testing scores of 5 replicates for each ablation.

## O  VARYING LITERATURE SEARCHED

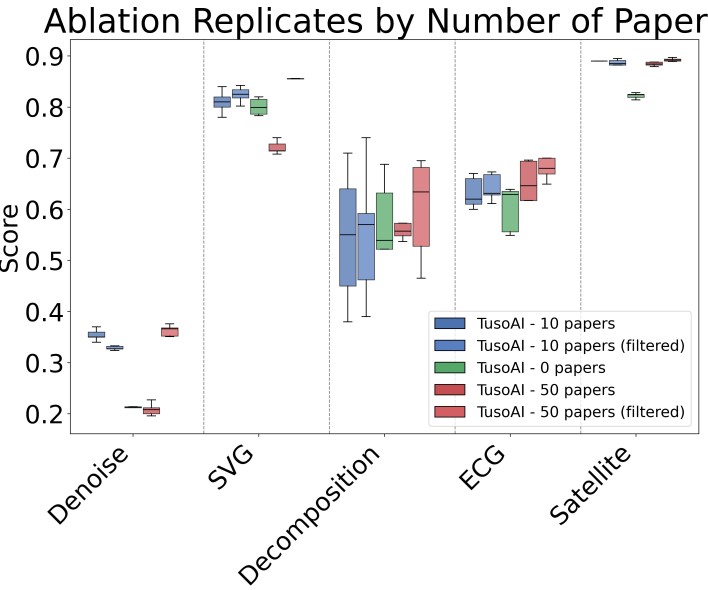

Figure 11: Box plot of the final testing scores of 5 replicates for collecting 10 papers (default), 0 papers, and 50 papers, alongside optional paper summary filtering step.

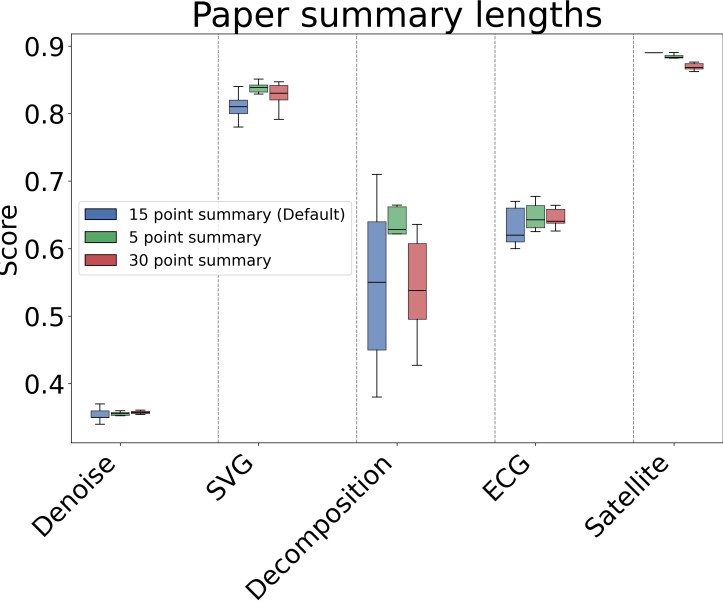

Figure 12: Box plot of the final testing scores of 5 replicates for having 15 bullet point summaries (default), 5 bullet point summaries, and 30 bullet point summaries.

# P    LLM ANALYSIS

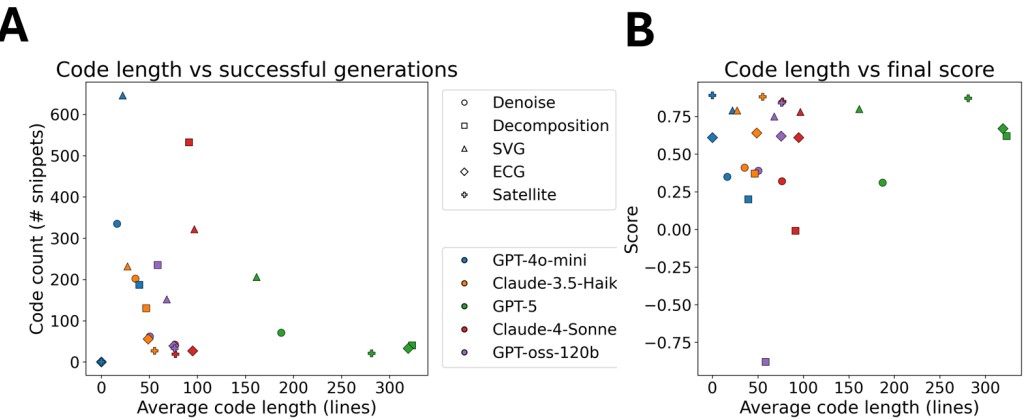

Figure 13: **Additional LLM information.** (A) Average length of generated methods for each task and LLM versus the total count of how many methods were generated. (B) Average length of generated methods for each task and LLM versus the final deployment performance.

# Q COST ANALYSIS

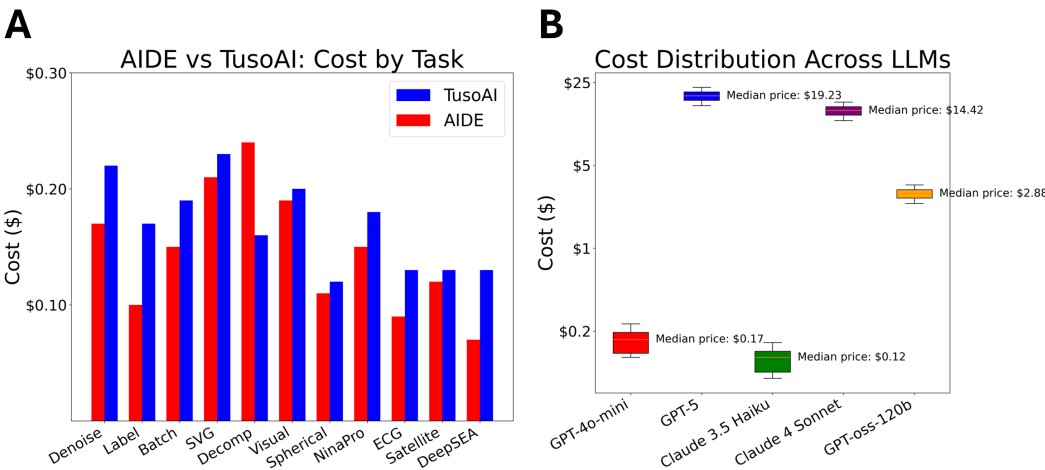

Figure 14: **Cost distribution.** (A) TusoAI vs. AIDE cost for each task. (B) Boxplot of costs for each LLM on 5 ablation tasks.

# R  scDRS ANALYSIS

**Optimization setup.** scDRS' codebase consists of several files. We construct a version of compute_score.py that exposes the compute_raw_score function. This is the only function TusoAI operates upon during optimization. For optimization, we construct causal simulations similar to the scDRS paper, subsampling 10k cells from TMS, perturbing 1000 disease genes in a cluster of cells, setting the geneset overlap to 25%, and varying effect size from 5 to 50%. TusoAI optimizes the compute_raw_score function based on the average (F1 + AUPRC)/2 across 3 replicates at effect size 15%, where scDRS has lower power. We run this experiment for 24 hours using default parameter settings for TusoAI.

**Additional simulation results.** We apply scDRS and the learned version by TusoAI to all 30 replicates of each effect size in causal simulations. We additionally apply it to 100 replicates of null simulations, identical to scDRS, where 1000 random genes are selected with no perturbation. Additional metrics in these simulations are reported in Figure 15.

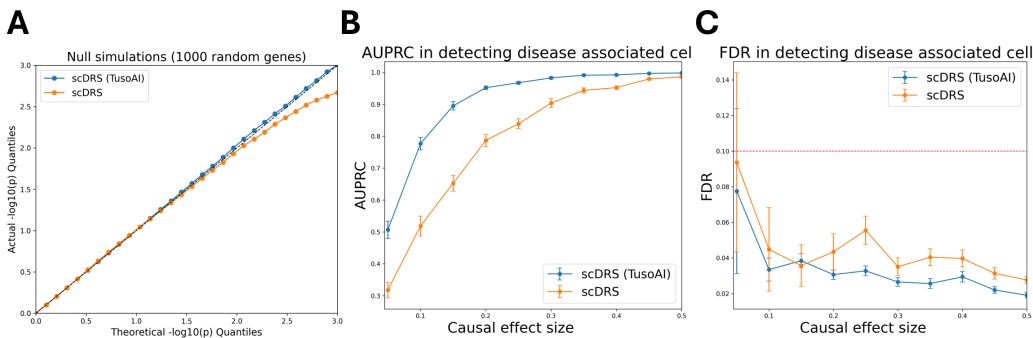

Figure 15: **Additional scDRS metrics.** (A) Q-Q plot of -log10 p-values in null simulations. 95% CI's are calculated at each point across 30 replicates. (B) AUPRC of associating individual cells in causal simulations. 95% CI's are calculated at each point across 30 replicates. (C) FDR of associating individual cells in causal simulations. 95% CI's are calculated at each point across 30 replicates.

# S PGBOOST ANALYSIS

**Optimization setup.** pgBoost discovers that distance-based features are critical for modeling SNP-gene distances. It's samples are SNP-gene pairs, and features include over 30 features derived from single-cell multiome methods and 2 distance features, the SNP distance to the gene's transcription start site (TSS), and a binary indicator of if this is the closest TSS of any gene to the SNP. We augment pgBoost's script with gene annotations from GENCODE V48 (Mudge et al., 2025), specifically the SNP's position, the gene's TSS, and the gene's transcription end site (TES). During both the knowledge tree construction and optimization process, TusoAI is encouraged to come up with instructions/optimizations relevant to distance-based modeling of SNP-gene links and avoid other model changes. Optimization is performed by increasing the average enrichment in pgBoost's primary evaluation of gold-standard links (eQTL and ABC) relative to the original pgBoost's enrichment. We run TusoAI for 24 hours using default parameter settings.

**Real data analysis.** We analyze fine-mapped SNPs from the same set of GWAS traits as in the pgBoost paper. pgBoost considers a true link to be in the top 95% percentile, and specifically looks for SNP-gene links that are not in such a percentile for other methods. We perform an identical analysis, looking for links in the top 95% of pgBoost (TusoAI), but not in other methods.

## T WARM START DEEP LEARNING

We show how TusoAI's warm-start capability might work in conjunction with a scientific deep learning task. For NinaPro, we re-implement the Expert model from NASBench-360. This is a feed-forward neural network with attention modules in place of optimizations. Optimizing this model for 8 hours with TusoAI improves testing performance from 0.91 to 0.94, now becoming the top performing model for this task (Figure 16). Where both the Expert and cold-start TusoAI's learned models were outperformed by NAS methods, their combination yields a new top model. We next analyze the optimization trajectory of TusoAI to see how this was achieved. We summarize the key optimizations below:

1. The dense attention network was replaced with a dilated temporal convolutional network (TCN).

2. Five separate TCN models were trained and ensembled.

3. Features were standardized with z-scaling.

4. Gradient clipping.

5. Log transformation of input features.

6. Ensemble is replaced by a mixture of experts (MOE) architecture.

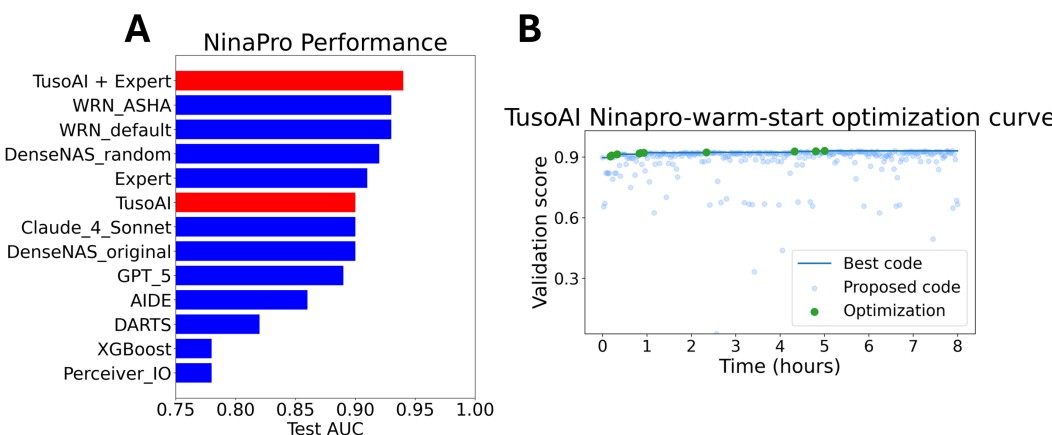

Figure 16: **Warm start example on NinaPro.** **(A)** Testing performance of all NinaPro methods. **(B)** Optimization trajectory of TusoAI with warm-start NinaPro.

## U    PROMPT TEMPLATES

### U.1    INSTRUCTIONS FOR PARSING LITERATURE

---

**Initializing paper description with abstract**

You are a scientific summariser. Draft a concise yet technically accurate description of the paper's method based **only** on the abstract below, to the extent possible. Capture the main points using bullet points. Avoid complete sentences and omit details irrelevant to the methods.
Abstract: """[ABSTRACT GOES HERE]"""

---

**Updating paper description with methods section**

The current method description:
"""[CURRENT DESCRIPTION GOES HERE]"""
New excerpt from the paper:
"""[NEW TEXT GOES HERE]"""
Update the description by **incorporating any new technical details or correcting existing ones** found in the excerpt. Keep the description concise and clear. Return **only** the revised description. Use bullet points. Avoid full sentences and exclude information unrelated to the methods. Do not exceed 15 bullet points.

---

## U.2 INSTRUCTIONS FOR CONSTRUCTING CATEGORIES

---

**Initializing categories with LLM**

We are building an LLM-powered AutoML system for the task:
"[TASK DESCRIPTION]"
As a reference, some generic categories for optimizing classification models include:
[CLASSIFICATION CATEGORIES]
You are a master of machine learning and the domain relevant to this task. First briefly reason about what kinds of modeling interventions or optimization strategies could be helpful for this specific task. Then propose a list of concise, task-relevant optimization categories.
Your list should include conceptual ideas tailored to this task and each should represent a specific axis of improvement (e.g., architectural choices, preprocessing strategies, domain constraints, evaluation metrics, robustness techniques, etc.).
Output exactly **N** proposed categories, one per line, each enclosed in:
<c>Category Name</c>
Do not include any other text, explanation, or formatting. By "optimization" we mean strictly performance improvements — not runtime, scalability, visualization, logging, post-evaluation tools, or similar considerations. You will only have access to: [DATA AVAILABLE].

---

**Updating categories with papers**

We are building an LLM-powered AutoML system for the task:
"[TASK DESCRIPTION]"
We will curate and refine our categories based on the current categories and a paper.
Current categories: [CURRENT CATEGORIES]
Paper: "[TITLE]" Key method points: [BULLET POINTS]
TASK 1. If the paper suggests a **new** axis of optimization missing from the list, propose a concise, task-relevant category for it. 2. If two or more current categories can be merged, provide a single category name that subsumes them. 3. Otherwise, if a category is irrelevant given only [DATA AVAILABLE], leave the list unchanged.
Return **one updated list only**, one category per line. Each line must be wrapped exactly like:
<c>Category</c>
No other text.

---

## U.3 INSTRUCTIONS FOR CONSTRUCTING WITHIN-CATEGORY INSTRUCTIONS

---

**Initializing within-category instructions with LLM**

We are designing an LLM-powered AutoML system for the task:
"[TASK DESCRIPTION]"
Current optimisation axis: **[CATEGORY]**
Below is a style example of prompts for a *regularisation* category for a classification task.
Each prompt begins with *by ...* and expresses a specific, actionable optimisation idea:
[FEW-SHOT EXAMPLES]
You are a master of machine learning and the domain relevant to this task. Keeping the same concise, actionable style, write **exactly N distinct prompts** that belong to the **[CATEGORY]** category **and are appropriate for this task**.
These should include a mix of general, conceptual, and complex prompts, not overly specific, similar to the examples.
Wrap *each* prompt in its own:
<p> ... </p>
Return **only** these <p>...</p> lines, nothing else.
By optimisation we mean strictly **performance**, not runtime, scalability, logging, visualization, evaluation, or post-processing. Assume evaluation metrics already exist. You will only have access to: [DATA AVAILABLE].

---

**Refining within-category instructions with LLM**

We are designing an LLM-powered AutoML system for the task:
"[TASK DESCRIPTION]"
We will only have access to: [DATA AVAILABLE].
Here is a concise summary of the baseline method: """[SUMMARY]"""
Below are style examples of valid prompt lines taken from earlier work: [FEW-SHOT EXAMPLES]
Your job is to generate **between N_min and N_max new prompts**. These prompts will ultimately be assigned to one of the following categories:
[CATEGORY LIST]
**Step 1**: Generate the prompts, independently of categories. **Step 2**: Assign each prompt to its most relevant category.
For each prompt, output a line in this exact format:
<c>CategoryName</c><p>by ...</p>
Rules:
* Every prompt must begin with **"by ..."** * Cover a mix of general, conceptual, and complex ideas * Focus strictly on **performance optimisation** (ignore runtime, scalability, logging, visualization, etc.) * Return **only** the <c>...</c><p>...</p> lines — nothing else.

---

## U.4 INSTRUCTIONS FOR CONSTRUCTING INITIAL SOLUTIONS

---

**Initializing solutions with LLM**

We are designing an LLM-powered AutoML system for the task:
"[TASK DESCRIPTION]"
Below is an example list of generic model initializations for a **classification** task: [FEW-SHOT EXAMPLES]
You are a master of machine learning and of the domain relevant to this task. Propose **exactly N concise model initializations** that could serve as starting baselines **for this specific task**, given that we only have [DATA AVAILABLE].
These should be general task-specific methods, model families, or high-level architectural descriptions — not fully specified pipelines.
Output one per line, each wrapped in:
<m> ... </m>
Return **only** these <m>...</m> lines — no explanations, no extra text.

---

**Refining initial solutions with LLM**

We are building an LLM-powered AutoML system for the task:
"[TASK DESCRIPTION]"
We will curate and refine our **model initializations** list using insights from the following paper.
Current initializations: [CURRENT INITIALIZATIONS]
Paper: "[TITLE]" Key method points: [BULLET POINTS]
TASK → 1. If the paper presents a **model family or architecture** not covered above, propose it as a concise initialization ($\leq$ 6 words). 2. If two or more current initializations are effectively the same family, merge them by giving a single, clear name that subsumes them. 3. If neither condition applies, or if the model cannot be implemented using **[DATA AVAILABLE]**, leave the list unchanged.
Return **one updated list only** — one initialization per line, each wrapped exactly like:
<m>Initialization</m>
No other text.

---

## U.5 Prompt for developing initial solutions

**Initialisation prompt**

Write a basic version of this model for {task_description} using {init}. Hints: - {hints} {base_fn_code} Output only python code, and do not include comments.

## U.6 PROMPT FOR OPTIMIZING WITH INSTRUCTIONS

---

**Instruction-based optimization prompt**

Write a basic version of this model for {task_description} using {init}. Hints: - {hints} {base_fn_code} Output only python code, and do not include comments.
by choosing one of the following strategies to guide optimisation, based on your assessment of what will most improve this model for {task_description}: {prompt_options}
Additionally, consider the following feedback from earlier attempts that used this same optimisation strategy: {fb_block}

---

## U.7   PROMPT FOR GENERATING DIAGNOSTICS

**Generating diagnostic info prompt**

Write a basic version of this model for {task_description} using {init}. Hints: - {hints} {base_fn_code} Output only python code, and do not include comments.
by choosing one of the following strategies to print diagnostic information, based on your assessment of what will be most informative for optimisation of this model for {task_description}. Ensure the information printed is concise enough to be used in an LLM prompt: {diagnostic_options}

## U.8   PROMPT FOR OPTIMIZING WITH DIAGNOSTICS

**Optimizing with diagnostic info prompt**

Write a basic version of this model for {task_description} using {init}. Hints: - {hints} {base_fn_code} Output only python code, and do not include comments.
by assessing this diagnostic info and proposing model/feature improvements for this model for {task_description}: {current_code}
Additionally, consider the following feedback from earlier attempts that used this same optimisation strategy: {fb_block}

## U.9 PROMPT FOR GENERATING FEEDBACK

**Feedback on code optimization**

We attempted to optimize this function: [ORIGINAL_CODE] Here is the proposed optimization: [NEW_CODE] Write a concise one line summary of the differences between the original function and the proposed optimization. It should be as short as possible while summarizing the differences.

## U.10 PROMPT FOR DEBUGGING

**Fix Function Prompt**

Fix this function: {suggestion}. Here's the error: {error_msg} Ignore warnings. If an error is related to installation, assume the package is not installed and try doing it without that specific package. Output only python code, and do not include comments.

## U.11 PROMPT TEMPLATE FOR BIOMNI AND CHATGPT-AGENT

**Single-cell denoising prompt template for scientific agents**

```
We are considering the task of single cell RNA-seq imputation.
We wish to create an expertly optimized model for this.
Here is a starter script. Create a top-performing model for our task within the
    tuso_model function.

import scanpy as sc
import pandas as pd
import numpy as np
import scipy as sp
import magic
from anndata import read_h5ad
import scprep
from scipy.sparse import csr_matrix
from sklearn.neighbors import NearestNeighbors
from scipy.sparse import issparse
from sklearn.decomposition import PCA
from anndata import AnnData
import random

def mse(adata):
    import anndata
    import scanpy as sc
    import scprep
    import sklearn.metrics

    test_data = anndata.AnnData(X=adata.obsm["test"], obs=adata.obs, var=adata.var)
    denoised_data = anndata.AnnData(
        X=adata.obsm["denoised"], obs=adata.obs, var=adata.var
    )

    # scaling and transformation
    target_sum = 10000

    sc.pp.normalize_total(test_data, target_sum=target_sum)
    sc.pp.log1p(test_data)

    sc.pp.normalize_total(denoised_data, target_sum=target_sum)
    sc.pp.log1p(denoised_data)

    error = sklearn.metrics.mean_squared_error(
        scprep.utils.toarray(test_data.X), denoised_data.X
    )
    return error
def tuso_model(adata):
    a = AnnData(
        X=adata.obsm["train"].copy(),
        obs=adata.obs.copy(),
        var=adata.var.copy()
    )

    out = a.X
    out = out.toarray() if issparse(out) else out
    adata.obsm["denoised"] = out
    return adata
def main():
    np.random.seed(42)
    random.seed(42)
    adata = read_h5ad('1k_pbmc_processed.h5ad')
    print("tuso_model_start")
    adata = tuso_model(adata)
    print("tuso_model_end")

    val_metric = 1-mse(adata)
    print(f"tuso_evaluate: {val_metric}")

main()
```

```
Make sure to store the denoised data in adata.obsm["denoised"].
Keep the function header, input, output the same.
```

```
Each time you generate code, run it, extract the tuso_evaluate metric, and try and
    build a better performing solution from the previous solutions.
```

## V GENERIC CLASSIFICATION EXAMPLE

| List | Contents |
|------|----------|
| Categories | ['regularisation', 'feature_engineering', 'hyperparameter_tuning', 'sampling', 'ensemble_methods', 'calibration', 'feature_selection'] |
| Initializations | ["logistic regression", "XGBoost", "random forest", "MLP classifier"] |
| Regularisation instructions | 1. by introducing L1 sparsity constraints to prune features

2. by subsampling training rows each iteration to inject stochasticity

3. by shrinking updates with a smaller learning rate for smoother convergence

4. by refining regularisation strategies

5. by combining complementary regularisation methods

6. by adapting regularisation strength across epochs

7. by scaling regularisation to the dataset size

8. by combining elastic-net with adaptive polynomial penalties to capture curved relationships

9. by adding Jacobian norm regularisation to control sharp non-linear gradients

10. by introducing spectral norm constraints for stable non-linear layers |

