# OpenReview forum: "TusoAI: Agentic Optimization for Scientific Methods"
_ICLR.cc/2026/Conference — ICLR 2026 Poster_

### Official Review · Reviewer_x86D · 2025-10-17

**Soundness:** 2
**Presentation:** 2
**Contribution:** 2
**Rating:** 2
**Confidence:** 4

**Summary:**

## Summary
- The paper introduces TusoAI, an LLM agent for developing computational methods for domain-specific scientific tasks. They apply it to several biological and deep learning problems and demonstrate improvement over other agentic and LLM models. They also demonstrate its performance on iterating on existing state-of-the-art methods and show new scientific insights.

## Recommendation
- Reject. While results comparing with other agentic models are there, the actual performance isn't that much better compared to existing models on the specific tasks. The most interesting result is the case studies, but the other models were not run on the same case studies. There is a lack of case studies in other domains. Code is poorly written and not amenable to open source.

**Strengths:**

## Strengths
- Model architecture is properly motivated and described in sufficient detail.
- The utility of a model is conveyed properly, I can see the usefulness of this.
- Code is provided.

**Weaknesses:**

## Weaknesses
- TusoAI is designed to work with any domain as long as domain knowledge is provided, however, there is a strong focus on biological applications. This paper would benefit from showing TusoAI's performance in other scientific domains similar to the two biological case studies. I see that it worked on deep learning but only a cold start was performed. I am specifically asking for a warm start case study in a non-biology domain.
- Diversity in methods produced is a core architecture consideration, but it is not substantiated in the text. While evidence is provided that TusoAI writes diverse code, there are no references supporting the assumption that generating more diverse code is better.
- Code is poorly written / formatted.

**Questions:**

## Questions
- Was an ablation study performed on the number of points saved for each paper's methods in Step 1 "Gather domain knowledge"? How does increasing/ descresing point summaries affect the final model performance?
- Why were other models not tested on the case studies? A lot seems to be riding on TusoAI's new findings, but other models were not given a chance.
- How is the iterative refinement done for "The initialization agent Ainit drafts 5 candidate solution descriptions from T and iteratively refines them using each paper summary in P"? It is not clear and appears to be super important.
- How many hyperparameters are there total for this agent? It appears to have a lot, and I would be curious how those hyperparameters affect the solutions. (ie. run time, integer values used in various places, utility scalars, probabilities, bug-fix attempts, etc..)
- Which hardware is used for all of this? 8 hours means a totally different thing on a laptop vs. a data center. Are all models tested on the same hardware? I see some models are only accessible on the web: how do you account for different compute capabilities?
- How many runs were used for ablation analysis (Table 3)? Was it just run once, or several times with mean score? If it was run once then it needs to be run more times, since LLMs are very stochastic and the results can vary significantly.
- Where is the conclusion/future work section? The paper seems to end abruptly.

## Feedback
- The code is a mess. There are import statements all over the place, comments are unstructured and lacking in many places, and there are no clear instructions on how to reproduce results. Please fix this so I can take a look at the code myself and run things locally.
- I_{diag} is not updated in Algorithm 1, it appears to remain as an empty list. Please consider removing or correct this.
- Line 265: Change "instruction by first uniformly draws" to "instruction by first uniformly drawing".
- The claims (line 297, first paragraph in section 4.1) that TusoAI provides "novel" and "computationally efficient" methods are not substantiated. Provide quantitative results for computational efficiency and a better empirical comparison of the results across all models. Simply stating the method TusoAI came up with does not mean it's novel.
- Please make it clear which model is presented in figure 2B in the caption. This specific sub-figure would benefit from a better description in general.
- Please provide motivation for why we care about generating diverse code. I would argue that if an agent can produce better methods, then it doesn't matter how diverse the methods produced along the way are. This is a central point studied in the paper that needs to be substantiated. In fact, producing lots of diverse models is at odds with the 8 hour time limit, since I can also imagine a model would benefit from trying to iterate on a model that works decently well instead of making smaller changes to a larger amount of models in the same period of time.
- Line 386, please correct: "Table ??".
- Line 430: "This improvement reflects TusoAI’s ability...", is not really a fair comparison. TusoAI started from the work the original authors had already done. Please correct this sentence to show how TusoAI expands upon the original work in a short amount of time, instead of making it seem it tested 167 unique versions _from scratch_.
  - Same comment for line 464 for the second case study.
- The text in the figures are tiny. Please make the text larger so it's easier to read instead of having to zoom in 300% to make out what it says.
- Line 1244: "It's samples" change to "Its samples".

---

> ### Author Response · Authors · 2025-11-23
> **Response to Reviewer x86D (1/6)**
>
> Thank you for noting the usefulness of TusoAI and for providing detailed comments that helped us substantially improve the manuscript. We address each of the reviewer’s points below. The manuscript has been revised accordingly, incorporating 10 new analyses and updates to 3 main figures, and 13 supplementary sections. Before proceeding to the point-by-point responses, we briefly address the reviewer’s 4 high-level concerns:
>
> (i) TusoAI did not significantly outperform existing models.
>
> We agree that the improvements on the tasks are moderate rather than dramatic. This outcome is largely driven by the nature of these problems, which differ substantially from typical ML-agent benchmarks (e.g., Kaggle, MLE-Bench). Many of our tasks—such as scRNA-seq denoising and cell-type deconvolution—are highly specialized scientific workflows that have been iteratively optimized by domain experts over many years [1]. We emphasize that in such settings, even modest gains over strong human-engineered baselines are uncommon, and our improvements to scientific tasks are comparable or even greater than the improvements that led to expert methods being published. See examples below:
>
> In the original scDRS paper Figure 2 (corresponding to our Figure 3), the authors report a 20-30% power improvement over the strongest baseline in causal simulations, whereas we discover a 40% improvement over scDRS.  In the original pgBoost paper Figure 3 (corresponding to our Figure 4), the authors report a 5-8% enrichment improvement over the strongest baseline, whereas we discover a further 7-14% improvement over pgBoost. This improvement is particularly large at distal links (emphasized in pgBoost as a very important case), where our enrichment is double that of pgBoost.
> Agentic optimization work by Aygun et al reports similarly modest gains, in Figure 2 of their paper they improve a single cell task from 0.5 to 0.6 and an expert deep learning model from 0.76 to 0.82, without comparing to NAS methods. Agentic optimization work for biology BioML bench finds that agentic approaches (such as AIDE or Biomni) often struggle to meet expert baselines, let alone outperform them. We show TusoAI can often outperform expert methods, both from scratch and by improving upon existing work.
> Deep learning tasks have been historically more difficult for agentic approaches and not the main focus (AIDE did not consider deep learning originally, AutoML-Agent does not outperform AutoML and humans in their deep learning tasks, CellForge is an agentic approach that does not outperform human deep learning models). This is replicated in our deep learning analysis where AIDE has an average rank of 9.8. In contrast, TusoAI is the top ranked (Avg. Rank 2.8) among all methods (including experts and AutoML). Thus, TusoAI being the top performing agentic NAS method, at least as good as the top performing NAS method, and meeting or outperforming human experts in 3 out of 5 deep learning tasks is indeed an important advance for agentic optimization performance.
>
> [1] M. D. Luecken, S. Gigante, D. B. Burkhardt, et al. Defining and benchmarking open problems in single-cell analysis. Nature Biotechnology
>
> (ii) Other models not run on case studies.
>
> For the two case studies (scDRS and pgBoost), we compared only against the original human-designed methods rather than other agentic baselines because these methods have large, specialized codebases that TusoAI can directly build upon, whereas other agentic baselines would require full code re-implementation, which is not practical. Optimizing existing methods is a core consideration of TusoAI, but less explored in related work. Additional details are provided in our response to below.
>
> (iii) Lack of case studies in other domains.
>
> We focused our case studies on genetics for two reasons. First, these problems—detecting disease-critical cell populations and linking variants to target genes—are among the most complex and high-impact problems in human genetics and are central to understanding disease mechanisms and therapeutic development (Lappalainen 2021 Science; Song 2025 Nature; Zhang 2022 Nat Genet; Dorans 2025 Nat Genet; Sakaue 2024 Nat Genet). Second, our goal was to demonstrate depth rather than breadth: each case study is a full scientific analysis in which TusoAI improves state-of-the-art methods (scDRS, pgBoost), uncovers additional biologically plausible findings, and yields ready-to-use implementations that would typically take years to develop manually. We acknowledge that our empirical focus is narrower in domain, and we clarify that although our evaluation centers on genomics and genetics due to their complexity and relevance, TusoAI is designed as a general framework applicable to other scientific domains (line 536).

---

> ### Author Response · Authors · 2025-11-23
> **Response to Reviewer x86D (2/6)**
>
> (iv) Code is poorly written.
>
> We agree that clear, easy-to-use code is important. We have substantially refactored the implementation (cleaned imports, restructured modules, improved documentation and reproduction instructions) and uploaded the revised version in the supplementary. We welcome further suggestions if the reviewer encounters any specific issues.
>
> The reviewer has raised 3 comments/weaknesses and 15 main questions/feedback (summarized from 7 questions and 10 feedbacks in the Questions section). We believe that we have completely addressed 16 out of these 18 points (W1,W2,W3,Q1,Q2,Q3,Q5,Q6,Q7,Q9,Q10,Q11,Q12,Q13,Q14,Q15,) and substantially addressed the remaining points (Q4,Q8). Please let us know if there are any additional questions or if further clarification would be helpful. If the reviewer feels that their concerns have been adequately addressed, we would be grateful if they would consider updating their evaluation.
>
> >TusoAI is designed to work with any domain as long as domain knowledge is provided, however, there is a strong focus on biological applications. This paper would benefit from showing TusoAI's performance in other scientific domains similar to the two biological case studies. I see that it worked on deep learning but only a cold start was performed. I am specifically asking for a warm start case study in a non-biology domain.
>
> x86D-W1: Warm start in a non-biology domain.
>
> Thank you for the comment. We agree that demonstrating TusoAI’s ability to improve existing methods in a non-biology scientific domain would strengthen the paper. In response, we conducted a new warm-start experiment on the NinaPro deep-learning task. In this experiment, TusoAI was initialized with the expert model rather than starting from scratch. While neither the TusoAI cold-start nor the original expert model achieved the top performance, the warm-started TusoAI discovered a new architecture—combining temporal convolutional networks with a mixture-of-experts design—that outperformed all baselines on this benchmark. We have added these results in Section 5 (lines 442-443) and Appendix T.
>
> | | AIDE | TusoAI | Expert | WRN ASHA | TusoAI + Expert |
> |---------|------|--------|--------|----------|------------------|
> | NinaPro | 0.86 | 0.90   | 0.91   | 0.93     | 0.94             |
>
>
> We also refer the reviewer to our response to high-level concern (iii), where we explain why our primary case studies focused on genetics: these domains allow us to evaluate the full cycle of scientific method development, not only metric improvement. In these settings, TusoAI improved state-of-the-art methods (scDRS, pgBoost), uncovered additional biologically plausible findings, and produced ready-to-use implementations that would normally take researchers years to develop manually.
>
> >Diversity in methods produced is a core architecture consideration, but it is not substantiated in the text. While evidence is provided that TusoAI writes diverse code, there are no references supporting the assumption that generating more diverse code is better.
>
> x86D-W2: Connection between diversity and performance.
>
> Thank you for this insightful comment. We agree that the connection between diversity and performance is important to substantiate. First, we performed a new analysis examining the optimization trajectories of all 13 tasks (Appendix J, K). We found that performance plateaus when diversity collapses (correlation between optimization plateau and diversity collapse Pearson’s R = 0.51, Appendix M). Second, the importance of maintaining diversity is supported by several related works on program synthesis and agentic optimization. For example, FunSearch [3] explicitly maintains a pool of diverse programs to avoid local optima through island-based evolution; MLE-STAR [4] incorporates diversity-promoting feedback in the refinement loop; and Aygun et al. [2] emphasize that generating diverse candidate solutions is essential for discovering high-performing ones. These works collectively motivate why diversity is a principled design choice rather than an arbitrary heuristic. We have updated Appendix J, K, M and Lines 359-360, Line 368 to incorporate these new analyses and to clearly articulate the motivation for encouraging diversity in TusoAI.
>
>
> [2] Eser Aygun, et al. An ai system to help ocientists write expert-level empirical software. arXiv preprint
>
> [3] Bernardino Romera-Paredes, Mohammadamin Barekatain, Alexander Novikov, Matej Balog, M Pawan Kumar, Emilien Dupont, Francisco JR Ruiz, Jordan Ellenberg, Pengming Wang, Omar Fawzi, et al. Mathematical discoveries from program search with large language models. Nature
>
> [4] Jaehyun Nam, Jinsung Yoon, Jiefeng Chen, Jinwoo Shin, Sercan O Arık, and Tomas Pfister. Mle- ¨star: Machine learning engineering agent via search and targeted refinement. arXiv preprint

---

> ### Author Response · Authors · 2025-11-23
> **Response to Reviewer x86D (3/6)**
>
> >Code is poorly written / formatted.
>
> x86D-W3: Code quality.
> We agree that clear, easy-to-use code is important. We have substantially refactored the implementation.Specifically, we have separated the core functionalities of TusoAI into separate files and greatly improved the code readability. In particular, run_tusoai.ipynb is the main starting point and provides easy functionality for running each step and viewing the results. We uploaded the revised version. We welcome further suggestions if the reviewer encounters any specific issues.
>
> >Was an ablation study performed on the number of points saved for each paper's methods in Step 1 "Gather domain knowledge"? How does increasing/ descresing point summaries affect the final model performance?
>
> x86D-Q1: Ablation study on number of points per paper.
>
> Thank you for the question. We performed a new ablation in which we varied the number of summary points extracted per paper (5 and 15, instead of the default 10, 5 replicates of each). We found that model performance remained largely unchanged across these settings, indicating that TusoAI is robust to this hyperparameter. We have added these results to Appendix O (lines 401-403). Here is the median across 5 replicates.
>
> | Summary Type      | Denoise | SVG  | Decomposition | ECG  | Satellite |
> |-------------------|---------|------|----------------|------|-----------|
> | 5 point summary   | 0.35    | 0.85 | 0.64           | 0.64 | 0.88      |
> | 15 point summary  | 0.35    | 0.82 | 0.55           | 0.63 | 0.89      |
> | 30 point summary  | 0.35    | 0.84 | 0.54           | 0.64 | 0.87      |
>
> >Why were other models not tested on the case studies? A lot seems to be riding on TusoAI's new findings, but other models were not given a chance.
>
> x86D-Q2: Reasons for not including other methods in case studies.
>
> Thank you for the question. For the two case studies (scDRS and pgBoost), we compared only against the original human-designed methods rather than other agentic baselines because these methods have large, multi-file codebases (e.g., scDRS contains >3,000 lines across 5 modules). TusoAI is designed to operate on a targeted function within a larger codebase, allowing it to modify critical components without rewriting the entire system. In contrast, existing agentic systems such as AIDE, Biomni, AutoML-Agent, DS-Agent, and R&D-Agent operate by editing or regenerating whole Python scripts. In practice, these systems could not handle multi-file scientific pipelines and failed to run without extensive re-engineering. Thus, a direct comparison in these settings was not technically feasible. This task of warm-start scientific optimization is less explored in related works.
>
> For this reason, we focused on comparing state-of-the-art human methods from the original papers, which represent the strongest and most relevant baselines for these highly specialized scientific tasks. We have updated Section 5 (lines 440-441) to explain these points.
>
> >How is the iterative refinement done for "The initialization agent Ainit drafts 5 candidate solution descriptions from T and iteratively refines them using each paper summary in P"? It is not clear and appears to be super important.
>
> x86D-Q3: Solution initialization.
>
> Thank you for the question. In our implementation, Ainit processes the paper summaries in P sequentially, operating on all solution descriptions. In practice, for each summary, Ainit either refines one of the existing candidate solutions using the information introduced in that summary or, when appropriate, drafts a new candidate solution that reflects a distinct modeling idea suggested by the summary. The prompts used for this refinement process are provided in Appendix U.4. We have also added a concrete example of the full refinement for the single-cell denoising task in Appendix G to improve clarity.

---

> ### Author Response · Authors · 2025-11-23
> **Response to Reviewer x86D (4/6)**
>
> >How many hyperparameters are there total for this agent? It appears to have a lot, and I would be curious how those hyperparameters affect the solutions. (ie. run time, integer values used in various places, utility scalars, probabilities, bug-fix attempts, etc..)
>
> x86D-Q4: Hyperparameters.
>
> Thank you for the question. We agree that understanding the influence of hyperparameters is important. To address this, we performed new analyses evaluating the effect of several representative hyperparameters on performance, including (i) the number of papers retrieved during literature search (0, 50, versus the default 10), and (ii) the number of bullet-points in the summary extracted per paper (5, 30 versus default 15). This is in addition to the 4 ablations in the original submission. Across these variations, TusoAI’s default setting generally achieved top performance (but not by far), indicating reasonable default settings and general robustness.
>
> More broadly, while the system contains additional hyperparameters (e.g., numerical thresholds, utility weights, and retry limits), these were fixed across all tasks and datasets, reducing the likelihood of overfitting. Exhaustively varying every parameter is impractical, but the representative analyses above suggest that the system is not highly sensitive to reasonable changes. We ablated the parameters we believe most represented our methodological contributions. We have updated Section 4.2 (lines 401-403) to describe these results.
>
> >Which hardware is used for all of this? 8 hours means a totally different thing on a laptop vs. a data center. Are all models tested on the same hardware? I see some models are only accessible on the web: how do you account for different compute capabilities?
>
> x86D-Q5: Hardware.
>
> Thank you for the question. We ensured that the computational resources were comparable and fair across all baselines. Specifically, TusoAI and AIDE were each run for the same duration (8 hours) on the same hardware and under identical conditions, including temperature, conda environment, LLM, task description, dataset, and evaluation function. Biomni and ChatGPT-Agent do not support fixed time budgets; following the recommendation of the Biomni authors, we allowed these agents to run until self-termination. Details are provided in Appendix F. We have also updated lines 302-303 in an effort to improve clarity.
>
> >How many runs were used for ablation analysis (Table 3)? Was it just run once, or several times with mean score? If it was run once then it needs to be run more times, since LLMs are very stochastic and the results can vary significantly.
>
> x86D-Q6: Number of runs in ablation studies.
>
> Thank you for the question. In the original submission, each experiment was run once and the confidence intervals were computed across datasets. We therefore added four additional replicates for each task in all ablation studies and found identical patterns in performance, specifically that removing each component lowers overall performance, which is consistent with the original results.. We have updated Section 4.2 (lines 401-403) to report these findings.

---

> ### Author Response · Authors · 2025-11-23
> **Response to Reviewer x86D (5/6)**
>
> >Where is the conclusion/future work section? The paper seems to end abruptly.
>
>
> x86D-Q7: Conclusion/future work section.
>
> Thank you for this comment. We have added a new Discussion section summarizing the main conclusions, limitations, and directions for future work (lines 523-529), as shown below:
>
> “We have presented TusoAI, an agentic system for scientific method optimization. By mimicking a scientist's cycle of method development, TusoAI achieves superior performance on single cell and scientific deep learning benchmarks, and is further able to discover significant optimizations to state of the art methods in genetics which revealed new biology missed by existing methods. We believe TusoAI represents a promising step towards automated scientific method development and optimization, thus accelerating scientific discovery.
>
> We acknowledge several limitations and areas for future work. TusoAI requires a separate validation experiment to base optimization on to prevent overfitting. This experiment should also be quick to run while representing final performance. When optimizing an existing method, TusoAI performs strongest when most of the method is in a single function, and may not perform well if the method is scattered throughout a large codebase. TusoAI does not consider making new evaluation procedures, but rather relies on existing ones and will be vulnerable to the same weaknesses those may have. Several algorithm components are heuristic and may benefit from theoretic justification. We generally focus on biological applications given their complexity and relevance, but achieve reliable performance in diverse scientific domains. Future work may include processing multiple functions in parallel by treating them as separate subtasks, or searching over not just the code-space for methods, but also the data-space for additional useful data to include in a method, as is common in scientific domains.”
>
> >The code is a mess. There are import statements all over the place, comments are unstructured and lacking in many places, and there are no clear instructions on how to reproduce results. Please fix this so I can take a look at the code myself and run things locally.
>
> x86D-Q8: Code quality.
>
> Thank you for the feedback. We apologize for the earlier state of the code and have undertaken a thorough refactoring to address the issues you raised. Specifically, we have separated the core functionalities of TusoAI into separate files and greatly improved the code readability. In particular, run_tusoai.ipynb is the main starting point and provides easy functionality for running each step and viewing the results. The updated codebase is included in the supplementary zip file. We hope this new version is substantially easier to navigate and to run locally. Please feel free to reach out if any part of the pipeline is unclear or if you would like guidance on reproducing a particular task.
>
> >I_{diag} is not updated in Algorithm 1, it appears to remain as an empty list.
>
> x86D-Q9: I_{diag}.
>
> Thank you for pointing this out. It is initialized as a predefined list of diagnostic instructions, and we have corrected the pseudocode accordingly (line 202).
>
> >The claims (line 297, first paragraph in section 4.1) that TusoAI provides "novel" and "computationally efficient" methods are not substantiated. Provide quantitative results for computational efficiency and a better empirical comparison of the results across all models. Simply stating the method TusoAI came up with does not mean it's novel.
>
> x86D-Q10: Novelty and computational efficiency.
>
> Thank you for raising this point. You are correct that the claims of novelty and computational efficiency should be substantiated more clearly, and we have updated the manuscript accordingly.
>
> For novelty, we have clarified why the TusoAI-optimized methods in the four qualitative case studies are distinct from existing approaches. In particular, we now explain how the discovered design choices lie outside AutoML search spaces and why they are not simple reimplementations of related methods. Specifically, we do a literature validation of the tasks to identify what novel components TusoAI has implemented, such as modelling count noise and dropout for denoising and combining smooth regression with neighborhood covariates in SVG. In deep learning tasks, we additionally validate that our models discovered lie outside the standard NAS search space. We have added Section 4.1 (line 318) and Appendix H andto clarify these points.
>
> For computational efficiency, we now provide quantitative comparisons. We report the end-to-end testing runtime of each baseline across tasks. The methods produced by TusoAI run in under 3 minutes on single-cell tasks and under 8 minutes on deep-learning tasks, substantially faster than the corresponding baselines (8 minutes and 30 minutes, respectively for AIDE). We added Section 4.1 ( lines 318-320) and Appendix I  to report these findings.

---

> ### Author Response · Authors · 2025-11-23
> **Response to Reviewer x86D (6/6)**
>
> >Please make it clear which model is presented in figure 2B in the caption. This specific sub-figure would benefit from a better description in general.
>
> x86D-Q11: Clarity of Figure 2B.
>
> Thank you for this comment. We agree that clear figures are essential for readability, and we have made several improvements accordingly. Specifically, we have (i) increased font sizes throughout the figures to enhance readability, and (ii) revised and expanded the figure legends and captions for greater clarity.
>
> >Please provide motivation for why we care about generating diverse code. I would argue that if an agent can produce better methods, then it doesn't matter how diverse the methods produced along the way are. This is a central point studied in the paper that needs to be substantiated. In fact, producing lots of diverse models is at odds with the 8 hour time limit, since I can also imagine a model would benefit from trying to iterate on a model that works decently well instead of making smaller changes to a larger amount of models in the same period of time.
>
> x86D-Q12: Motivation for generating diverse code.
>
> Thank you for this insightful comment. We agree that the connection between diversity and performance is important to substantiate. First, we performed a new analysis examining the optimization trajectories of all 13 tasks (Appendix J, K). We found that performance plateaus when diversity collapses (correlation between optimization plateau and diversity collapse Pearson’s R = 0.51, Appendix M). Second, the importance of maintaining diversity is supported by several related works on program synthesis and agentic optimization. For example, FunSearch [3] explicitly maintains a pool of diverse programs to avoid local optima through island-based evolution; MLE-STAR [4] incorporates diversity-promoting feedback in the refinement loop; and Aygun et al. [2] emphasize that generating diverse candidate solutions is essential for discovering high-performing ones. These works collectively motivate why diversity is a principled design choice rather than an arbitrary heuristic. We have updated Appendix J, K, M and Lines 359-360, Line 368 to incorporate these new analyses and to clearly articulate the motivation for encouraging diversity in TusoAI.
>
> [2] Eser Aygun, Anastasiya Belyaeva, Gheorghe Comanici, Marc Coram, Hao Cui, Jake Garrison, Renee Johnston Anton Kast, Cory Y McLean, Peter Norgaard, ZahraShamsi, et al. An ai system to help ocientists write expert-level empirical software. arXiv preprint
>
> [3] Bernardino Romera-Paredes, Mohammadamin Barekatain, Alexander Novikov, Matej Balog, M Pawan Kumar, Emilien Dupont, Francisco JR Ruiz, Jordan Ellenberg, Pengming Wang, Omar Fawzi, et al. Mathematical discoveries from program search with large language models. Nature
>
> [4] Jaehyun Nam, Jinsung Yoon, Jiefeng Chen, Jinwoo Shin, Sercan O Arık, and Tomas Pfister. Mle- ¨star: Machine learning engineering agent via search and targeted refinement. arXiv preprint
>
> >Line 386, please correct: "Table ??".
>
> x86D-Q13: Line 386. We have corrected this typo.
>
> >Line 430: "This improvement reflects TusoAI’s ability...", is not really a fair comparison. TusoAI started from the work the original authors had already done. Please correct this sentence to show how TusoAI expands upon the original work in a short amount of time, instead of making it seem it tested 167 unique versions from scratch.
>
> x86D-Q14: Line 430.
>
> Thank you for the feedback. We agree that the original wording could give the impression that TusoAI generated 167 versions entirely from scratch. To avoid this misunderstanding, we have revised the sentence to: “This improvement reflects TusoAI’s ability to efficiently explore variations built on the original method: it tested 167 such variations within 24 hours at a cost of $0.37, whereas the original authors evaluated fewer than 10 versions over 3 months.” (lines 456–458).
>
> >The text in the figures are tiny. Please make the text larger so it's easier to read instead of having to zoom in 300% to make out what it says.
>
> x86D-Q15: Figure font size. We have increased font sizes for Figures 2, 3, and 4 to improve readability.

---

> ### Comment · Reviewer_x86D · 2025-11-25
> **Updated Review**
>
> Thank you for your detailed response to my questions and concerns. I read through everything and appreciate the updates. I mostly appreciate the updated code, it looks a lot cleaner than before. I have updated my rating from a 2 to a 4. My reason for not going over the line to acceptance is because the code does not work and still is a bit messy. There are imports all over the place, inconsistent comments and docstrings, and the code `run_tusoai.ipynb` does not work as provided.
>
> Particularly, in `optimizer.py`:
> - `from pathlib import Path` appears twice (line 7, 26)
> - `import os` appears three times (line 12, 17, 142)
> - `import random` appears inside a function despite already being imported at the top of the code twice (line, 3, 18, 470)
> - These errors appear in other files as well. I recommend using something like "black" and other python packages to clean up the code.
>
> In `run_tusoai.ipynb`:
> - On the call: `best_model, optimization_history = tusoai.discover_method(llm_model=LLM_MODEL,...`
> - No such file or directory: '/home/user/tusoai/single_cell_denoise/denoise0_k_nearest_neighbors_.py'
>
> I am satisfied with the contents of the paper but can't recommend for acceptance if the code doesn't work. It makes me lose trust in the results.

---

> > ### Author Response · Authors · 2025-12-01
> > **Followup response and code update**
> >
> > x86D-R1
> >
> > Thank you for carefully reviewing our code and for the constructive feedback. We recognize that it is our responsibility to provide clean, functional, and well-documented open-source code, which is essential for transparency and reproducibility. In response to your updated review, we have made the following improvements:
> >
> > First, we substantially improved code readability and consistency. We removed redundant and duplicate import statements. We also cleaned up comments to ensure that all functions and modules are easier to understand.
> >
> > Second, we now provide a fully runnable example for single-cell denoising so the reviewer can run TusoAI end-to-end. Previously, we did not include example data because the input files typically exceed the 100 MB upload limit. We have now curated a smaller (70 MB) example dataset that fits within the size constraints while still demonstrating the full workflow. Running run_tusoai.ipynb with this dataset now works out of the box. Simply provide the API keys and run all cells in run_tusoai.ipynb. If tusoai.generate_summaries returns a 429 error it is because the semantic scholar rate limit has been hit, try waiting a few minutes.
> >
> > Third, to further improve credibility, we have uploaded the full optimization trajectories for all tasks for both TusoAI and AIDE. These files show the step-by-step history of how each method was generated and allow users to trace the evolution of the optimized models.
> >
> > All new materials are included in the updated supplementary zip file (updated on 12/1):
> > - New example data: openproblems_datasets/1k_pbmc_processed.h5ad, runnable under run_tusoai.ipynb
> > - Example histories: under the history/ directory

---

### Official Review · Reviewer_kn4q · 2025-10-26

**Soundness:** 2
**Presentation:** 1
**Contribution:** 2
**Rating:** 2
**Confidence:** 4

**Summary:**

The paper presents TusoAI, a method that hand designs a set of steps for automated model discovery using LLMs. The authors compare their method against baseline methods on set of task and find that their approach achieves good performance with high diversity in the suggested solutions.

**Strengths:**

- the problem of automated computational model design is important
- ablation studies are provided to reveal some insight into the importance of the methods components
- strong results are achieved on a set of tasks

**Weaknesses:**

- the paper lacks any discussion or conclusion
- the Bayesian update step is unclear and not described
- the method seems overall very heuristic. A (theoretic) justification of the algorithm would be of great value.
- additionally the algorithm is rather complex and the explanation are difficult to follow at times
- figure are sometimes unclear and would need re-working

**Questions:**

- what does your acronym TusoAI stand for?
- Table 1 and 2: How is performance defined in these tasks?
- Is the comparison in the experimental section fair in terms of computational load? How long are baseline methods run compared to the 8h for TusoAI?
- Figure 2, 3 and 4: font size is too small and hard to read.
- line 323: How were the text embeddings computed?
- Figure 2B: Does this "optimization trajectory" look similar for all other tasks? Are there examples where the algorithm converges fast and then does not find a better solution?
- In the methods there is no explanation of how the Bayesian update is performed. Can you explain where and how exactly a Bayesian update is performed and what the prior and posteriors involved are?
- Table 3: Do the ablations only disable one of the parts (i.e. "No Bayesian" only disables the Bayesian part but keeps all other parts)?
- line 380: How do you define the mean time to optimize? What is the criterion that optimization is achieved?
- line 385: "Results are shown in Table ??"
- Figure 3 B/C: What does the circles show? How should we read and interpret those figures?
- Figure 4:
  - what does the distance threshold mean?
  - How should read and interpret figure 4C?

---

> ### Author Response · Authors · 2025-11-23
> **Response to Reviewer kn4q (1/4)**
>
> Thank you for noting that the empirical results of TusoAI are strong. We address each of the review comments in detail below. The manuscript has been revised accordingly, incorporating 10 new analyses and updates to 3 main figures, and 13 supplementary sections.
>
> We also wish to provide additional context regarding the case studies, which were not discussed in detail in the review but represent a central part of our contribution. These case studies focus on two high-impact problems in human genetics: (i) detecting disease-critical cell populations and (ii) linking genetic variants to their target genes; both problems are central to understanding disease mechanisms and advancing therapeutic development (Lappalainen 2021 Science). These topics are the focus of many recent publications in top journals (e.g., Song 2025 Nature; Ma 2023 Cell Genom; Zhang 2022 Nat. Genet; Dorans 2025 Nat. Genet; Sakaue 2024 Nat. Genet; Xie 2023 Cell Genom). In our study, we rigorously evaluated TusoAI-optimized versions of two state-of-the-art methods (scDRS, Zhang 2022 Nat Genet; pgBoost, Dorans 2025 Nat Genet) using the same simulations and benchmarking protocols as the original papers. The TusoAI-optimized methods consistently outperformed the originals, uncovered additional biologically plausible findings missed by prior methods, and produced ready-to-use improved implementations that would typically take researchers years to develop manually.
>
> The reviewer has raised 5 comments/weaknesses and 12 questions. We believe that we have completely addressed 15 out of these 17 points (W1,W2,W5,Q1,Q2,Q3,Q4,Q5,Q6,Q7,Q8,Q9,Q10,Q11,Q12) and substantially addressed the remaining points (W3,W4). Please let us know if there are any additional questions or if further clarification would be helpful. If the reviewer feels that their concerns have been adequately addressed, we would be grateful if they would consider updating their evaluation.
>
> >the paper lacks any discussion or conclusion
>
> kn4q-W1: Discussion and conclusion.
>
> Thank you for this comment. We have added a new Discussion section summarizing the main conclusions, limitations, and directions for future work at Section 6, as shown below:
>
> “We acknowledge several limitations and areas for future work. TusoAI requires a separate validation experiment to base optimization on to prevent overfitting. This experiment should also be quick to run while representing final performance. When optimizing an existing method, TusoAI performs strongest when most of the method is in a single function, and may not perform well if the method is scattered throughout a large codebase. TusoAI does not consider making new evaluation procedures, but rather relies on existing ones and will be vulnerable to the same weaknesses those may have. Several algorithm components are heuristic and may benefit from theoretic justification. We generally focus on biological applications given their complexity and relevance, but achieve reliable performance in diverse scientific domains. Future work may include processing multiple functions in parallel by treating them as separate subtasks, or searching over not just the code-space for methods, but also the data-space for additional useful data to include in a method, as is common in scientific domains.”
>
>
> >the Bayesian update step is unclear and not described
>
> kn4q-W2: Details for the Bayesian update.
>
> Thank you for the question. We recognize that it is our responsibility to provide a clear exposition. The Bayesian update in TusoAI corresponds to lines 2, 8, and 9 of Algorithm 1. Each instruction category c is initialized with a prior probability c (line 2). At each iteration, a category is sampled according to these probabilities (line 8), and if the resulting instruction produces an improved solution, its probability is updated multiplicatively to c1.1c (line 9). We have updated lines 257-259 to clarify these points.
>
> We also provide additional rationale for this design. The Bayesian update step serves as a heuristic that mimics how a scientist might prioritize among several categories of actions (e.g., refining preprocessing, revisiting modeling assumptions, or modifying optimization strategies). The update increases the likelihood of selecting categories that have previously led to performance improvements, thereby implementing an implicit optimization over expected performance gain. Alternatively, a fully formalized optimization formulation could yield a more principled Bayesian update procedure. For instance, one could adopt a Thompson Sampling-style formulation in which each category is associated with a reward probability pc governed by a Beta prior, and at each iteration the category with the largest sampled value is selected. We have clarified this interpretation and discussed this limitation in Section 6 (line 535).

---

> ### Author Response · Authors · 2025-11-23
> **Response to Reviewer kn4q (2/4)**
>
> >the method seems overall very heuristic. A (theoretic) justification of the algorithm would be of great value.
>
> kn4q-W3: Justification of the method.
>
> Thank you for the comment. We agree that TusoAI is largely heuristic and that proper justification of its components is important. While providing a rigorous theoretical analysis for agentic systems remains challenging—largely due to the complex interactions among LLM reasoning, external tools, stochastic sampling, and iterative feedback loops—we justify our design through empirical evidence and high-level reasoning.
>
> First, our ablation studies demonstrate that each key component of TusoAI—the knowledge-tree structure, the Bayesian update mechanism, and the diagnosis-based optimization—contributes independently to performance improvements (Table 3). This provides empirical support for the necessity of these design choices.
>
> Second, we provide the high-level intuitions guiding the design of TusoAI, which aims to mimic how human scientists iteratively develop computational methods:
>
> (i) Knowledge tree & categorical structure: This reframes the optimization problem from “improve the model” to “improve the model by doing X”. This hierarchical structure provides guidance for the optimization process, analogous to how few-shot prompting provides additional structure compared to zero-shot prompting.
>
> (ii) Bayesian update: If a scientist tries an idea from a particular conceptual category (e.g., preprocessing) and observes improved performance, they are more likely to revisit that category. The Bayesian update encodes this intuition by increasing the probability of selecting successful categories.
>
> (iii) Diagnosis-based optimization: As a model evolves, new diagnostic information becomes available (e.g., intermediate representations or residual patterns). Reviewing such signals closely resembles how scientists debug or refine methods beyond simply reprinting original metrics.
>
> We have updated Section 1(lines 62), Section 3 (lines 267-269), and Section 6 (line 535) to clarify these points and explicitly discuss the heuristic nature and limitations of the approach.
>
> >additionally the algorithm is rather complex and the explanation are difficult to follow at times
>
> kn4q-W4: Presentation of the algorithm.
>
> Thank you for the comment. We recognize that it is our responsibility to provide a clear exposition of the algorithm. We have thus made the following updates in an effort to improve clarity:
>
> (i) Expanded high-level overview: We revised the introduction of TusoAI to provide a clearer, high-level intuition for the overall design (lines 62-64; see also our response to kn4q-W3).
>
> (ii) More detailed step-by-step description: We added additional explanations for each step of the method in the Methods section, including clarifying the role of each component and its interaction within the workflow (lines 190-192, 267-269).
>
> We would be grateful if the reviewer could point to any specific parts of the Methods section that remain unclear; we are happy to further refine the exposition.
>
> >figure are sometimes unclear and would need re-working
>
>
> kn4q-W5: Figure clarity
>
> Thank you for this comment. We agree that clear figures are essential for readability, and we have made several improvements accordingly. Specifically, we have (i) increased font sizes throughout the figures to enhance readability, and (ii) revised and expanded the figure legends and captions for greater clarity. In total, we have updated Figures 2, 3, and 4 to improve presentation quality.
>
> >what does your acronym TusoAI stand for?
>
>
> kn4q-Q1: What does TusoAI stand for.
>
> Thank you for the question. TusoAI doesn’t particularly stand for something, it’s just something unique.
>
> >Table 1 and 2: How is performance defined in these tasks?
>
> kn4q-Q2: Performance metrics in Tables 1,2.
>
> Thank you for the question. Performance in Tables 1 and 2 is computed on held-out test sets using each task’s standard evaluation protocol. Full details of the evaluation setup for all single-cell tasks are provided in Appendix D, and for deep-learning tasks in Appendix E. Specifically, we use largely the same metrics from the original benchmarks. We have updated Section 4 (lines 288-289) and Appendix D and E to make these definitions explicit and to clarify that each method is evaluated under identical conditions for fair comparison.

---

> ### Author Response · Authors · 2025-11-23
> **Response to Reviewer kn4q (3/4)**
>
> >Is the comparison in the experimental section fair in terms of computational load? How long are baseline methods run compared to the 8h for TusoAI?
>
> kn4q-Q3: Computational load across baselines.
>
> Thank you for the question. We ensured that the computational resources were comparable and fair across all baselines. Specifically, TusoAI and AIDE were each run for the same duration (8 hours) on the same hardware and under identical conditions, including temperature, conda environment, LLM, task description, dataset, and evaluation function. Biomni and ChatGPT-Agent do not support fixed time budgets; following the recommendation of the Biomni authors, we allowed these agents to run until self-termination. Details are provided in Appendix F. We have also updated Section 4 (lines 302-303) in an effort to improve clarity.
>
> >Figure 2, 3 and 4: font size is too small and hard to read.
>
> kn4q-Q4: Fontsize for Figures 2,3,4.
>
> Please see response to kn4q-W5.
>
> >line 323: How were the text embeddings computed?
>
> kn4q-Q5: Text embeddings.
>
> Thank you for the question. We computed text embeddings using the scikit-learn TF–IDF vectorizer applied to cleaned code (with comments, imports, repetitive data loading and evaluation code removed). We have updated Section 4.1 (lines 322-323) to clarify this.
>
> >Figure 2B: Does this "optimization trajectory" look similar for all other tasks? Are there examples where the algorithm converges fast and then does not find a better solution?
>
> kn4q-Q6: Optimization trajectory.
>
> Thank you for the question. To address the reviewer’s concern, we analyzed the optimization trajectories of TusoAI across all 11 benchmark tasks. In the majority of tasks (9 out of 11), TusoAI continued to discover new and improved solutions throughout the full optimization window. For relatively easier tasks (e.g., label projection), meaningful improvements were achieved early within the first ~2 hours, and TusoAI tied for first place. Importantly, we did not observe any cases where the algorithm converged early and subsequently achieved low performance despite available time budget. We have updated Section 4.1 (line 368) accordingly and added a new Appendix J that reports the full optimization trajectories for all tasks.
>
> >In the methods there is no explanation of how the Bayesian update is performed. Can you explain where and how exactly a Bayesian update is performed and what the prior and posteriors involved are?
>
> kn4q-Q7: Details for Bayesian update.
>
> Please see response to kn4q-W2.
>
>
> >Table 3: Do the ablations only disable one of the parts (i.e. "No Bayesian" only disables the Bayesian part but keeps all other parts)?
>
> kn4q-Q8: Ablation.
>
> Thank you for the question. Yes, each ablation disables only one component at a time while keeping all other parts of the system unchanged. This allows us to isolate and quantify the individual performance contribution of each component. We have updated Section 4.2 (lines 373) to clarify this.
>
> >line 380: How do you define the mean time to optimize? What is the criterion that optimization is achieved?
>
> kn4q-Q9: Mean time to optimize.
>
> Thank you for the question. Mean time to optimize is defined as the average amount of time required for the method to improve the best-so-far performance by at least 1%. This metric is intended to capture TusoAI’s ability to quickly produce meaningful gains early in the optimization process. We have updated Section 4 (lines 284-285) to clarify this.
>
> >line 385: "Results are shown in Table ??"
>
> kn4q-Q10: Line 385. We have corrected this typo.
>
> >Figure 3 B/C: What does the circles show? How should we read and interpret those figures?
>
> kn4q-Q11: Circles in Figure 3B/C.
>
> Thank you for the question. The circles in Figure 3B,C represent Venn diagrams. The overlapping regions (14 in B and 17 in C) show discoveries shared between the original method and the TusoAI-optimized version, while the non-overlapping regions (3 and 0 in B; 9 and 0 in C) show discoveries unique to one method.
>
> These figures should be interpreted as follows: in both analyses, scDRS (TusoAI) successfully recovered all discoveries made by the original scDRS, missing none, while also identifying additional biologically plausible associations. This indicates that scDRS (TusoAI) is strictly more powerful than the original implementation. We have updated the captions for Figures 3B,C to clarify the presentation and add interpretation.

---

> ### Author Response · Authors · 2025-11-23
> **Response to Reviewer kn4q (4/4)**
>
> >Figure 4: what does the distance threshold mean? How should read and interpret figure 4C?
>
> kn4q-Q12: Figure 4.
>
> Thank you for the question. We recognize that it is our responsibility to provide a clear exposition.
>
> Distance threshold in Figure 4A,B. The distance threshold refers to the genomic distance between the variant and the gene. The y-axis shows the enrichment of true variant-gene links (eQTLs in panel A; ABC links in panel B) among all inferred candidate links within that distance range. This visualization follows the same style used in the original pgBoost paper (Dorans 2025 Nat Genet). This analysis is important because identifying distal regulatory links—those located far from the gene body—is substantially more challenging and remains an open problem. We have updated the captions of Figures 4A and 4B as well as lines 488-489 to clarify this.
>
> Interpretation of Figure 4C. Figure 4C plots genomic coordinates on the x-axis and linking scores between the focal variant-gene pair (rs138917529 and GCK) on the y-axis for pgBoost (TusoAI) and baseline methods, with gene positions shown in the lower panel. The key result is that pgBoost (TusoAI) identifies the biologically plausible link between the Glucose/HbA1c-associated variant rs138917529 and GCK (linking score > 0.95), whereas pgBoost and other baselines assign lower scores (< 0.95) and therefore miss this link. This demonstrates pgBoost (TusoAI)’s improved ability to detect meaningful variant–gene connections. We have updated the caption of Figure 4C to clarify these points.

---

> > ### Comment · Reviewer_kn4q · 2025-11-27
> >
> > Thank you for the extensive rebuttal. The additions improve the clarity of the manuscript and resolve many of my specific technical questions. However, my central concern remains only partially addressed: the method is still highly heuristic. For example both Figure 1 and Algorithm 1 are highly complex and I belive at this can be further simplified to ease understanding of what the algorithm does at its core.
> >
> > I have two follow-up questions:
> > - It would help to include a minimal illustrative example on a simple synthetic problem to demonstrate the essential workflow without the full context as in the biological examples presented so far. This could make the operational logic of Figure 1 and Algorithm 1 easier to follow.
> > - Regarding the update rule: Have you considered a robustness or sensitivity analysis comparing the Bayesian update to alternative heuristic updates? More fundamentally, what is the core justification for choosing a Bayesian-style formulation in the first place?
> > - Can you make the Bayesian formulation more explicit in the manuscript besides the somewhat hidden formulation in Algoirthm 1?

---

> > > ### Author Response · Authors · 2025-12-01
> > > **Followup response (1/2)**
> > >
> > > Thank you for carefully reading our response. We sincerely appreciate your constructive feedback, which has helped us improve the paper substantially. We address the 3 remaining reviewer questions in detail below.
> > >
> > > >It would help to include a minimal illustrative example on a simple synthetic problem to demonstrate the essential workflow without the full context as in the biological examples presented so far. This could make the operational logic of Figure 1 and Algorithm 1 easier to follow.
> > >
> > > kn4q-R1-Q1: minimal illustrative example.
> > >
> > > Thank you for the suggestion. Following the reviewer’s request, we have added a simple example on a generic classification task to demonstrate the essential TusoAI workflow without requiring full biological context. Below we summarize the main behaviors of TusoAI in this example.
> > >
> > > Task description: “build a classification model”
> > > Dataset: “simulated X and Y”
> > > Evaluator: “accuracy on held out validation split”
> > >
> > > Line 1: TusoAI retrieves and summarizes 10 highly cited papers relevant to the task.
> > >
> > > Line 2: Build structured instructions. TusoAI constructs 10 categories of instructions (e.g., preprocessing, regularization, hyperparameter tuning; $\pi_c$= 0.6, 0.2, 0.2).
> > > - Example optimization instructions include “L1 loss
> > >  for category “regularization” and “log transformation” for category “preprocessing”.
> > >
> > > Line 3: Initialize solutions. TusoAI generates 5 initial solutions, such as logistic regression or random forest.
> > >
> > > Lines 4-15: Iterations. TusoAI selects instructions across categories to iteratively refine the solutions, allocating ~80% of iterations to optimization instructions (lines 8–10) and ~20% to diagnostic instructions (lines 11–13). During Bayesian updates (line 9), TusoAI initially assigns high utility to category preprocessing ($\pi_c=0.6$, time=0 hours), then shifts focus to category regularization ($\pi_c=0.5$, time=2 hours), and finally prioritizes category hyperparameter tuning ($\pi_c=0.5$, time=4 hours).
> > >
> > > Line 16: Final output. The final classification model code is output.
> > >
> > > We have updated Appendix V to add example instructions for a generic classification task.
> > >
> > > >Regarding the update rule: Have you considered a robustness or sensitivity analysis comparing the Bayesian update to alternative heuristic updates? More fundamentally, what is the core justification for choosing a Bayesian-style formulation in the first place?
> > >
> > > kn4q-R1-Q2: Bayesian update.
> > >
> > > Thank you for the comment. The reviewer raised 2 questions: (i) robustness or sensitivity analysis comparing the Bayesian update to alternative heuristic updates, and (ii) core justification for choosing a Bayesian-style formulation. We first summarize the Bayesian update and then address each question in turn.
> > >
> > > The Bayesian update: TusoAI groups instructions into categories (e.g., a “regularization” category containing “L1 loss, L2 loss” and a “preprocessing” category containing “normalization, log scaling”). Each category is assigned an initial category-specific probability (line 2). In each iteration, TusoAI samples a category (line 8), selects an instruction from the category, and—if the resulting solution improves performance—increases the probability of that category via a multiplicative update (line 9).
> > >
> > > (i) Robustness or sensitivity analysis comparing the Bayesian update to alternative heuristic updates.
> > >
> > > We compared the default Bayesian-style update against a simple heuristic baseline that samples categories uniformly throughout all iterations (TusoAI - No Bayesian) and an even simpler heuristic baseline that samples instructions uniformly throughout all iterations (TusoAI - No Categories). This isolates the contribution of the Bayesian update over the categorical structure. As shown in Table 3 of the original submission, the Bayesian update yields substantially higher performance (average score 0.66 vs. 0.55 and 0.57, resp.). We clarified this in Section 4.2 (lines 401–402).

---

> > > > ### Author Response · Authors · 2025-12-01
> > > > **Followup response (2/2)**
> > > >
> > > > (ii) Core justification for Bayesian-style formulation.
> > > >
> > > > We recognize that it is our responsibility to provide a clear exposition. We have expanded our previous response to kn4q-W3 to provide more clear justification. First, the design reflects how scientists typically develop computational methods: they reason at the level of broad categories—such as preprocessing, modeling, or regularization—and tend to revisit categories that previously led to improvements. This motivates assigning probabilities at the category level rather than the instruction level. Second, the multiplicative update provides a soft mechanism for incorporating evidence: a successful improvement slightly increases the likelihood that the same category will be sampled again. A more formal alternative would be a bandit-style update (e.g., Thompson Sampling with a Beta prior for each category’s success probability $\pi_c$), but we opted for the simpler constant–multiplicative rule because it is easy to interpret and performs well in practice. We have clarified this interpretation and discussed this limitation in Section 6 (line 535).
> > > >
> > > > >Can you make the Bayesian formulation more explicit in the manuscript besides the somewhat hidden formulation in Algoirthm 1?
> > > >
> > > > kn4q-R1-Q3: Making the Bayesian formulation more explicit.
> > > >
> > > > Thank you for the comment. We have updated Algorithm 1 line 9 to explicitly state the Bayesian update.

---

### Official Review · Reviewer_SvJ8 · 2025-10-29

**Soundness:** 3
**Presentation:** 2
**Contribution:** 2
**Rating:** 4
**Confidence:** 5

**Summary:**

The paper develops TusoAI, an agentic AI system that, given a scientific task description and an evaluation function, autonomously develops and optimizes computational methods for the application. TusoAI is tested across several single-cell tasks and some general deep learning tasks.

**Strengths:**

1) Mixing Bayesian optimization in the loop of an agentic AI system is interesting. I wish the paper had substantiated whether this is the first agent of this type or whether the idea is inspired by existing agents.

2) The results show that some of the methods constructed by TusoAI are novel rather than simple re-implementation of existing approaches or calls to standard packages.

3) The agentic system is tested in many scenarios, from final performance to behavior over time.

4) Single-cell results show promising biological results.

**Weaknesses:**

1) The writing can be improved. For example, the Abstract is excessively long. Instead of listing all results, please summarize the key points and condense the Abstract's opening.

2) While the evaluations involve tasks beyond a single cell, it seems like TusoAI's performance is more pronounced in the single cell case, which might suggest the paper could have made that the central focus rather than a general-purpose multi-agent AI.

3) The code is not available for a full review of the paper.

4) Narrow scope to problems that can be solved using off-the-shelf ML models and strategies rather than those that need in-depth model development (e.g., new deep learning models rather than say fine-tuning Restnet).

5) As discussed in the *LLM-based general learning agents* section, there are already agents developed for ML engineering. This work differentiates itself from those by focusing on scientific domains, but some baselines are still missing (R&D, DS, etc).

6) Limitation in scale: only 10 papers retrieved from Semantic Scholar.

7) I have some concerns about model selection and evaluation, which I have asked in the box below.

**Questions:**

1) The key motivation behind TusoAI is to develop new computational solutions to existing problems in science.  Do the results show a new computational method developed beyond your expectations? One that is far beyond the reach of human ML engineers?

2) Is there an underlying optimization problem that the Bayesian framework is solving?

3) What is the convergence criterion of the system? Figure 2A and 2B suggest that TusoAI never really converges, as code diversity oscillates and performance dips. Is this expected or desired? How did you pick when the algorithm should be terminated? Are there any prompting guardrails that avoid oscillations?

4)  Does increasing the number of papers retrieved from Semantics Scholar (from 10) improve the results?

5) Can you elaborate on how the paper ensures the model selection has been done in an apples-to-apples way across the datasets/baselines? Is model evaluation also handled by TusoAI or externally?

---

> ### Author Response · Authors · 2025-11-23
> **Response to Reviewer SvJ8 (1/4)**
>
> Thank you for noting that the design of TusoAI is interesting and that several of the methods it constructs are novel. We address each of the review comments in detail below. The manuscript has been revised accordingly, incorporating 10 new analyses and updates to 3 main figures, and 13 supplementary sections.
>
> We also wish to provide additional context regarding the case studies, which were not mentioned in the review but represent an essential part of our contribution. These case studies focus on two high-impact problems in human genetics: (i) detecting disease-critical cell populations and (ii) linking genetic variants to their target genes; both problems are central to understanding disease mechanisms and advancing therapeutic development (Lappalainen 2021 Science). These topics are the focus of many recent publications in top journals (e.g., Song 2025 Nature; Ma 2023 Cell Genom; Zhang 2022 Nat Genet; Dorans 2025 Nat Genet; Sakaue 2024 Nat Genet; Xie 2023 Cell Genom). In our study, we rigorously evaluated TusoAI-optimized versions of two state-of-the-art methods (scDRS, Zhang 2022 Nat Genet; pgBoost, Dorans 2025 Nat Genet) using the same simulations and benchmarking protocols as the original papers. The TusoAI-optimized methods consistently outperformed the originals, uncovered additional biologically plausible findings missed by prior methods, and produced ready-to-use improved implementations that would typically take researchers years to develop manually.
>
> The reviewer has raised 5 comments (Strengths comment 1 and Weaknesses comments 1-6) and 5 questions. We believe that we have completely addressed 10 out of these 12 points (S1,W1,W3,W5,W6,Q1,Q2,Q3,Q4,Q5) and substantially addressed the remaining points (W2,W4). Please let us know if there are any additional questions or if further clarification would be helpful. If the reviewer feels that their concerns have been adequately addressed, we would be grateful if they would consider updating their evaluation.
>
> > Mixing Bayesian optimization in the loop of an agentic AI system is interesting. I wish the paper had substantiated whether this is the first agent of this type or whether the idea is inspired by existing agents.
>
> SvJ8-S1: Novelty for mixing Bayesian optimization in the loop of an agentic AI system.
>
> Thank you for noting that this design is interesting. We confirm that our work is, to our knowledge, the first to integrate a knowledge-tree structure with category-level probabilistic weights updated through Bayesian optimization within an agentic AI framework. We have updated lines 142-143 to clarify this.
>
> > The writing can be improved. For example, the Abstract is excessively long. Instead of listing all results, please summarize the key points and condense the Abstract's opening.
>
> SvJ8-W1: Abstract is excessively long.
>
> Thank you for this helpful suggestion. We agree that a more concise summary improves readability. We have simplified the abstract by removing the specific results from case studies. We further simplified contributions 3 and 4 to remove specific results in the Introduction section (lines 74-79) in an effort to improve readability.
>
> >While the evaluations involve tasks beyond a single cell, it seems like TusoAI's performance is more pronounced in the single cell case, which might suggest the paper could have made that the central focus rather than a general-purpose multi-agent AI.
>
> SvJ8-W2: Focus of single-cell tasks.
>
> Thank you for this comment. The reviewer is correct that our benchmark experiments and case studies place stronger emphasis on single-cell and genetics applications (6 single-cell tasks, DeepSEA genetics task, and 2 genetics case studies). Nonetheless, TusoAI also achieved superior performance on the remaining 4 tasks outside genomics and genetics (Spherical, NinaPro, ECG, Satellite), with an average rank of 3.0 compared to 4.0 for the second-best method.
>
> We have thus updated Section 6 (lines 536-537) to clarify that while our evaluation primarily focuses on genomics and genetics applications—given their complexity and relevance—TusoAI is designed as a general framework applicable across diverse scientific domains.
>
> > The code is not available for a full review of the paper
>
> SvJ8-W3: Code availability.
>
> Thank you for this comment. The code was indeed included in the supplementary zip file submitted with the paper. We realize the reviewer may have been confused by the statement in the abstract, “Our code will be publicly available upon publication.” We apologize for this misunderstanding. To further improve clarity and accessibility, we have also uploaded a cleaner and better-organized version of the code in the supplementary zip file.

---

> ### Author Response · Authors · 2025-11-23
> **Response to Reviewer SvJ8 (2/4)**
>
> > Narrow scope to problems that can be solved using off-the-shelf ML models and strategies rather than those that need in-depth model development (e.g., new deep learning models rather than say fine-tuning Restnet).
>
> SvJ8-W4: Scope of problems and use of off-the-shelf ML models.
>
> Thank you for this comment. The reviewer raised a concern that the empirical studies may be limited to problems solvable with standard ML models. We would like to clarify that this is not the case. Specifically, 5 of the 11 benchmark tasks and both case studies require customized algorithmic design beyond off-the-shelf models. For example, the scDRS case study involves designing a new disease-association score function, the pgBoost case study requires constructing novel genomic features informative of variant–gene links, and all single-cell tasks but label projection fall outside the standard train/test ML setup, as they are inherently algorithmic without ground truth data to learn a model from and have non-standard evaluation procedures.
>
> Importantly, across all 13 tasks and case studies—including those formulated in standard ML formats—TusoAI generated original solutions, such as extending beyond the predefined neural architecture search (NAS) space for the 5 NAS tasks (Appendix H). This demonstrates that TusoAI can develop novel methods rather than merely fine-tuning existing models.
>
> We have thus updated Section 4 (lines 301-302) to clarify these points.
> > As discussed in the LLM-based general learning agents section, there are already agents developed for ML engineering. This work differentiates itself from those by focusing on scientific domains, but some baselines are still missing (R&D, DS, etc).
>
> SvJ8-W5: ML engineering agent baselines.
>
> Thank you for this comment. We appreciate the reviewer’s suggestion and clarify that we did not include certain ML-engineering agent baselines (e.g., R&D Agent, DS-Agent) for two main reasons: (i) Both R&D Agent and DS-Agent rely on hard-coded templates that assume a conventional ML setup (e.g., data splitting and standard evaluation metrics), which cannot be easily adapted to non-ML or domain-specific scientific tasks (7 of our 13 total tasks);  we also attempted to adapt R&D Agent, DS-Agent, AutoML-Agent, FunSearch, and Dolphin, but found that they could not be effectively applied to non-standard ML settings. (ii) We instead included baselines that demonstrated both applicability and stronger performance in prior work, such as AIDE (Aygun et al[2], BioML Bench[1]), providing a more meaningful comparison.
>
> We have updated Section 4 (lines 301-302) to clarify these points.
>
> [1] Henry E. Miller, Matthew Greenig, Benjamin Tenmann, and Bo Wang. Bioml-bench: Evaluation of ai agents for end-to-end biomedical ml. bioRxiv, 2025.
>
> [2] Eser Aygun, Anastasiya Belyaeva, Gheorghe Comanici, Marc Coram, Hao Cui, Jake Garrison, Renee Johnston Anton Kast, Cory Y McLean, Peter Norgaard, ZahraShamsi, et al. An ai system to help ocientists write expert-level empirical software. arXiv preprint
>
> >Limitation in scale: only 10 papers retrieved from Semantic Scholar.
>
> SvJ8-W6: Number of papers from Semantic Scholar.
>
> Thank you for this comment. To assess the effect of the number of retrieved papers on performance, we conducted additional analyses varying this number from 0 to 50 (0 and 50, instead of 10). We found that using 10 papers yielded the best overall performance (outperforming 0 and 50 in 4 out of 5 tasks), likely because retrieving too few papers (e.g., 0) omits useful domain knowledge, whereas too many (e.g., 50) introduce irrelevant information that can hinder optimization. We have updated Section 4.2 (lines 401-403) and added Appendix O to report these findings. Here is the median across 5 replicates.
>
> | Papers     | Denoise | SVG  | Decomposition | ECG  | Satellite |
> |------------|---------|------|----------------|------|-----------|
> | 0 papers   | 0.21    | 0.79 | 0.51           | 0.60 | 0.82      |
> | 10 papers (default)  | 0.35    | 0.82 | 0.55           | 0.63 | 0.89      |
> | 50 papers  | 0.21    | 0.73 | 0.49           | 0.66 | 0.88      |

---

> ### Author Response · Authors · 2025-11-23
> **Response to Reviewer SvJ8 (3/4)**
>
> >The key motivation behind TusoAI is to develop new computational solutions to existing problems in science. Do the results show a new computational method developed beyond your expectations? One that is far beyond the reach of human ML engineers?
>
> SvJ8-Q1: Novelty of computational methods developed by TusoAI.
>
> Thank you for this insightful question. We agree that it is important to characterize the properties and novelty of the computational methods developed by TusoAI. To assess this, we manually examined the methods generated across the 4 examples discussed in Section 4.1 and the 2 case studies.
> We found that TusoAI often produces computational methods that are substantially different from existing approaches, primarily by creatively combining and repurposing existing techniques rather than inventing entirely new ones, perhaps because TusoAI relies on domain knowledge from scientific papers. This reflects a pattern similar to how human ML engineers innovate: by synthesizing established ideas into new, effective solutions. We have added a detailed discussion of this observation, along with representative examples in Appendix H, referenced in line 318. For example:
>
> (i) In single-cell denoise, TusoAI designed a non-negative matrix factorization (NMF) approach that models dropout rates, Poisson noise, and performs iterative refinement. The other NMF approaches in the literature ALRA (Linderman et al. Nat Commun, 2022) and DenoiseIt (Jeon et al BMC Bioinfo 2024) do not model dropout rates or Poisson noise.
>
> (ii) In Satellite, TusoAI combined preprocessing, training procedures, loss functions, and ensembling techniques to build the top-performing model. First, this is distinct from the expert model, which is a linear classifier (Dempster et al. Data Mining and Knowledge Discovery, 2020). Second, these kinds of pipeline-level decisions lie outside the search space of the NAS baselines used in NASBench-360.
>
> >Is there an underlying optimization problem that the Bayesian framework is solving?
>
> SvJ8-Q2: Underlying optimization problem for the Bayesian framework.
>
> Thank you for this insightful question. The Bayesian update step (lines 2,8,9 in Algorithm 1 in the original submission) serves as a heuristic that mimics how a scientist might prioritize among several categories of actions (e.g., refining preprocessing, revisiting modeling assumptions, or modifying optimization strategies). The update increases the likelihood of selecting categories that have previously led to performance improvements, thereby implementing an implicit optimization over expected performance gain. We acknowledge that a fully formalized optimization formulation could yield a more principled Bayesian update procedure. For instance, one could adopt a Thompson Sampling-style formulation in which each category is associated with a reward probability pc governed by a Beta prior, and at each iteration the category with the largest sampled value is selected. We have clarified this interpretation and discussed this limitation in Section 3 (lines 258-259) and Section 6 (line 535).
>
> >What is the convergence criterion of the system? Figure 2A and 2B suggest that TusoAI never really converges, as code diversity oscillates and performance dips. Is this expected or desired? How did you pick when the algorithm should be terminated? Are there any prompting guardrails that avoid oscillations?
>
> SvJ8-Q3: Convergence criterion.
>
> Thank you for this question. TusoAI is designed to terminate after a fixed 8-hour optimization window rather than based on an explicit convergence criterion. We note that Figure 2B may be somewhat misleading: TusoAI always optimizes the best solution obtained up to a given time (dark blue line in Figure 2B), not the proposed code in the current iteration (light blue line), and thus oscillations do not happen. We have updated Figure 2B to improve clarity.
>
> >Does increasing the number of papers retrieved from Semantics Scholar (from 10) improve the results?
>
> SvJ8-Q4: Number of papers from Semantic Scholar.
>
> Please see response to SvJ8-W6 above.

---

> ### Author Response · Authors · 2025-11-23
> **Response to Reviewer SvJ8 (4/4)**
>
> >Can you elaborate on how the paper ensures the model selection has been done in an apples-to-apples way across the datasets/baselines? Is model evaluation also handled by TusoAI or externally?
>
> SvJ8-Q5: Model evaluation and fairness across baselines.
>
> Thank you for this question. We agree that it is important to ensure an apples-to-apples comparison across datasets and baselines. Model evaluation is handled externally and conducted identically for all methods. Specifically, TusoAI and AIDE were each run for the same duration on the same hardware, under identical conditions, including temperature, conda environment, LLM, task description, dataset, and evaluation function. Biomni and ChatGPT-Agent do not support fixed time budgets; therefore, we followed the recommendation of the Biomni authors and allowed these agents to run until self-termination. All methods used the same validation datasets for optimization and were evaluated on the same held-out test datasets (by an external function) to ensure fair comparison. Details are provided in Appendix D, E. We have also updated Section 4 (lines 288-289) in an effort to improve clarity.

---

> > ### Comment · Reviewer_SvJ8 · 2025-11-25
> > **Thanks for the notes**
> >
> > I thank the authors for their detailed response. Many of my questions, such as cross-validation, code availability, and improvement in organizing the papers, have been resolved. I am increasing my score to 6. I cannot be more positive due to some remaining issues, for example, performance that does not scale (and deteriorates) with the # of papers (maybe an improved knowledge graph would help in this case) and a lack of a concrete optimization objective (heuristic nature of the work).

---

> > > ### Author Response · Authors · 2025-12-01
> > > **Thank you for your response**
> > >
> > > Svj8-R1
> > >
> > > Thank you for carefully reading our response and updating your evaluation. We sincerely appreciate your constructive feedback, which has helped us improve the paper substantially. The reviewer has raised 2 additional concerns: (i) performance not scaling (and deteriorating) with the # of papers; and (ii) lack of a concrete optimization objective. We address each in turn.
> > >
> > > (i) Performance not scaling (and deteriorating) with the # of papers.
> > >
> > > The reviewer is correct that in our secondary analysis (detailed in response to SvJ8-W6), TusoAI’s performance did not consistently improve with more papers. Upon further investigation, we found that this was largely due to decreasing paper relevance as the number of included papers increased from 10 to 50.
> > >
> > > To address this, we implemented an (optional) simple but effective filtering step during TusoAI’s paper-summary stage. After generating the summaries, we query an LLM to assess whether each summary contains task-relevant information and remove those that do not. We then re-ran the analysis with this filtering in place. The results show that filtering substantially improves performance when using 50 papers and does not affect performance under the default setting (10 papers); see the table below. Thus, this filtering step resolves the scalability issue observed when increasing the number of input papers. We have updated Appendix O to report these findings. Here’s the median testing scores across 5 replicates showing improved performance with filtering.
> > >
> > > | Papers     | Denoise | SVG  | Decomposition | ECG  | Satellite |
> > > |------------|---------|------|----------------|------|-----------|
> > > | 0 papers   | 0.21    | 0.79 | 0.51           | 0.60 | 0.82      |
> > > | 10 papers (default)  | 0.35    | 0.82 | 0.55           | 0.63 | 0.89      |
> > > | 10 papers (filtered)  | 0.34    | 0.83 | 0.58           | 0.65 | 0.89      |
> > > | 50 papers  | 0.21    | 0.73 | 0.49           | 0.66 | 0.88      |
> > > | 50 papers (filtered)  | 0.36    | 0.86 | 0.65           | 0.66 | 0.89      |
> > >
> > >
> > > (ii) Lack of a concrete optimization objective.
> > >
> > > We believe the reviewer is referring to our earlier response to question SvJ8-Q2 (Underlying optimization problem for the Bayesian framework), where we noted that the Bayesian framework “serves as a heuristic.” To clarify: the paper does have a concrete, well-defined optimization objective ($\arg\max_{s \in S_{\text{full}}} h(s)$); line 159 of the original submission), and the Bayesian framework is a heuristic approach to aid in this optimization problem.
> > >
> > > This design is consistent with recent agentic-method literature at top ML venues, where a principled optimization objective is paired with heuristic search procedures. For example:
> > > - AIDE uses the same objective but selects candidate solutions in its tree search via LLM prompting and relies on a human-designed set of mutation operations (Draft / Debug / Improve).
> > > - Aygün et al. uses the same objective and employs a generic “Generate-And-Execute” heuristic for code mutation.
> > > - R&D-Agent is fully heuristic in its workflow.
> > > - DS-Agent provides a probabilistic formulation but still relies on heuristic operations such as Retrieve / Reuse / Execute.
> > > - This is also similar to many classical AutoML approaches, which repeatedly update, train, and validate model performance over many iterations, such as Auto-sklearn or Autogluon.
> > >
> > > We agree with the reviewer that having a clear and concrete optimization objective is important for principled method development. We have updated Section 2 and 6 (lines 152-154, 535) to make the objective explicit and to discuss potential future directions for developing more principled search strategies.
> > >
> > > Again, we thank the reviewer for the constructive feedback, which has meaningfully improved the clarity and rigor of the paper. We are happy to address any further questions.

---

### Official Review · Reviewer_aNYX · 2025-11-02

**Soundness:** 3
**Presentation:** 2
**Contribution:** 3
**Rating:** 4
**Confidence:** 4

**Summary:**

This paper introduces TusoAI, an agentic AI framework that automatically develops and optimizes computational methods for scientific tasks. Given task descriptions and evaluation functions, it can generate new algorithms rather than just run existing analysis pipelines. The main idea is to structure domain knowledge into a “knowledge tree,” combine it with Bayesian-guided iterative optimization and diagnostic feedback, and iteratively generate, implement, and evaluate candidate solutions. On 11 benchmarks, TusoAI achieved a higher average rank compared to expert-designed methods and existing AI agents.

**Strengths:**

- Clear motivation and problem framing. The paper focuses on creating methods rather than just executing pipelines. It specifies the objective and agent loop, including the task description and the evaluator $h(\cdot)$, with cold/warm start.
- Breadth of benchmarks with application case study. Empirical scope spans 11 tasks across two domains (six single-cell, five scientific deep learning).
- The warm-start improvements on scDRS and pgBoost are interesting. It is interesting to consider concrete empirical deltas and biological findings (e.g., new T-cell disease associations; rs138917529→GCK link).

**Weaknesses:**

- Too many symbols in Algorithm 1. If the authors want to introduce these annotations, they should include an appendix table.
- Baseline coverage is insufficient, mainly including general-purpose agents (AIDE, Biomni, ChatGPT-Agent) and simple expert or AutoML models. It lacks domain-specific, publication-level baselines for each scientific task. Including parts of the leaderboard is recommended.
- Ablation and diagnosis analyses lack depth. The ablation study in Table 3 mixes tasks from unrelated domains. It omits per-task variance or statistical testing, making it impossible to isolate the contribution of each component (knowledge tree, Bayesian updates, diagnosis).
- No detailed analysis of failure cases or performance drops is provided, so readers cannot understand when the system fails or regresses.
- Case studies serve as illustrations but lack control. The genetic applications (scDRS, pgBoost) show improvements, yet they lack independent validation or error analysis.

**Questions:**

- Can the authors clearly specify what metric is used in each column of Tables 1 and 2, and how the “Avg” and “Avg rank” values are computed across heterogeneous tasks? If normalization was applied (e.g., 1 – MSE, scaled decomposition R²), please make this explicit in the main text or table captions.
- How do the qualitative examples, such as NMF with dropout/Poisson modeling and Satellite ensembling, support the narrative of the methods being "distinct" or "custom"?
- Is using text similarity (Fig. 2A) a sufficient proxy for diversity, or should a more principled novelty metric be employed in the evaluation?
- This work’s broad, application-driven scope and descriptive evaluation would be better suited for an applied-science journal rather than a methodology-focused venue like ICLR.

---

> ### Author Response · Authors · 2025-11-23
> **Response to Reviewer aNYX (1/3)**
>
> Thank you for the thoughtful summary of our work and for noting that the warm-start improvements on scDRS and pgBoost are interesting. We summarize the review comments and provide point-by-point responses below. The manuscript has been revised accordingly, incorporating 10 new analyses and updates to 3 main figures, and 13 supplementary sections.
>
> The reviewer has raised 5 comments/weaknesses and 4 questions. We believe that we have completely addressed 7 out of these 9 points (W1,W2,W3,Q1,Q2,Q3,Q4) and substantially addressed the remaining points (W4,W5). Please let us know if there are any additional questions or if further clarification would be helpful. If the reviewer feels that their concerns have been adequately addressed, we would be grateful if they would consider updating their evaluation.
>
> > Too many symbols in Algorithm 1. If the authors want to introduce these annotations, they should include an appendix table.
>
> aNYX-W1: Symbols in Algorithm 1.
>
> We recognize that it is our responsibility to provide a clear exposition. We have added a table in the appendix summarizing all symbols and annotations used in Algorithm 1 to improve readability (lines 173-174).
>
> > Baseline coverage is insufficient, mainly including general-purpose agents (AIDE, Biomni, ChatGPT-Agent) and simple expert or AutoML models. It lacks domain-specific, publication-level baselines for each scientific task. Including parts of the leaderboard is recommended.
>
> aNYX-W2: Baseline coverage.
>
> Thank you for raising this point. The reviewer noted the lack of domain-specific, publication-level baselines for each scientific task. We would like to clarify that such baselines were indeed included for all 11 tasks evaluated in the paper, under the label “Expert”, which represents the best domain-specific, publication-level method from the corresponding leaderboard in OpenProblems and NASBench-360. For example, “Expert” corresponds to MAGIC (van Dijk 2018 Cell) for Denoise, SPARK-X (Zhu 2021 Genome Biol) for SVG, and the DeepSEA model (Zhou 2015 Nat Methods) for DeepSea. We recognize that it is our responsibility to provide a clear exposition. We updated line 293 and Appendix F to report the full list of published “Expert” methods in an effort to improve clarity.
>
> > Ablation and diagnosis analyses lack depth. The ablation study in Table 3 mixes tasks from unrelated domains. It omits per-task variance or statistical testing, making it impossible to isolate the contribution of each component (knowledge tree, Bayesian updates, diagnosis).
>
> aNYX-W3: Ablation and diagnosis analyses.
>
> Thank you for the comment. The reviewer is correct that providing per-task results for the ablation study can help assess task-specific variance and provide further insight into the contribution of different components to performance across tasks. We therefore added four additional replicates for each task in all ablation studies and found similar patterns in performance, specifically that removing each component lowered overall performance, which is consistent with the original results. We have updated lines 401-403 to report these findings.
>
> > No detailed analysis of failure cases or performance drops is provided, so readers cannot understand when the system fails or regresses.
>
> aNYX-W4: Analysis of failure cases or performance drops.
>
> Thank you for the comment. We’ve added a Discussion section addressing limitations and future work. In short, TusoAI might perform poorly in the following situations:
> If a representative validation experiment takes exceedingly long, TusoAI won’t be able to complete iterations very quickly.
> When optimizing an existing scientific method, if the actual method is very scattered around in different functions and any single function has only a small amount of the method, TusoAI might struggle to discover optimizations.
>
> For performance drop, by definition, performance cannot decrease, as TusoAI selects the top performing solutions after each round of optimizations. We have updated Section 6 (lines 530-533) to discuss these points.

---

> ### Author Response · Authors · 2025-11-23
> **Response to Reviewer aNYX (2/3)**
>
> > Case studies serve as illustrations but lack control. The genetic applications (scDRS, pgBoost) show improvements, yet they lack independent validation or error analysis.
>
> aNYX-W5: Validation and error analysis for case studies.
>
> Thank you for this comment. The reviewer raises an important point regarding independent validation and error analysis in the case studies. We would like to clarify that both TusoAI-optimized methods were applied to multiple datasets from the original studies, and consistent improvements were observed across all of them, as summarized below. We therefore believe our level of validation is comparable to that of prior genetic studies, which typically employ rigorous multi-dataset evaluations and are more extensive than what is typical for ML case studies.
>
> First, the TusoAI-optimized methods were validated using the same experimental setups as in the original publications, ensuring comparable rigor. Specifically, we assessed the calibration and power of TusoAI+scDRS using the same null and causal simulations as well as real-data benchmarks as Zhang 2022 Nat Genet, and evaluated the accuracy of TusoAI+pgBoost using the same orthogonal variant-gene link comparisons as Dorans 2025 Nat Genet. TusoAI+scDRS was applied to 2 sets of simulations (null+causal) and 1 ground truth trait-cell type dataset, and 1 additional scRNA-seq dataset with literature validation of findings, and TusoAI+pgBoost was applied across four single-cell multiome datasets and 49 tissues used in the original paper and validated against 2 independent variant-gene link datasets (eQTL+ABC).
>
> Second, the additional discoveries uniquely identified by the TusoAI-optimized methods were cross-referenced against existing literature to evaluate biological plausibility. This level of validation aligns with the standards of prior computational genomics studies, where in silico validation is generally accepted (Zhang 2022 Nat Genet; Dorans 2025 Nat Genet).
>
> We have updated lines 438-440 to clarify these points. We would also welcome any specific validation or error analyses the reviewer recommends and would be happy to conduct additional experiments to address them.
>
> > Can the authors clearly specify what metric is used in each column of Tables 1 and 2, and how the “Avg” and “Avg rank” values are computed across heterogeneous tasks? If normalization was applied (e.g., 1 – MSE, scaled decomposition R²), please make this explicit in the main text or table captions.
>
> aNYX-Q1: Evaluation metrics.
>
> Thank you for the comment. The metrics used for each column in Tables 1 and 2 are listed in Appendices D and E. “Avg” denotes the average test score across all tasks, and “Avg Rank” represents the mean rank of each method across tasks relative to the other evaluated methods; Denoise reports normalized 1-MSE such that 0 is the same as no denoising, Decomp is normalized so that random assignments are 0, and NASBENCH-360 results are normalized to be between 0 and 1 and such that higher is better. We have updated lines 288-289 and Table 2’s caption to clarify these.
>
> >How do the qualitative examples, such as NMF with dropout/Poisson modeling and Satellite ensembling, support the narrative of the methods being "distinct" or "custom"?
>
> aNYX-Q2: Qualitative examples.
>
> Thank you for the question. We agree that it is important to clarify why the TusoAI-optimized methods in the four qualitative examples are distinct and novel. For example:
>
> (i) In single-cell denoise, TusoAI designed a non-negative matrix factorization (NMF) approach that models dropout rates, Poisson noise, and performs iterative refinement. The other NMF approaches in the literature ALRA (Linderman et al. Nat Commun 2022) and DenoiseIt (Jeon et al BMC Bioinfo 2024) do not model dropout rates or Poisson noise.
>
> (ii) In Satellite, TusoAI combined preprocessing, training procedures, loss functions, and ensembling techniques to build the top-performing model. First, this is distinct from the expert model, which is a linear classifier (Dempster et al. Data Mining and Knowledge Discovery 2020). Second, these kinds of pipeline-level decisions lie outside the search space of the NAS baselines used in NASBench-360.
>
> We have updated lines 318 and added a full literature comparison for the 4 examples in the paper to Appendix H to clarify these points.

---

> ### Author Response · Authors · 2025-11-23
> **Response to Reviewer aNYX (3/3)**
>
> > Is using text similarity (Fig. 2A) a sufficient proxy for diversity, or should a more principled novelty metric be employed in the evaluation?
>
> aNYX-Q3: Text similarity.
>
> We agree that it is important to assess whether the text similarity measure used in Fig. 2A is a sufficient proxy for diversity. We believe text similarity is a reasonable approximation, as it has been widely adopted in related studies, such as Aygun et al [1]. Moreover, we further examined the optimization trajectories of AIDE (Appendix K) and found that noticeable performance changes ceased in single-cell tasks around the two-hour mark, the same point at which diversity (as measured by text similarity) collapsed in Figure 2A. This suggests that text similarity effectively captures the convergence and redundancy of generated methods. Nonetheless, we acknowledge that more principled novelty metrics could provide a more rigorous assessment of diversity. We have updated Section 6 (line 535) to discuss these points.
>
> [1] Eser Aygun, Anastasiya Belyaeva, Gheorghe Comanici, Marc Coram, Hao Cui, Jake Garrison, Renee Johnston Anton Kast, Cory Y McLean, Peter Norgaard, ZahraShamsi, et al. An ai system to help ocientists write expert-level empirical software. arXiv preprint
>
> > This work’s broad, application-driven scope and descriptive evaluation would be better suited for an applied-science journal rather than a methodology-focused venue like ICLR.
>
> aNYX-Q4: Suitability for ICLR.
>
> Thank you for this comment. We agree that our work has a broad, application-driven scope, but we emphasize that its primary contribution lies in methodological innovation. TusoAI introduces a general framework for agentic optimization, integrating LLM reasoning, knowledge-tree organization, and iterative search—validated through extensive quantitative evaluations across 11 benchmark tasks and two case studies.
>
> This focus on new methodology with broad applicability aligns well with ICLR’s scope, which includes “applications to the physical sciences (physics, chemistry, biology, etc.)” when they demonstrate methodological advances. Similar agentic or automated research systems—such as DS-Agent (ICML), AutoML-Agent (ICML), and AIDE (ICLR)—have been presented at top ML conferences. Moreover, ICLR has recently featured works that bridge scientific discovery and methodology (e.g., BEND, CellPLM, CF-Gen).
>
> We therefore believe TusoAI fits ICLR’s methodological focus while illustrating its utility in real scientific domains.

---

> > ### Comment · Reviewer_aNYX · 2025-11-24
> > **Thank you for your response**
> >
> > Most concerns have been addressed, and I’ve increased my score from 4 to 8. I apologize if my previous comment about the journal fit caused any confusion; I did not mean the work is inappropriate for ICLR. The paper is well done; I merely suggested that future extensions to other areas of computational science could expand its influence beyond ML and reach a broader audience. I have no additional questions. The author has put significant effort into their response. Cheers!

---

> ### Author Response · Authors · 2025-11-24
>
> Thank you for carefully reading our response and updating your evaluation. We sincerely appreciate your constructive feedback, which has helped us strengthen the paper.

---

### Official Review · Reviewer_ff1y · 2025-11-02

**Soundness:** 2
**Presentation:** 3
**Contribution:** 3
**Rating:** 6
**Confidence:** 4

**Summary:**

This paper introduces TusoAI, an LLM-based agentic system for automatically developing and optimizing computational methods for scientific applications. The system structures domain knowledge into a hierarchical knowledge tree, uses Bayesian updates to sample optimization strategies across categories, and incorporates diagnostic feedback during iterative refinement. The authors evaluate TusoAI on 11 scientific benchmarks (6 single-cell analysis tasks and 5 deep learning tasks), showing consistent improvements over baselines including AIDE, Biomni, and expert-designed methods. Two case studies in genetics demonstrate practical utility by improving existing methods (scDRS and pgBoost) and uncovering novel biological insights.

**Strengths:**

1. The paper evaluates TusoAI on 11 tasks spanning single-cell analysis and scientific deep learning, demonstrating broad applicability. The inclusion of both cold-start and warm-start settings, along with real-world case studies in genetics, strengthens the practical relevance.

2. The knowledge tree representation provides a structured way to organize domain knowledge, and the Bayesian update mechanism offers a principled approach to balancing exploration and exploitation. The diagnostic-based optimization is a nice addition that grounds the system in empirical data patterns.

3. TusoAI consistently outperforms strong baselines including AIDE and domain-specific expert methods. The ablation studies effectively demonstrate that each component (categories, Bayesian updates, diagnostics, domain knowledge) contributes meaningfully to overall performance.

**Weaknesses:**

1. The core algorithmic components are largely existing techniques (LLM agents, tree-based search, iterative optimization), and the main contribution is their combination for scientific method development. The concurrent work by Aygun et al. (2025) appears to address very similar problems, but the paper dismisses direct comparison due to unavailable code without sufficiently differentiating the approaches conceptually.

2. The 8-hour optimization budget is relatively short and may favor methods that converge quickly over those requiring longer refinement. The diversity analysis (Figure 2A) shows TusoAI explores more than AIDE, but it's unclear whether this translates to better generalization. The selection strategy (shortest code within 0.1% of best performance) seems ad-hoc and could inadvertently favor simpler but less robust solutions.

3. The paper doesn't discuss when TusoAI fails or performs poorly, nor does it address practical concerns like how to set appropriate evaluation functions for novel scientific problems, how sensitive the system is to task description quality, or how to validate generated methods when ground truth is unavailable. The computational cost analysis is limited to one brief mention ($0.37-$0.41 for case studies).

4. The case studies claim to discover "9 new associations" and "7 previously unreported links," but these are computational predictions that would require experimental validation to be considered true biological discoveries. The statistical methodology for declaring associations as "novel" versus "missed by previous methods" is not clearly described.

5. While the authors promise to release code upon publication, key details are missing: which specific LLM API versions were used, how were papers selected from Semantic Scholar (just citation count?), what are the exact prompts for different agents, and how stable are results across different random seeds beyond the reported 95% CIs?

**Questions:**

1. How does TusoAI handle the cold start problem for truly novel scientific domains? The knowledge tree construction relies on retrieving papers from Semantic Scholar based on task descriptions. For emerging research areas with limited literature, how does the system bootstrap domain knowledge? Have you tested TusoAI on tasks where relevant literature is scarce or the task description is deliberately vague?

2. Can you provide more details on the comparison with Aygun et al. (2025) and clarify the key technical differences? Beyond the unavailability of their code, what are the conceptual and methodological differences between your approach and theirs? This seems critical for establishing the novelty of your work.

---

> ### Author Response · Authors · 2025-11-23
> **Response to Reviewer ff1y (1/5)**
>
> Thank you for the accurate summary of our method and for highlighting its strong practical utility in uncovering novel biological insights. We summarize each review comment and provide a point-by-point response below. The manuscript has been revised accordingly, incorporating 10 new analyses and updates to 3 main figures, and 13 supplementary sections.
>
> The reviewer has raised 5 comments/weaknesses and 2 questions. We believe that we have completely addressed 4 out of these 7 points (W3,W4,W5,Q1) and substantially addressed the remaining points (W1,W2,Q2). Please let us know if there are any additional questions or if further clarification would be helpful. If the reviewer feels that their concerns have been adequately addressed, we would be grateful if they would consider updating their evaluation.
>
>
> > The core algorithmic components are largely existing techniques (LLM agents, tree-based search, iterative optimization), and the main contribution is their combination for scientific method development. The concurrent work by Aygun et al. (2025) appears to address very similar problems, but the paper dismisses direct comparison due to unavailable code without sufficiently differentiating the approaches conceptually.
>
> ff1y-W1: Comparison with Aygun et al. (2025).
>
> Thank you for raising this comment. The reviewer is correct that the concurrent work by Aygun et al. (2025) appears to address very similar problems. Thus, it is warranted to provide a conceptual comparison despite the code being unavailable. Both methods integrate LLMs with domain knowledge to implement an executable solution that maximizes a predefined score function for a given task. However, the two approaches differ substantially in how they incorporate domain knowledge and perform optimization.
>
> (1) Domain knowledge integration. TusoAI is designed to integrate multiple research ideas simultaneously into a single unified solution, whereas the Aygun method focuses on integrating one idea for each solution and later combines selected solutions (e.g., pairwise combinations of top methods). Simultaneous integration of many research ideas, as done by TusoAI, is more challenging and central to our contribution. To achieve this, TusoAI organizes domain knowledge as a knowledge tree, where each iteration incorporates a separate research idea. Consequently, the final solution synthesizes multiple ideas. Our ablation results confirm that this structure improves performance (average performance of 0.66 for TusoAI vs. 0.57 for “No categories”; Table 3 in the original submission).
>
> (2) Optimization. While both methods use a tree search framework, TusoAI employs a more dynamic optimization strategy. Specifically, the Aygun method uses a generic “GenerateAndExecute” procedure same for all iterations, whereas TusoAI uses a different instruction for each iteration (lines 7-10 of Algorithm 1), either (i) an adaptively optimization instruction or (ii) a model diagnostic instruction (lines 12-13), allowing TusoAI to dynamically optimization the solutions. Ablation experiments show that both mechanisms independently enhance performance (Table 3, average performance of 0.66 for TusoAI vs. 0.55 for (i) “No Bayesian” and 0.64 for (ii) “No diagnosis”).
>
> We have updated Section 1.1 (lines 127-130) to clarify these distinctions.

---

> ### Author Response · Authors · 2025-11-23
> **Response to Reviewer ff1y (2/5)**
>
> > The 8-hour optimization budget is relatively short and may favor methods that converge quickly over those requiring longer refinement. The diversity analysis (Figure 2A) shows TusoAI explores more than AIDE, but it's unclear whether this translates to better generalization. The selection strategy (shortest code within 0.1% of best performance) seems ad-hoc and could inadvertently favor simpler but less robust solutions.
>
> Thank you for the comment. The reviewer has raised 3 related points (i) the 8-hour optimization budget may favor simpler methods, (ii) more diverse solutions may not lead to better generalization, and (iii) the selection strategy (shortest code within 0.1% of best performance) seems ad-hoc. We address each of these points in turn.
>
> (i) The 8-hour optimization budget may favor simpler methods.
> Thank you for the comment. The 8-hour optimization budget follows common practice in recent work (e.g., ~7 hours for single-cell tasks in Aygun et al., ~8 hours in BioML Bench [1]). We also verified that this choice yields a fair comparison: AIDE used the same 8-hour budget but typically converged within 2 hours, while Biomni and ChatGPT-Agent were not given fixed budgets, as recommended by the Biomni authors. In addition, generalization was evaluated using disjoint training and evaluation datasets, further supporting the robustness of the comparison. We acknowledge effectively utilizing longer budgets is an important future direction to consider. We have updated Lines 279-280, 289-290, and 368 and added Appendices J,K to clarify these points.
>
> (ii) More diverse solutions may not lead to better generalization.
> Thank you for this insightful comment. We agree that the connection between diversity and performance is important to substantiate. First, we performed a new analysis examining the optimization trajectories of all 13 tasks (Appendix J, K). We found that performance plateaus when diversity collapses (correlation between optimization plateau and diversity collapse Pearson’s R = 0.51, Appendix M). Second, the importance of maintaining diversity is supported by several related works on program synthesis and agentic optimization. For example, FunSearch [3] explicitly maintains a pool of diverse programs to avoid local optima through island-based evolution; MLE-STAR [4] incorporates diversity-promoting feedback in the refinement loop; and Aygun et al. [1] emphasize that generating diverse candidate solutions is essential for discovering high-performing ones. These works collectively motivate why diversity is a principled design choice rather than an arbitrary heuristic. We have updated Appendix J, K, M and Lines 359-360, Line 368 to incorporate these new analyses and to clearly articulate the motivation for encouraging diversity in TusoAI.
>
> (iii) The selection strategy (shortest code within 0.1% of best performance) seems ad-hoc.
> We agree that the selection strategy is heuristic. It was designed to balance model performance with simplicity and has empirically produced stable generalization across tasks. We have acknowledged this limitation in the Discussion (Section 6, line 535) and highlighted the development of a more principled selection criterion as an important direction for future work.
>
> [1] Henry E. Miller, Matthew Greenig, Benjamin Tenmann, and Bo Wang. Bioml-bench: Evaluation of ai agents for end-to-end biomedical ml. bioRxiv, 2025.
>
> [2] Eser Aygun, Anastasiya Belyaeva, Gheorghe Comanici, Marc Coram, Hao Cui, Jake Garrison, Renee Johnston Anton Kast, Cory Y McLean, Peter Norgaard, ZahraShamsi, et al. An ai system to help ocientists write expert-level empirical software. arXiv preprint
>
> [3] Bernardino Romera-Paredes, Mohammadamin Barekatain, Alexander Novikov, Matej Balog, M Pawan Kumar, Emilien Dupont, Francisco JR Ruiz, Jordan Ellenberg, Pengming Wang, Omar Fawzi, et al. Mathematical discoveries from program search with large language models. Nature
>
> [4] Jaehyun Nam, Jinsung Yoon, Jiefeng Chen, Jinwoo Shin, Sercan O Arık, and Tomas Pfister. Mle- ¨star: Machine learning engineering agent via search and targeted refinement. arXiv preprint

---

> ### Author Response · Authors · 2025-11-23
> **Response to Reviewer ff1y (3/5)**
>
> > The paper doesn't discuss when TusoAI fails or performs poorly, nor does it address practical concerns like how to set appropriate evaluation functions for novel scientific problems, how sensitive the system is to task description quality, or how to validate generated methods when ground truth is unavailable. The computational cost analysis is limited to one brief mention (0.41 for case studies).
>
> ff1y-W3: Practical considerations.
> Thank you for these thoughtful questions. The reviewer has raised 5 practical considerations, which we address in turn:
>
> (i) When TusoAI may fail or perform poorly.
> TusoAI may underperform when (a) each validation experiment is extremely time-consuming, limiting the number of optimization iterations, or (b) in warm-start scenarios where the initial method contains many small interdependent functions, making it harder for TusoAI to identify meaningful optimizations for any individual one. We have added a discussion of these considerations in Section 6 (lines 530–533).
>
> (ii) Setting appropriate evaluation functions for novel problems.
> Detailed evaluation setups are provided in Appendices D and E, referenced in the main text (lines 288-289). In general, we recommend that validation procedures (1) be relatively fast to enable multiple iterations and (2) differ from the final evaluation procedure (akin to training and testing sets) to mitigate overfitting. These principles are now discussed in Section 6 (lines 533–535).
>
> (iii) Sensitivity to task description quality.
> Our task descriptions are concise (e.g., “single-cell batch integration,” “classify sEMG signals into 18 gesture classes”), directly taken from OpenProblems and NASBench-360. This suggests that TusoAI is not highly sensitive to task description quality. We have clarified this in Section 4 (lines 281–282).
>
> (iv) How to validate generated methods when ground truth is unavailable.
> TusoAI relies on a well-defined evaluation metric (e.g., validation accuracy). In our case studies, validation followed established orthogonal approaches: simulation and curated cell type-disease associations for the scDRS study, and fine-mapped eQTL and ABC links for the pgBoost study. More generally, validation of generated methods requires domain expertise, which aligns with our intended user base: domain experts seeking automated method development. We have expanded this discussion in Section 5 (lines 438–440).
>
> (v) Computational cost analysis
>
> We now include detailed computational cost analyses for each benchmark and LLM (Appendix Q). Both TusoAI and AIDE cost under $0.25 per task.
>
> The median 8-hour optimization costs are approximately:
>
> - **$0.17** — GPT-4o-mini
> - **$19.23** — GPT-5
> - **$0.12** — Claude 3.5 Haiku
> - **$14.42** — Claude 4 Sonnet
> - **$2.88** — GPT-oss-120b
>
> Full details are provided in Appendix Q.

---

> ### Author Response · Authors · 2025-11-23
> **Response to Reviewer ff1y (4/5)**
>
> > The case studies claim to discover "9 new associations" and "7 previously unreported links," but these are computational predictions that would require experimental validation to be considered true biological discoveries. The statistical methodology for declaring associations as "novel" versus "missed by previous methods" is not clearly described.
>
> ff1y-W4: Validation of new biological discoveries
>
> Thank you for the comment. The reviewer raised 2 related questions, which we address in turn.
>
> (i) Level of validation of the new discoveries.
> We performed 2 levels of validation. First, the TusoAI-optimized methods were validated using the same experimental setups as in the original studies, ensuring comparable rigor. Specifically, we evaluated the calibration and power of TusoAI + scDRS using the same simulations and real-data benchmarks as in scDRS [5], and assessed the accuracy of TusoAI + pgBoost using the same orthogonal variant-gene link comparisons as in pgBoost [6]. Second, the additional discoveries uniquely identified by the TusoAI-optimized methods were cross-referenced against existing literature to assess biological plausibility. This level of validation is consistent with prior computational studies, where in silico validation is considered acceptable [5, 6]. We have updated Section 5 (lines 438–440) to clarify these points.
>
> (ii) The statistical methodology for declaring association as “novel” vs. “missed by previous methods”.
> We recognize that it is our responsibility to provide a clear exposition. Associations were defined following the same statistical criteria as in the original studies (FDR < 0.05 in scDRS and linking percentile > 95% in pgBoost). A finding is considered “new” if it is detected by our method but not by the original baselines. We have revised Section 5 (lines 459 and 497) to clarify this procedure and consistently used the term “new” (instead of “novel”) throughout the paper to avoid overinterpretation.
>
> [5] Martin Jinye Zhang, Kangcheng Hou, Kushal K Dey, Saori Sakaue, Karthik A Jagadeesh, Kathryn
> Weinand, Aris Taychameekiatchai, Poorvi Rao, Angela Oliveira Pisco, James Zou, et al. Polygenic enrichment distinguishes disease associations of individual cells in single-cell rna-seq data. Nature genetics
>
> [6] Elizabeth Dorans, Karthik Jagadeesh, Kushal Dey, and Alkes L Price. Linking regulatory variants to target genes by integrating single-cell multiome methods and genomic distance. Nature Genetics
>
> > While the authors promise to release code upon publication, key details are missing: which specific LLM API versions were used, how were papers selected from Semantic Scholar (just citation count?), what are the exact prompts for different agents, and how stable are results across different random seeds beyond the reported 95% CIs?
>
> ff1y-W5: Code and implementation details.
>
> Thank you for this comment. The reviewer raises 4 questions regarding implementation details, which we address below.
>
> (i) LLM API versions.
> GPT-4o-mini and GPT-5 were accessed via the OpenAI API, while Claude 3.5 Haiku, Claude 4 Sonnet, and GPT-oss-120b were accessed via the OpenRouter API. We have updated Section 4 (lines 295-296) to clarify this.
>
> (ii) Paper selection from Semantic Scholar.
> Papers were retrieved from Semantic Scholar and ranked purely by citation count. We have updated line 217 to clarify this.
>
> (iii) Prompts for different agents.
> Exact prompts used for each TusoAI component are provided in Appendix U (Appendix L in the original submission).
>
> (iv) Stability across random seeds beyond the reported 95% CIs.
> We performed additional stability analyses across multiple random seeds Appendix L (Appendix F in the original submission). TusoAI achieved highly consistent results across runs. We have updated line 401-403 to reference these findings more clearly.
>
>
> We have included the full code in the supplementary zip file at submission. In this rebuttal, we have also uploaded a cleaner and more organized version of the implementation in the same supplementary package. Please let us know if the reviewer has any additional questions regarding the implementation.

---

> ### Author Response · Authors · 2025-11-23
> **Response to Reviewer ff1y (5/5)**
>
> > How does TusoAI handle the cold start problem for truly novel scientific domains? The knowledge tree construction relies on retrieving papers from Semantic Scholar based on task descriptions. For emerging research areas with limited literature, how does the system bootstrap domain knowledge? Have you tested TusoAI on tasks where relevant literature is scarce or the task description is deliberately vague?
>
> ff1y-Q1: Cold start problem for emerging research areas with limited literature.
>
> Thank you for raising this important question. We conducted additional experiments to assess the impact of literature on the performance in which TusoAI was provided with 0 papers (instead of 10) across 5 benchmark tasks. The results were mixed: performance remained comparable in 2/5 tasks (e.g., SVG, average test score 0.54 vs. 0.55) but dropped noticeably in others (e.g., Denoise, 0.21 vs. 0.35). Importantly, TusoAI still achieved non-trivial performance even without external papers, indicating that the underlying LLM prior provides useful domain knowledge for many tasks. These findings suggest that TusoAI can operate in literature-limited domains but performs best when at least a moderate body of related work exists. This is a realistic assumption, as computational method development in most scientific fields typically spans multiple years or decades. We have updated Section 4.2 (lines 401-403) to report these findings. Here is the median across 5 replicates.
>
> | Papers     | Denoise | SVG  | Decomposition | ECG  | Satellite |
> |------------|---------|------|----------------|------|-----------|
> | 0 papers   | 0.21    | 0.79 | 0.51           | 0.60 | 0.82      |
> | 10 papers  | 0.35    | 0.82 | 0.55           | 0.63 | 0.89      |
> | 50 papers  | 0.21    | 0.73 | 0.49           | 0.66 | 0.88      |
>
> > Can you provide more details on the comparison with Aygun et al. (2025) and clarify the key technical differences? Beyond the unavailability of their code, what are the conceptual and methodological differences between your approach and theirs? This seems critical for establishing the novelty of your work.
>
> ff1y-Q2: Comparison with Aygun 2025.
> Please see our response to ff1yW1 above.

---

### Author Response · Authors · 2025-12-02
**Final Overall Response**

Thank you for your efforts in handling submissions during the recent unexpected situation. To facilitate your evaluation, we provide a concise summary of the review history and our rebuttal progress.

During the rebuttal period, we conducted substantial new analyses, including:

(i) new warm-start, non-biological case study in which TusoAI achieved top performance,

(ii) comprehensive analyses demonstrating that code diversity is a key driver of performance gains, and

(iii) new analyses incorporating more replicates, evaluating the impact of the number of papers, and evaluating the impact of paper summary lengths.

The manuscript has been revised accordingly, incorporating 10 new analyses and updates to 3 main figures, and 13 supplementary sections.

Before reviewer updates were reverted, the mean score had risen from 3.6→5.2. We believe this improvement reflects the clarifications provided during the rebuttal, especially regarding the genetics and single-cell applications. 3 reviewers substantially increased their scores (aNYX 4→8, SvJ8 4→6, x86D 2→4); and 2 of these 3 showed willingness to increase their scores further pending additional clarifications (SvJ8, x86D). A 4th reviewer (kn4q, score 2) noted that most concerns had been addressed and that the remaining issues were primarily related to presentation. We have fully addressed these follow-up comments in our second-round responses, and therefore expect a higher final average score than 5.2 had updates been allowed.

We briefly summarize the status with each reviewer:

**Reviewer ff1y (score 6).**

Raised 5 comments and 2 questions. We believe that we have completely addressed 4 out of these 7 points (W3-W5,Q1) and substantially addressed the remaining (W1,W2,Q2). While we did not receive a response due to the shortened discussion period, we believe the reviewer would be satisfied with our rebuttal.

**Reviewer aNYX (score 4→8).** (“I’ve increased my score from 4 to 8.” in comment Ac1gnVwycD by Reviewer aNYX)

Raised 5 comments and 4 questions. We believe that we have completely addressed 7 out of these 9 points (W1-W3,Q1-Q4) and substantially addressed the remaining (W4,W5). The reviewer stated “Most concerns have been addressed…The paper is well done” (comment Ac1gnVwycD by Reviewer aNYX).

**Reviewer SvJ8 (score 4→6).** (“I am increasing my score to 6” in comment bb6FJKM8Yd by Reviewer SvJ8)

Raised 7 comments and 5 questions. We believe that we have completely addressed 10 out of these 12 points (S1,W1,W3,W5,W6,Q1-Q5) and substantially addressed the remaining (W2,W4). The reviewer mentioned two remaining issues that limited further increase (comment bb6FJKM8Yd by Reviewer SvJ8), which we have adequately addressed in our second-round responses (Response to Reviewer SvJ8 “Thank you for your response”). We thus expect a further score increase.

**Reviewer kn4q (score 2).**

Raised 5 comments and 12 questions. We believe that we have completely addressed 15 out of these 17 points (W1,W2,W5,Q1-Q12) and substantially addressed the remaining (W3,W4). The reviewer acknowledged that most concerns were resolved and that remaining issues were primarily presentation (comment 4MY711LcKi by Reviewer kn4q), which we have addressed in detail in our second-round responses (Response to Reviewer kn4q “Followup response (1/2)”). We thus expect an increased score.

**Reviewer x86D (score 2→4).**(“I have updated my rating from a 2 to a 4.” from comment PBkpTLbjaC by Reviewer x86D)

Raised 3 comments/weaknesses and 15 main questions/feedback. We believe that we have completely addressed 16 out of these 18 points (W1-W3,Q1-Q3,Q5-Q15,) and substantially addressed the remaining (Q4,Q8). Their remaining concern was that the provided code did not run (comment PBkpTLbjaC by Reviewer x86D); this is due to data upload limits and we resolved this by providing runnable code with a smaller dataset in our second-round responses (Response to Reviewer x86D “Followup response and code update”). We therefore expect a further score increase.

In summary, we have substantially addressed all reviewer concerns, and multiple reviewers explicitly indicated they were prepared to raise their scores based on our revisions. We believe the paper is significantly stronger than the frozen score of 5.2 reflects. Thank you for your consideration.

---

### Meta-Review · Area_Chair_Mf9b · 2026-01-10

**Summary:**

Reviewers acknowledge strong empirical results and interesting design choices (for example the knowledge tree and iterative optimization). The main concerns were largely regarding the overall methodological package: whether the contribution is primarily a heuristic assembly of known components, whether the Bayesian update and agentic optimization framing are sufficiently principled and clearly explained, and if claims around scientific discoveries and generality are supported with appropriate validation and transparency. A separate cluster of concerns focused on reproducibility, low quality of code, and the presentation: overall writing quality, missing discussion/conclusion, figure clarity, symbols.

**Reviewer Concerns:**

It appears that the quality of the paper has substantially improved during rebuttal. The following points are addressed in a substantive way. Several presentation and clarity issues were fixed: a new discussion section was added, the abstract was shortened, and figures were reworked for readability (SvJ8, kn4q, x86D). Reproducibility concerns received updates: the paper clarifies LLM access details, added prompt appendices, added additional stability/replicate analyses for ablations, provided hardware and fairness details, and responded directly to old start by running an additional experiment with zero papers and paper-count studies (ff1y, aNYX, SvJ8). The rebuttal also strengthened breadth/generalization by adding a non-biology warm-start case study (NinaPro) and additional analyses linking diversity collapse to optimization plateaus, plus literature support for diversity as a design choice (x86D, ff1y). Finally, the scientific discovery language was clarified.

The following seems still outstanding at the core-idea level. Multiple reviewers’ overarching critique that the method remains highly heuristic and lacks a concrete, principled optimization. They seem concerned that the Bayesian update component is not clearly specified or justified: they found the update rule under-described (including what priors/posteriors are intended) and viewed the overall approach as still largely heuristic, despite added clarification. Relatedly, novelty versus closely related concurrent systems is better differentiated conceptually, but the contribution may still read as an integration of existing ingredients with empirical evidence rather than a clearly new algorithmic advance. On reproducibility, the rebuttal materially improved the code and provided a runnable smaller example, but at least one reviewer’s remaining blocking concern was code reliability and polish, which depends on whether the updated package truly runs cleanly end-to-end.

**Reviewer Scores:**

aNYX has increased their score from 4 to 8. It is likely that reviewer SvJ8 has increased score to 6 but they mentioned they cannot be more positive due to remaining issues. It is likely that reviewer x86D would increase from strong reject to a 4. Reviewer ff1y would likely stay mildly positive (around 6) given the sentiment of their review and the breadth of clarifications. kn4q might move improve slightly if they view the added minimal example and explicit update rule as sufficient clarity, but their central concern about the overall method appears only partially alleviated.

---

### Decision · Program_Chairs · 2026-01-26

Accept (Poster)